# Serine ADP-ribosylation marks nucleosomes for ALC1-dependent chromatin remodeling

Jugal Mohapatra[1], Kyuto Tashiro[1], Ryan L Beckner[1], Jorge Sierra[1], Jessica A Kilgore[1,2], Noelle S Williams[1,2], Glen Liszczak[1]*

[1]Department of Biochemistry, The University of Texas Southwestern Medical Center, Dallas, United States; [2]Preclinical Pharmacology Core, Department of Biochemistry, University of Texas Southwestern Medical Center, Dallas, United States

*For correspondence: glen.liszczak@utsouthwestern. edu

Competing interest: The authors declare that no competing interests exist.

**Abstract** Serine ADP-ribosylation (ADPr) is a DNA damage-induced post-translational modification catalyzed by the PARP1/2:HPF1 complex. As the list of PARP1/2:HPF1 substrates continues to expand, there is a need for technologies to prepare mono- and poly-ADP-ribosylated proteins for biochemical interrogation. Here, we investigate the unique peptide ADPr activities catalyzed by PARP1 in the absence and presence of HPF1. We then exploit these activities to develop a method that facilitates installation of ADP-ribose polymers onto peptides with precise control over chain length and modification site. Importantly, the enzymatically mono- and poly-ADP-ribosylated peptides are fully compatible with protein ligation technologies. This chemoenzymatic protein synthesis strategy was employed to assemble a series of full-length, ADP-ribosylated histones and show that ADPr at histone H2B serine 6 or histone H3 serine 10 converts nucleosomes into robust substrates for the chromatin remodeler ALC1. We found ALC1 preferentially remodels 'activated' substrates within heterogeneous mononucleosome populations and asymmetrically ADP-ribosylated dinucleosome substrates, and that nucleosome serine ADPr is sufficient to stimulate ALC1 activity in nuclear extracts. Our study identifies a biochemical function for nucleosome serine ADPr and describes a new, highly modular approach to explore the impact that site-specific serine mono- and poly-ADPr have on protein function.

## Editor's evaluation

Poly-ADP-ribosylation (poly-ADPr) is a major histone modification that plays critical roles in DNA damage. However careful mechanistic dissection of the role of poly-ADPr has been challenging as the modification is found on multiple proteins and there is heterogeneity in terms of poly-ADP-ribosylation chain length and amino acid location of attachment. The PARP1-dependent semi-synthetic strategy developed by the authors allows generation of nucleosomes with mono ADP ribose and defined lengths of poly-ADPr chains at specific histone serine residues. The utility of this method is clearly demonstrated by the authors' findings that ALC1, a chromatin remodeler that recognizes poly-ADPr is stimulated substantially by the presence of poly-ADPr on H2A and H3.

## Introduction

Protein ADP-ribosylation (ADPr) has been implicated in diverse mammalian cellular signaling pathways (*Gupte et al., 2017*). In this process, the ADP-ribose moiety from an NAD[+] co-factor is deposited onto one of several chemically distinct amino acid side chain functionalities (*Daniels et al., 2015*). In cells, proteins can be modified with a mono-ADP-ribose adduct or variable length

ADP-ribose polymers that emanate from specific protein sites, a process henceforth referred to as poly-ADPr. Among the 17-member poly(ADP-ribose) polymerase (PARP) enzyme family, PARP1/2 have emerged as the most extensively studied owing to the success of PARP1/2 inhibitors to treat DNA repair-deficient cancers (*Lord and Ashworth, 2017*). As the clinical utility of PARP1/2 inhibitors continues to expand, it is critical to understand how PARP1/2-dependent ADPr impacts cellular physiology and disease. In light of intense PARP1/2 substrate identification efforts (*Bonfiglio et al., 2017*; *Larsen et al., 2018*; *Leidecker et al., 2016*), several creative methods have been developed to install serine mono-ADPr onto synthetic peptides for biochemical interrogation (*Bonfiglio et al., 2020*; *Voorneveld et al., 2018*; *Zhu et al., 2020*). However, these technologies have been limited to relatively short peptide constructs. Additionally, no methods exist to reconstitute well-defined ADP-ribose chains at specific sites on isolated proteins for functional analysis. Hence, there is a dearth of mechanistic insight into how specific PARP1/2:HPF1-dependent mono- and poly-ADPr events regulate protein function.

Upon binding to single or double-stranded DNA breaks, PARP1/2 undergo conformational changes that induce the formation of a catalytically competent complex with $NAD^+$ and the PARP1/2-interacting protein HPF1 (*Benjamin and Gill, 1980*; *Dawicki-McKenna et al., 2015*; *Gibbs-Seymour et al., 2016*; *Langelier et al., 2012*; *Suskiewicz et al., 2020*). It has long been appreciated that DNA damage-induced ADPr has a profound effect on chromatin architecture through a variety of proposed mechanisms (*Poirier et al., 1982*; *Ray Chaudhuri and Nussenzweig, 2017*; *Tulin and Spradling, 2003*). Indeed, there are several ATP-dependent chromatin remodeling enzymes that localize to damage sites in an ADPr-dependent manner and contribute to decompaction of higher order chromatin structure, ultimately increasing repair factor accessibility (*Ahel et al., 2009*; *Chou et al., 2010*; *Luijsterburg et al., 2016*; *Smeenk et al., 2013*). One such chromatin remodeler, ALC1, harbors a macrodomain module that has been shown to specifically interact with tri-ADP-ribose (*Singh et al., 2017*). This binding event relieves an autoinhibited ALC1 conformation and activates the ATPase domain that powers nucleosome remodeling (*Lehmann et al., 2017*; *Singh et al., 2017*). ALC1 activation via ternary complex formation with auto-ADP-ribosylated PARP1 and nucleosomes has been extensively studied (*Gottschalk et al., 2009*; *Gottschalk et al., 2012*; *Lehmann et al., 2017*; *Singh et al., 2017*), and it has been suggested that other DNA-bound, ADP-ribosylated proteins may contribute to this process. However, it remains unclear which PARP1/2:HPF1 substrates and corresponding modification sites can lead to ALC1 activation, and if any are sufficient to do so in the absence of automodified PARP1. Such questions surrounding ALC1 regulation are increasingly important as recent studies show that abrogating ALC1 activity vastly increases the efficacy of PARP inhibitors (*Blessing et al., 2020*; *Verma et al., 2021*) and may even be useful for treatment of PARP inhibitor-resistant cancers (*Juhász et al., 2020*).

The core histones H2B and H3 are consistently identified as some of the most abundantly modified PARP1/2:HPF1 substrates (*Bonfiglio et al., 2017*; *Huletsky et al., 1989*; *Larsen et al., 2018*). While much effort has been directed toward deciphering the regulatory mechanisms that govern serine ADPr (*Bilokapic et al., 2020*; *Bonfiglio et al., 2017*; *Bonfiglio et al., 2020*; *Gibbs-Seymour et al., 2016*; *Palazzo et al., 2018*; *Suskiewicz et al., 2020*), the functional consequences of specific nucleosome serine ADPr sites remain unclear. We and others have demonstrated that histone H2B serine 6 (H2BS6) and histone H3 serine 10 (H3S10) are the primary PARP1/2:HPF1 target sites in biochemical and cellular systems (*Liszczak et al., 2018*; *Palazzo et al., 2018*). Building upon these studies, we sought to determine how mono- and poly-ADPr on H2BS6 and H3S10 contribute to PARP1/2-dependent DNA repair activities such as ATP-dependent chromatin remodeling.

Here we employ an HPLC/MS-based analysis to investigate PARP1-dependent peptide ADPr activity in the absence and presence of HPF1. Reaction analyses guided the development of an approach that combines peptide chemistry, enzymatic catalysis, and protein ligation technologies to generate full-length proteins that bear mono- or poly-ADPr at user-defined serine sites. Key to this method is the separation of two enzyme-based peptide modification steps: 1. mono-ADPr of unmodified peptides by the PARP1:HPF1 complex, and 2. ADP-ribose chain elongation from mono-ADP-ribosylated peptides by the uncomplexed PARP1 enzyme. We prepare eight unique, semi-synthetic ADP-ribosylated nucleosomes and demonstrate that histone serine poly-ADPr marks nucleosomes for ALC1-dependent chromatin remodeling, with ALC1 remodeling rate constants increasing up to ~370 fold relative to unmodified nucleosome substrates. Additional data support a model wherein

nucleosome serine ADPr is sufficient to initiate ALC1-dependent chromatin structure alterations with a high degree of spatial precision.

## Results

### An HPLC/MS-based approach to analyze peptide ADPr by PARP1:HPF1

While synthetic and enzyme-based methodologies exist to prepare mono-ADP-ribosylated peptides and proteins (*Bonfiglio et al., 2020*; *Hananya et al., 2021*; *Voorneveld et al., 2018*; *Zhu et al., 2020*), installation of poly-ADP-ribose is synthetically more complex and has not been reported. Therefore, we envisioned an enzyme-based approach that employs the PARP1:HPF1 complex to modify specific serine sites on synthetic peptides with homogenous ADP-ribose polymers. A similar elegant approach was recently reported by the Matic group to prepare mono-ADP-ribosylated peptides, which included H2B and H3 tail constructs (*Bonfiglio et al., 2020*). However, in that study, a post-reaction poly-ADP-ribose glycohydrolase (PARG) treatment was carried out to reduce any poly-ADP-ribosylated species to the mono-ADP-ribose adduct. Our method is unique in that we developed an RP-HPLC-MS-based assay to simultaneously monitor recombinant PARP1:HPF1 complex activity on a peptide substrate and separate distinct mono- and poly-ADP-ribosylated peptide products (*Figure 1A*).

We began our study by incubating a synthetic histone H3 peptide (amino acids 1–20) that contains a single known serine target site (H3S10) with the PARP1:HPF1 complex, $NAD^+$, and stimulating DNA. Multiple H3 peptide product peaks were observed via chromatography-based reaction analysis. ESI-MS characterization revealed a single, unique mass in each HPLC product peak, which corresponded to an H3 peptide modified with mono-, di-, tri-, or tetra-ADP-ribose (henceforth $H3S10ADPr_n$) (*Figure 1B* and *Supplementary file 1*). Notably, all products are sensitive to the H3S10A mutation, indicating the presence of an ADP-ribose chain that elongates from the S10 site (*Figure 1B*). Thus, each individual peptide product corresponding to mono-, di-, tri-, or tetra-ADP-ribosylated H3S10 can be separated via RP-HPLC.

We next treated ADPr reactions with recombinant ADP-ribosylhydrolase enzymes to validate the modification site and chemical identity of modified peptide products (*Figure 1C*). Analysis via HPLC-MS demonstrates that PARG (*Slade et al., 2011*) treatment quantitatively converts all observed ADP-ribosylated H3 peptide products to the mono-ADP-ribosylated species, which is consistent with a single modification site (*Figure 1D*). When the serine-specific ADP-ribosylhydrolase 3 (ARH3) (*Fontana et al., 2017*) enzyme is substituted for PARG, all ADP-ribosylated species are converted to the unmodified H3 peptide, thus confirming a serine-linked modification (*Figure 1D*). An established LC-MS/MS analysis protocol (*Chen et al., 2018*) was used to determine that the peptide-linked ADP-ribose chains were principally linear, with negligible branching ( < 0.03%) (*Figure 1—figure supplement 1A* and *Supplementary file 2*).

The workflow and characterization strategies described here were next implemented to install ADP-ribose chains at the known PARP1:HPF1 target site on a synthetic H2B peptide (amino acids 1–16). Despite the presence of two serine residues in the H2B peptide, our mutagenesis and ADP-ribosylhydrolase-based characterizations confirmed H2BS6 as the sole acceptor residue (*Figure 1—figure supplement 1B* and C). Notably, while conversion of up to 1 mM (~20 mg) of unmodified H2B or H3 peptides to the $H2BS6ADPr_1$ or $H3S10ADPr_1$ products could be routinely achieved, a more scalable approach for peptide poly-ADPr would be required to deploy these molecules in protein ligation reactions and biochemical assays.

Analysis of the PARP2:HPF1 structure suggests that HPF1 binding, while required for serine ADPr, would interfere with the PARP1/2 ADP-ribose chain elongation mechanism (*Suskiewicz et al., 2020*). This observation is consistent with several recent reports that show HPF1-dependent shortening of PARP1/2-catalyzed ADP-ribose chains in cellular and biochemical assays (*Bonfiglio et al., 2020*; *Gibbs-Seymour et al., 2016*; *Rudolph et al., 2021*). We therefore hypothesized that the concentration of HPF1 in the peptide modification reaction may affect the final distribution of our mono- and poly-ADP-ribosylated peptide products. To explore this, an HPF1 titration from 5 µM to 100 µM was performed in an ADPr reaction containing the H3 peptide. Notably, unmodified peptide starting material and ADP-ribosylated peptide products could be separated via RP-HPLC and quantified by chromatogram peak integration at $A_{214}$ and $A_{280}$, respectively (see Materials and methods for details). Near quantitative conversion ( > 95%) of the unmodified H3 substrate to ADP-ribosylated products

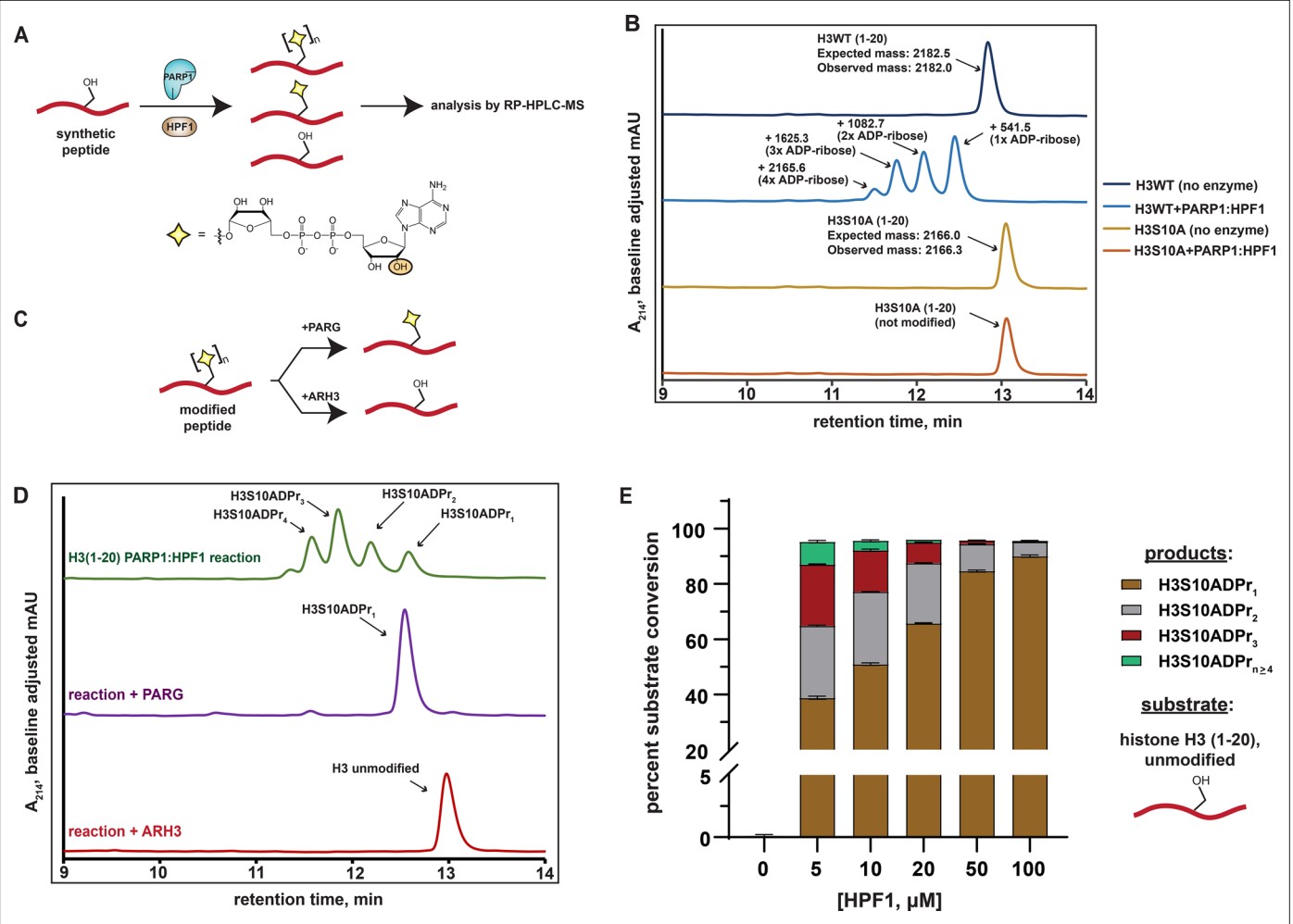

**Figure 1.** Analysis of serine mono- and poly-ADPr by the PARP1:HPF1 complex on synthetic peptide substrates. (**A**) A schematic showing the workflow employed to analyze peptide poly-APDr by the recombinant PARP1:HPF1 complex. Peptide products are separated by polymer length via RP-HPLC. The yellow star represents a serine-linked ADP-ribose modification, 'n' represents variable polymer length, and the orange circle indicates the site of linear ADP-ribose polymerization. (**B**) RP-HPLC and MS analysis of substrate peptides (histone H3 wild-type or S10A mutant, amino acids 1–20) and corresponding PARP1:HPF1 reaction products (for raw MS data, see ***Supplementary file 1***). RP-HPLC gradients are from 0 to 35% Solvent B (2–22 min). (**C**) A schematic describing the ADP-ribosylhydrolase-based characterization strategy. Enzymes and their respective reaction products are depicted. (**D**) RP-HPLC traces from PARG- or ARH3-treated H3 peptide ADPr reactions that were optimized for ADP-ribose chain elongation. The number of ADP-ribose units was verified by MS analysis. (**E**) Product analysis of a PARP1 ADPr reaction in the presence of increasing HPF1 concentrations. Histone H3 substrate peptide starting material and each unique ADP-ribosylated product were quantified via HPLC chromatogram peak integration (see Methods and ***Figure 1—figure supplement 1D***). The columns represent the percent substrate conversion to each ADP-ribosylated product. Data are represented as mean ± s.d. (n = 3).

The online version of this article includes the following figure supplement(s) for figure 1:

**Figure supplement 1.** The recombinant PARP1:HPF1 complex installs linear poly-ADP-ribose chains at biologically relevant target sites.

was achieved at HPF1 concentrations as low as 5 µM (***Figure 1E***). Interestingly, we observed a gradual increase in mono-ADPr activity and decrease in poly-ADPr activity as HPF1 is titrated into the reaction (***Figure 1E*** and ***Figure 1—figure supplement 1D***). In the 5 µM HPF1 reaction, the mono-ADP-ribosylated peptide represents ~41% of the total product, with the remaining ~59% comprising a distribution of di- to penta-ADP-ribosylated peptide. In the 100 µM HPF1 reaction, mono-ADP-ribosylated peptide increases to ~94% of the total product, with di-ADP-ribose representing the remaining ~6%. This is consistent with a mechanism wherein PARP1:HPF1 complex formation switches PARP1 activity from an ADP-ribose chain elongator to a mono-ADP-ribosyltransferase. Indeed, these experimental data are congruent with the structure-based hypothesis put forth by Suskiewicz, et al. that HPF1 limits PARP1/2 activity to mono-ADPr.

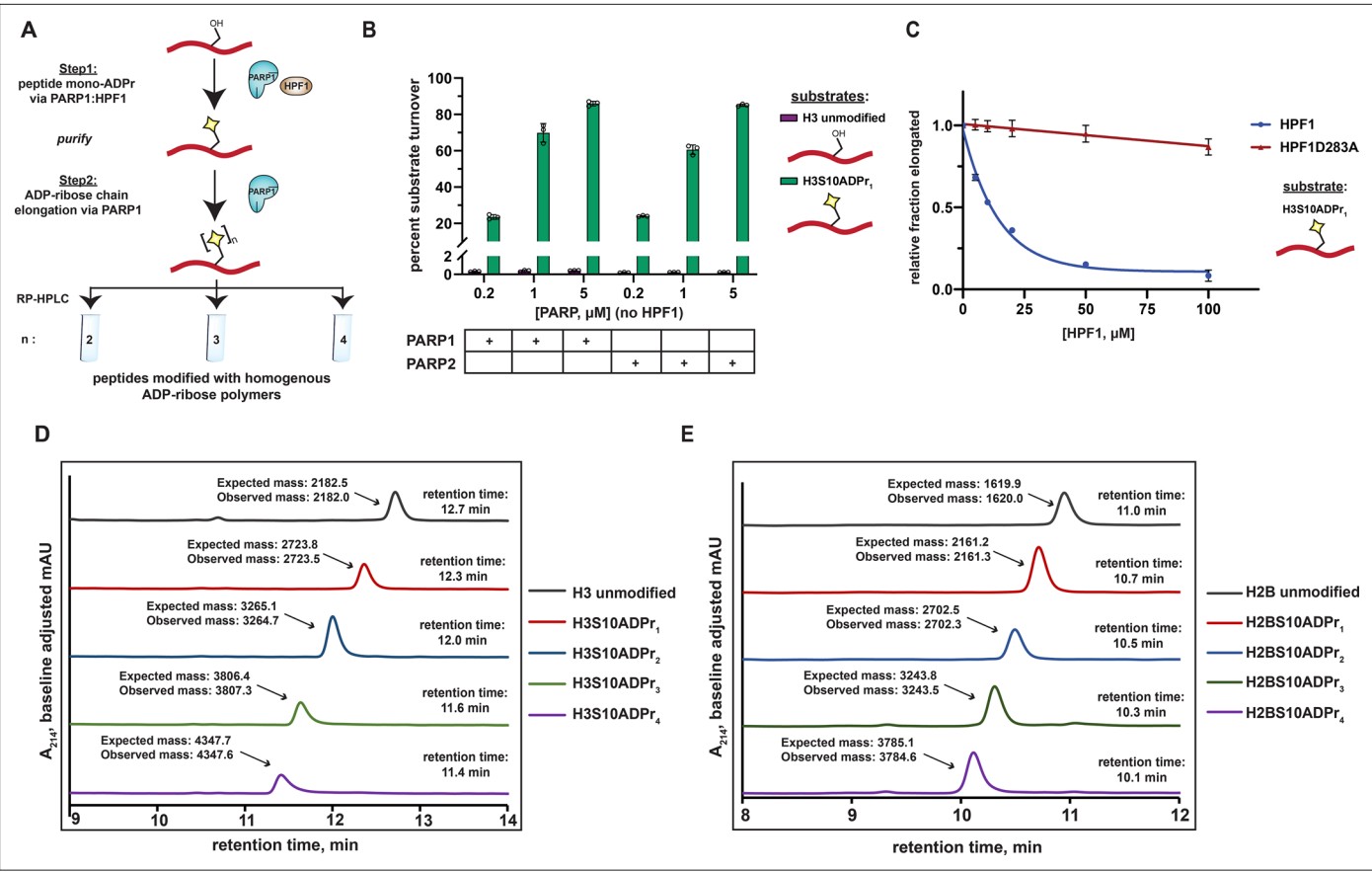

**Figure 2.** A two-step enzymatic process to prepare poly-ADP-ribosylated peptides with defined ADP-ribose chain lengths. (**A**) A schematic showing the two-step enzymatic procedure implemented to synthesize and purify poly-ADP-ribosylated peptides. The mono-ADP-ribosylated peptide product from Step 1 was purified using preparative RP-HPLC prior to use in Step 2. (**B**) Substrate turnover analysis of PARP1 and PARP2 ADPr reactions in the absence of HPF1. Purple bars represent total percent turnover of an unmodified H3 peptide to mono- or poly-ADP-ribosylated products. Green bars represent total percent turnover of the H3S10ADPr$_1$ peptide to poly-ADP-ribosylated products (for poly-ADP-ribosylated product distribution, see *Figure 2— figure supplement 1A* and B). Data are represented as mean ± s.d. (n = 3). (**C**) Analysis of PARP1 elongation activity on the H3S10ADPr$_1$ peptide substrate in the presence of increasing amounts of HPF1 or HPF1D283A. Fraction elongated represents the fraction of H3S10ADPr$_1$ peptide converted to poly-ADP-ribosylated products. Data are normalized to fraction of substrate elongated in the absence of HPF1. Data are represented as mean ± s.d. (n = 3). The curves represent the fit of the data into a non-linear regression model for one-phase exponential decay. (**D**) RP-HPLC and MS analysis of mono- and poly-ADP-ribosylated H3 peptides that have been purified to homogeneity via semi-preparative HPLC. (**E**) As in (**D**), but for H2B (amino acids 1–16) peptides.

The online version of this article includes the following source data and figure supplement(s) for figure 2:

**Figure supplement 1.** PARP1 and PARP2 efficiently elongate ADP-ribose chains from mono-ADP-ribosylated peptides.

**Figure supplement 1—source data 1.** Uncropped SDS-PAGE gel from *Figure 2—figure supplement 1D* (Coomassie stain).

**Figure supplement 2.** The two-step enzymatic process is broadly applicable to install poly-ADP-ribosylation at PARP1:HPF1 target sites.

## Synthesis of poly-ADP-ribosylated peptides via two enzymatic steps

Based on the mechanistic interpretation described above, we surmised that PARP1 would display efficient ADP-ribose chain elongation activity on mono-ADP-ribosylated peptides in the absence of HPF1 in our reconstituted system. To investigate this, we employed our purified H3S10ADPr$_1$ peptide as a substrate in PARP1 activity assays that lack HPF1 (*Figure 2A*). Importantly, we maintained all reaction conditions, substrate concentrations, and stimulating DNA concentrations described for the PARP1:HPF1 activity assays. Strikingly, incubation of the H3S10ADPr$_1$ peptide with PARP1 resulted in robust ADP-ribose chain elongation at all enzyme concentrations tested (0.2, 1, and 5 μM). Nearly 70% conversion of the H3S10ADPr$_1$ substrate to poly-ADP-ribosylated products was achieved at 1 μM PARP1 (*Figure 2B* and *Figure 2—figure supplement 1A*). The di-, tri-, and tetra-ADP-ribosylated species were the most abundant products with yield decreasing precipitously for chains greater than

four units in length (*Figure 2—figure supplement 1A*). Notably, PARP2 also catalyzes ADP-ribose chain elongation from the H3S10ADPr$_1$ substrate and similar polymerization activity was observed with both PARP1 and PARP2 on the H2BS6ADPr$_1$ substrate (*Figure 2B* and *Figure 2—figure supplement 1B* and C).

To further characterize the inhibitory effect that HPF1 has on PARP1-dependent ADP-ribose chain elongation, we incubated PARP1 with the H3S10ADPr$_1$ substrate peptide in the presence of increasing concentrations of HPF1. As expected, HPF1 exhibits dose-dependent inhibition of PARP1-catalyzed ADP-ribose polymerization from the mono-ADP-ribosylated substrate, with 50% inhibition occurring at ~14 µM HPF1 for 1 µM PARP1. A binding-deficient HPF1 mutant (D283A) (*Rudolph et al., 2021*; *Suskiewicz et al., 2020*) is unable to appreciably inhibit ADP-ribose polymerization (*Figure 2C* and *Figure 2—figure supplement 1D* and E). These data complement our unmodified peptide substrate:HPF1 titration analysis and provide additional evidence that the PARP1:HPF1 complex is a dedicated mono-ADP-ribosyltransferase.

Importantly, by first isolating mono-ADP-ribosylated peptides from a PARP1:HPF1 reaction for use in a PARP1 elongation reaction, each poly-ADP-ribosylated H2BS6 and H3S10 product (up to four ADP-ribose units in length) could now be purified to homogeneity in milligram quantities for downstream applications (*Figure 2D and E*). The broad applicability our peptide poly-ADPr strategy was further validated with additional known PARP1:HPF1 target sequences (*Bonfiglio et al., 2020*) including TMA16 (amino acids 2–19, target residue S9), a fragment of the PARP1 automodification domain (amino acids 501–515, target residue S507), and a secondary histone H3 site (amino acids 21–34; target residue S28). The mono-, di-, tri-, and tetra- ADP-ribosylated species were isolated for each of these peptides (*Figure 2—figure supplement 2A* and B). Thus, PARP1 can dependably elongate ADP-ribose chains from peptides that have been 'primed' with serine mono-ADP-ribose by PARP1:HPF1. We do note that overall poly-ADP-ribosylated product yields vary depending upon

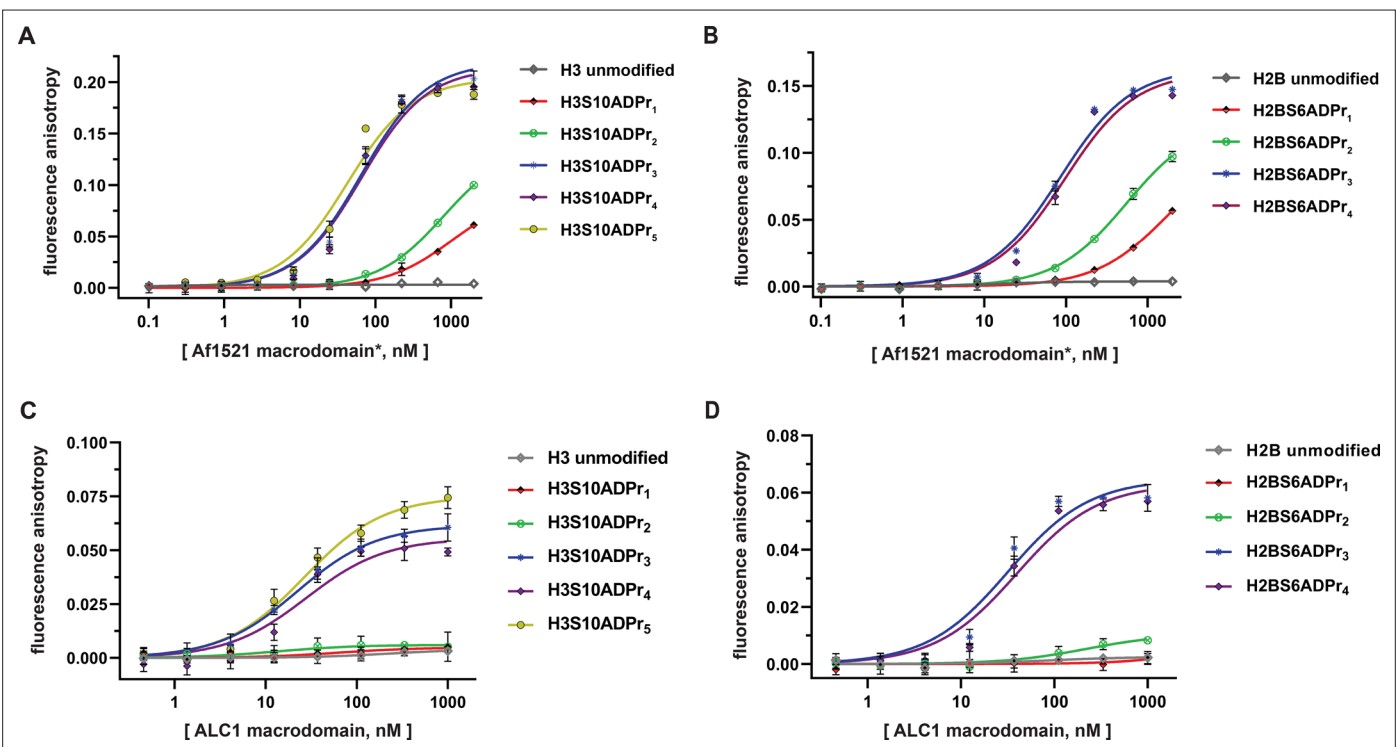

**Figure 3.** The ALC1 macrodomain engages ADP-ribosylated H2B and H3 peptides with equal affinity. (**A**) Fluorescence polarization (FP) assays to evaluate binding affinities of different ADP-ribosylated, fluorescein-labeled H3 (1–20) peptides to the Af1521 macrodomain. Data are represented as mean ± s.d. (n = 3). All curves represent fit of the data into a non-linear regression equation for one-site, specific binding (for $K_{d, app}$ values, see *Supplementary file 4*). *The Af1521 macrodomain is from the commercially available pan-ADP-ribose detection reagent. (**B**) As in (**A**), but with fluorescein-labeled H2B (1-16) peptides. (**C**) FP assays as described in (**A**) to evaluate binding affinities of ADP-ribosylated, fluorescein-labeled H3 (1–20) peptides to the ALC1 macrodomain. (**D**) As in (**C**), but with fluorescein-labeled H2B (1-16) peptides.

target peptide identity, but all reactions could be optimized to obtain milligram quantities of each unique product (see 'Materials and methods' for details).

## ADP-ribosylated H2B and H3 peptides engage the ALC1 macrodomain with equal affinity

Extensive precedent exists demonstrating that chromatin remodeling enzymes are regulated by modifications on the nucleosome substrate (*Clapier and Cairns, 2012*; *Hauk et al., 2010*). The Ladurner lab recently reported that the ALC1 macrodomain exhibits high affinity ($K_d$ ~10 nM) for free tri-ADP-ribose with little to no binding detectable for free mono- and di-ADP-ribose molecules (*Singh et al., 2017*). We therefore chose to pursue ALC1 for our initial ADP-ribosylated histone peptide interaction studies. Nine fluorescently-labeled, ADP-ribosylated histone peptides (H2BS6ADPr$_{1-4}$ and H3S10ADPr$_{1-5}$) were prepared for fluorescence polarization-based interaction assays (*Supplementary file 3*). We note that the ADP-ribose polymerization reaction is more efficient with the H3 peptide and hence longer peptide-conjugated ADP-ribose chains could be isolated relative to H2B. Initial assay development was carried out by titrating a commercially available pan-ADP-ribose detection reagent (an Af1521 macrodomain-Fc region fusion) (*Gibson et al., 2017*) into each peptide. This reagent exhibits ADPr-dependent binding for all H2B and H3 peptides, with affinity decreasing precipitously for chains less than three ADP-ribose units in length (*Figure 3A and B*, and *Supplementary file 4*).

Similar experiments were performed by titrating the ALC1 macrodomain into each fluorescently-labeled histone peptide for apparent dissociation constant ($K_{d, app}$) calculations. Consistent with free ADP-ribose binding preferences (*Singh et al., 2017*), the mono- and di-ADP-ribosylated H2B and H3 peptides failed to appreciably interact with the ALC1 macrodomain. Contrastingly, all tri-, tetra-, and penta-ADP-ribosylated peptides are high-affinity ligands with $K_{d, app}$ ranging from ~21 to 37 nM (*Figure 3C and D*, and *Supplementary file 4*). Considering the H2BS6ADPr$_{3-4}$ and H3S10ADPr$_{3-5}$ peptides exhibit similar affinities, we concluded that the tri-ADP-ribose modification is likely sufficient for optimal ALC1 macrodomain:peptide engagement. These data also indicate that while the ALC1 macrodomain engages the H2BS6 and H3S10-modified peptides, it does not exhibit sequence-based preference for either site.

## Preparation of full-length, homogenously ADP-ribosylated histone proteins and assembly into nucleosomes

Chromatin remodelers comprise multiple domains that function synergistically to recognize nucleosome substrates and mobilize histone proteins (*Bowman and Poirier, 2015*). This phenomenon implies that macrodomain-ligand specificity may not represent the sole determinant of ALC1 substrate preference. To address this, we sought to analyze full-length ALC1 remodeling activity in the context of ADP-ribosylated nucleosome substrates. The first step towards reconstituting modified nucleosomes requires preparation of full-length, ADP-ribosylated histones. We generated a series of ADP-ribosylated H2B and H3 peptides with C-terminal thioesters to enable an eventual native chemical ligation reaction to the remainder of the corresponding histone fragment (*Figure 4A*). The following six semi-synthetic, full-length histones were prepared: H2BS6ADPr$_1$, H2BS6ADPr$_3$, H2BS6ADPr$_4$, H3S10ADPr$_1$, H3S10ADPr$_3$, and H3S10ADPr$_4$ (*Supplementary file 5*). The tri- and tetra-ADP-ribosylated H2B and H3 proteins were essential to probe the effect of chain length and nucleosome modification site on ALC1 activation. Mono-ADP-ribosylated histones were prepared to serve as negative controls and to further corroborate ALC1 macrodomain interaction results. All final protein products were characterized via HPLC/MS analysis and determined to be >95% pure, hence validating our workflow to reconstitute homogenously ADP-ribosylated proteins (*Figure 4B* and *Supplementary file 5*).

Each of the six semi-synthetic ADP-ribosylated histones were combined with the necessary recombinant histones to form stable histone octamer complexes (henceforth labeled as H2BS6ADPr$_n$ or H3S10ADPr$_n$, depending on the modified histone they possess) via established protocols (*Luger et al., 1999*). We also prepared an octamer that contains both H2BS6ADPr$_3$ and H3S10ADPr$_3$ (H2BS6/H3S10ADPr$_3$), and another that contains both H2BS6ADPr$_4$ and H3S10ADPr$_4$ (H2BS6/H3S10ADPr$_4$). Following purification via gel filtration chromatography, octamer quality and ADPr stability was determined via SDS-PAGE/western blot analysis. Histone detection via western blotting with H2B and H3 antibodies revealed single, distinct species for each ADP-ribosylated H2B and H3 histone (*Figure 4C and D*). We found that ADP-ribose chain length is inversely proportional to histone gel

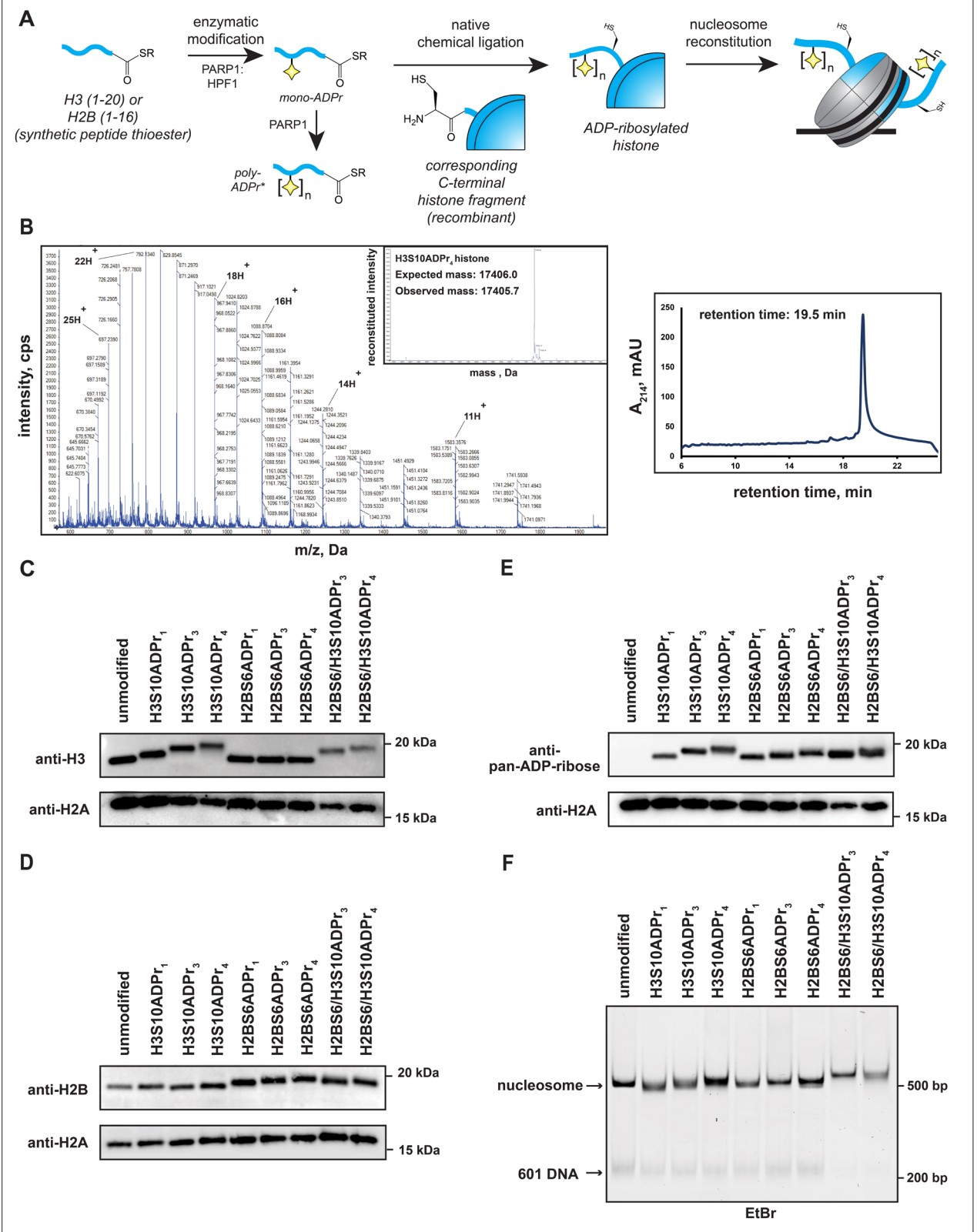

**Figure 4.** Installation of homogenous ADP-ribose polymers onto reconstituted nucleosomes via a chemoenzymatic strategy. (**A**) A schematic depicting the protein semi-synthesis-based strategy to install homogenous ADP-ribose polymers at specific sites on histone proteins. The nucleosome cartoon includes DNA (black line), as well as the histone protein octamer core (gray = recombinant histones, blue = semi-synthetic histone). *The poly-ADP-ribosylated peptides are separated via HPLC to yield homogenous species prior to the ligation reaction. (**B**) Representative HPLC/MS characterization

*Figure 4 continued on next page*

*Figure 4 continued*

of the full-length H3S10ADP$_4$ protein. Raw ESI-MS spectra, MS deconvolution, and RP-HPLC chromatogram are shown. RP-HPLC gradient is from 0 to 80% Solvent B (2–22 min). For additional histone HPLC and MS characterizations, see *Supplementary file 5*. (**C**) Western blot analysis of histone H3 following nucleosome assembly. ADP-ribose-dependent gel migration shifts (12% bis-tris SDS-PAGE gel in MES running buffer) demonstrate sample homogeneity. (**D**) Histone H2B analysis as described in (**C**). (**E**) Pan-ADP-ribose western blot analysis of all assembled nucleosomes. (**F**) Native gel analysis of assembled nucleosomes. Single nucleosome bands and trace levels of free 601 DNA demonstrate sample homogeneity and assembly efficiency. EtBr = ethidium bromide stain.

The online version of this article includes the following source data for figure 4:

**Source data 1.** Uncropped western blots and TBE gel from *Figure 4*.

migration distance, suggesting that single migration bands for H2B and H3 are a reliable indicator of modification stability and sample homogeneity. Additionally, all gel species that correspond to ADP-ribosylated histones exhibited strong signal in a pan-ADP-ribose detection blot (*Figure 4E*). Next, the eight ADP-ribosylated octamers were assembled into unique nucleosomes using a DNA template that contains the '601' nucleosome positioning sequence and is compatible with a previously reported restriction enzyme accessibility (REA)-based chromatin remodeling assay (see Materials and methods for details) (*He et al., 2006*). Nucleosome quality was analyzed on a native polyacrylamide TBE gel, which shows a single, distinct nucleosome species for each assembly and only trace levels of free 601 DNA (*Figure 4F*). Notably, ADP-ribose has a polymer length-dependent effect on nucleosome gel migration patterns, again indicating sample homogeneity and modification stability. We also note that all H2B and H3 histones have an alanine-to-cysteine mutation at the respective ligation junction (H2BA17C and H3A21C). To ensure that no disulfide bonds were present, native TBE gel analyses were conducted after incubating the nucleosomes in the chromatin remodeling buffer and conditions. Additionally, a recent publication has shown that mono-ADPr is stable under desulfurization conditions, and conversion to the native alanine residue can be employed if desired (*Hananya et al., 2021*). We thus concluded that all of our site-specifically ADP-ribosylated histones could be efficiently incorporated into nucleosomes for downstream chromatin remodeling experiments.

## Serine ADPr converts nucleosomes into robust ALC1 substrates

Recombinant, full-length ALC1 was isolated to determine chromatin remodeling rate constants with each ADP-ribosylated nucleosome substrate. The DNA from each remodeling reaction was isolated at various time points and remodeling-dependent restriction enzyme cleavage was visualized on a polyacrylamide TBE gel and quantified via densitometry (*Figure 5A* and *Figure 5—figure supplement 1A*). Consistent with the macrodomain interaction results, ALC1 exhibits relatively low remodeling rate constants ( $< 3 \times 10^{-4}$ min$^{-1}$) with unmodified and mono-ADP-ribosylated nucleosome substrates (*Figure 5B* and *Supplementary file 6*). Contrastingly, robust chromatin remodeling activity is observed with all nucleosomes that contain tri- or tetra-ADP-ribose at the H2B or H3 sites. The H2BS6/H3S10ADPr$_4$ nucleosome has the most striking effect on the ALC1 remodeling rate constant, which increases ~370 fold relative to the unmodified nucleosome. Further rate constant analyses show that ALC1 exhibits modest preference for the H2BS6 modification site and tetra-ADP-ribose polymers (*Figure 5B*). Importantly, a macrodomain deletion construct of ALC1 (1–673) showed no preference towards H2BS6ADPr$_4$ nucleosomes over unmodified nucleosomes (*Figure 5—figure supplement 1B*). Therefore, the modified histone tail:macrodomain interaction is important for ALC1 substrate selectivity. We also found that freely diffusing macrodomain ligands are unable to appreciably stimulate ALC1-dependent nucleosome remodeling activity, as observed in a reaction comprising H2BS6ADPr$_4$ or H3S10ADPr$_4$ peptide, ALC1 and unmodified nucleosomes (*Figure 5C* and *Figure 5—figure supplement 1A* and C). Furthermore, freely diffusing macrodomain ligands do not inhibit nucleosome ADPr-dependent stimulation of ALC1 (*Figure 5—figure supplement 1D*), suggesting that substrate engagement is also influenced by additional ALC1:nucleosome interfaces, of which several have been identified (*Bacic et al., 2021*; *Lehmann et al., 2020*).

We next asked how ALC1 activation by nucleosome serine ADPr compares to activation by auto-ADP-ribosylated PARP1 (*Gottschalk et al., 2009*; *Gottschalk et al., 2012*; *Lehmann et al., 2017*; *Singh et al., 2017*). As previously described, chromatin remodeling reactions were performed on unmodified nucleosome substrates in the presence of NAD$^+$ and PARP1 (*Gottschalk et al., 2009*; *Gottschalk et al., 2012*). In this experimental setup, PARP1 maintains auto-ADPr activity but is unable

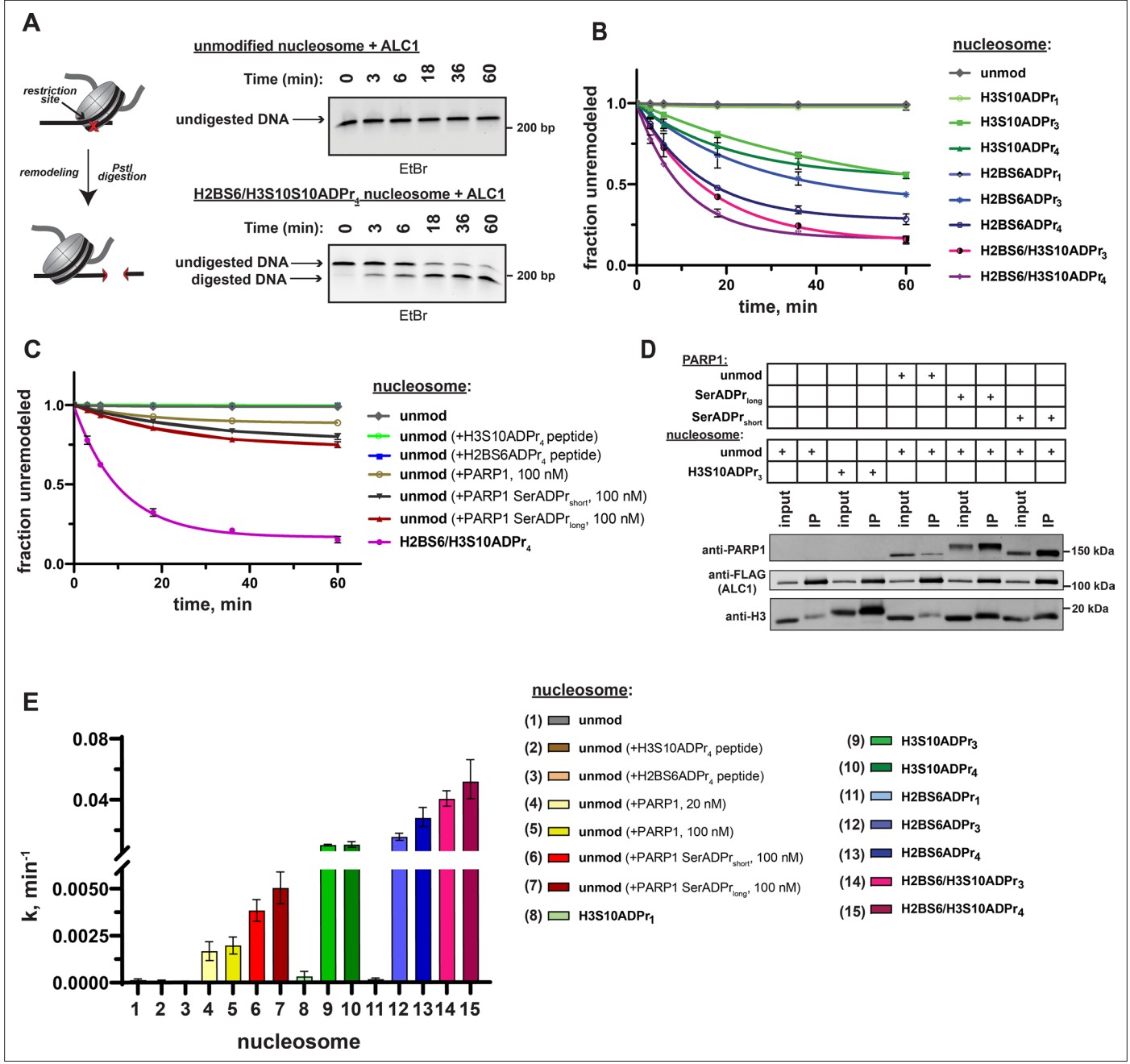

**Figure 5.** ADPr at H2BS6 and H3S10 convert nucleosomes into robust ALC1 substrates. (**A**) Schematic depicting the REA assay for chromatin remodeling and representative TBE gel analyses of recombinant ALC1 activity on unmodified or H2BS6/H3S10ADPr$_4$ nucleosomes. (**B**) ALC1 nucleosome remodeling assay time-course wherein each reaction comprises ALC1 and the indicated nucleosome ('unmod' = unmodified). (**C**) As in (**B**), but each reaction comprises ALC1, unmodified nucleosome (20 nM), and the indicated modified histone peptide or PARP1. Modified histone peptide concentration is equal to the corresponding full-length histone concentration (40 nM). The H2BS6/H3S10ADPr$_4$ nucleosome remodeling data is included for direct comparison. (**D**) Western blot analysis of a FLAG immunoprecipitation (IP) wherein ALC1 is FLAG-tagged and its association with nucleosomes is analyzed in the presence and absence of unmodified or automodified PARP1. The corresponding input (5%) was loaded alongside the IP (elution) lanes for comparison. (**E**) ALC1 remodeling rate constants calculated from data in (**B, C**) and *Figure 5—figure supplement 2A*. Rate constants were determined by fitting data to a non-linear regression model for one phase exponential decay. Data in (**B**) and (**C**) are represented as mean ± s.d. (n = 3), while the error bars in (**E**) represent 95% CI. Curves in (**B**) and (**C**) represent data fitting to a non-linear regression model for one-phase exponential decay.

The online version of this article includes the following source data and figure supplement(s) for figure 5:

**Source data 1.** Uncropped TBE gels and western blot from *Figure 5*.

*Figure 5 continued on next page*

*Figure 5 continued*

**Figure supplement 1.** Characterization of ALC1 remodeling activity in presence of various macrodomain ligands.

**Figure supplement 1—source data 1.** Uncropped TBE gels and western blot from *Figure 5—figure supplement 1*.

**Figure supplement 2.** Characterization of ALC1 remodeling activity in presence of automodified PARP1.

**Figure supplement 2—source data 1.** Uncropped western blot from *Figure 5—figure supplement 2*.

to modify histones due to absence of HPF1. Quantitative PARP1 auto-ADPr was observed within 5 min of initiating the reaction as judged by altered PARP1 gel migration in SDS-PAGE/western blot analyses (*Figure 5—figure supplement 1E*). PARP1 was added to the reaction at equimolar concentrations relative to nucleosome substrates 20 nM to closely mimic ADP-ribose concentrations in our modified nucleosome experiments or 100 nM to ensure optimal ALC1 activation. We found that auto-ADP-ribosylated PARP1 leads to an ~12 fold increase in ALC1 remodeling rate constant on unmodified nucleosomes (*Figure 5C*, *Figure 5—figure supplement 2A* and *Supplementary file 6*). Notably, higher PARP1 concentrations were unable to further stimulate ALC1 remodeling activity (*Figure 5—figure supplement 2B*).

In the PARP1 automodification reaction described above, aspartate and glutamate side chains are the primary targets for ADPr as no HPF1 is present. However, in the cellular DNA damage response, it is now well-established that automodification occurs primarily on serine residues (*Bonfiglio et al., 2017*; *Palazzo et al., 2018*). We also observed partial ADPr of ALC1 during this assay (*Figure 5—figure supplement 2C*), which may impact remodeling activity. We therefore performed a PARP1 automodification reaction in the presence of low (5 µM) or high (25 µM) amounts of HPF1 as a separate step prior to the remodeling assay. By employing different HPF1 concentrations, a full-length PARP1 construct with relatively short (PARP1 SerADPr_short) and long (PARP1 SerADPr_long) serine-linked ADP-ribose chains could be generated. These constructs were purified over a heparin column to separate automodified PARP1 from DNA, NAD$^+$ and HPF1, which could otherwise abrogate the nucleosome interaction or induce spurious ADPr of reaction components such as ALC1. The auto-ADPr linkage identity was then validated via hydroxylamine treatment, which specifically cleaves ADPr from aspartate and glutamate side chains. As expected, the ADP-ribose chains conjugated to PARP1 SerADPr_short and PARP1 SerADPr_long are largely resistant to hydroxylamine cleavage (*Figure 5—figure supplement 2D*). We note that a small amount of hydroxylamine-dependent ADP-ribose chain cleavage was observed in the PARP1 SerADPr_long sample. Thus, a small population of aspartate- and/or glutamate-linked ADPr is present in this sample, likely due to the relatively low HPF1:PARP1 ratio that was required to produce long ADP-ribose chains. Immunoprecipitations with FLAG-tagged ALC1 revealed that PARP1 SerADPr_short and PARP1 SerADPr_long are able to induce formation of an ALC1:nucleosome:PARP1 complex (*Figure 5D*). We then titrated each construct into an ALC1 remodeling reaction with unmodified nucleosomes and observed optimal remodeling stimulation at 100 nM of automodified PARP1 (*Figure 5—figure supplement 2E*). Remodeling rate constant calculations show that PARP1 SerADPr_short and PARP1 SerADPr_long stimulate ALC1 activity ~28 fold and ~36 fold, respectively, when compared to activity in the absence of automodified PARP1 (*Figure 5C* and *Supplementary file 6*). We also found automodified PARP1 is unable to further stimulate ALC1 remodeling activity when an ADPr nucleosome substrate is employed (*Figure 5—figure supplement 2F*), which clearly demonstrates that nucleosome ADPr significantly influences ALC1 remodeling even in the presence of automodified PARP1. We stress that while nucleosome serine ADPr is superior to PARP1 auto-ADPr for ALC1 activation in biochemical assays (*Figure 5E*), these data do not allow us to conclude that this is the case in the cellular DNA damage response. However, our work does raise interesting new questions about regulatory mechanisms underlying ALC1 activity (see Discussion).

## ALC1 specificity persists within mixed nucleosome pools

To further probe ALC1 nucleosome substrate selectivity, we designed a method to pool unmodified, mono-, tri-, and tetra-ADP-ribosylated nucleosomes into a single reaction and analyze nucleosome remodeling activity for each unique substrate simultaneously (*Figure 6A*). Similar next-generation sequencing-based approaches have been implemented for rate constant analysis of the ISWI chromatin remodeler family (*Dann et al., 2017*). If ALC1 activity is dependent upon the ADPr status of target nucleosomes, only the tri- and tetra-ADP-ribosylated species should be efficiently remodeled

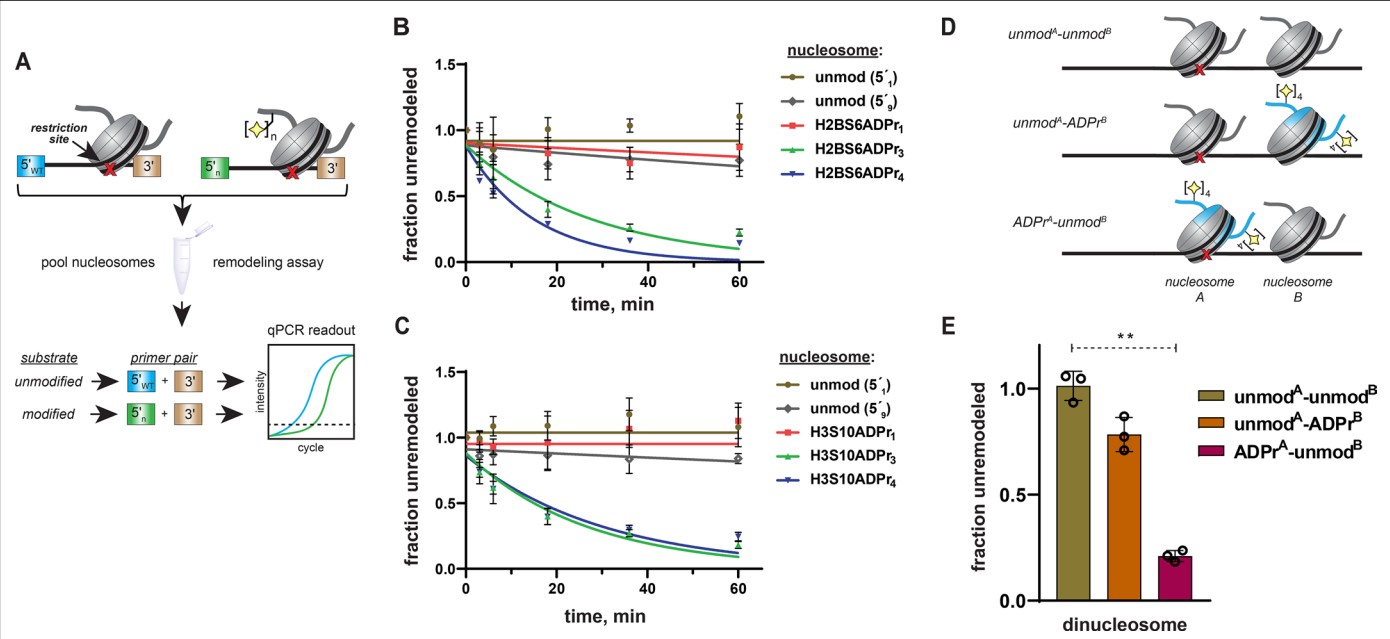

**Figure 6.** Specificity of ALC1 is preserved in heterogenous nucleosome populations and on asymmetrically ADP-ribosylated dinucleosome substrates. (**A**) Schematic depicting the strategy to prepare heterogenous nucleosome substrate pools and determine ALC1 remodeling activity on specific nucleosomes. (**B**) ALC1 nucleosome remodeling assay time-course for each nucleosome in the histone H2B mixed substrate pool. Two unmodified nucleosomes with different 5′ primer sequences (5′$_1$ and 5′$_9$) were included as internal controls. (**C**) As in (**B**), but with the histone H3 substrate pool. (**D**) A diagram depicting the various dinucleosome constructs assembled in this study. Blue shade represents the modified histone H2B, the red cross represents the PstI restriction site and the yellow star represents the ADP-ribose modification. (**E**), Chromatin remodeling assays with ALC1 on the indicated dinucleosome substrates. ** indicates p-value < 0.001, obtained using an unpaired Student's t-test with Welch's correction. Data in (**B**), (**C**), and (**E**) are represented as mean ± s.d. (n = 3). Curves in (**B**) and (**C**) represent data fitting to a non-linear regression model for one-phase exponential decay.

The online version of this article includes the following source data and figure supplement(s) for figure 6:

**Figure supplement 1.** ALC1 remodeling analysis on heterogenous nucleosome populations.

**Figure supplement 1—source data 1.** Uncropped TBE gels from *Figure 6—figure supplement 1*.

in this substrate competition-based platform. We again turned to the REA assay but appended a unique 5′ 15-base pair primer binding site to each 601 DNA template. Importantly, we designed priming sequences with similar primer binding efficiencies and found that DNA sequence alterations in this region of the template do not affect remodeling rates (*Figure 6—figure supplement 1A*). In this assay, restriction enzyme-dependent destruction of a given 601 template amplicon is quantified by qPCR to monitor remodeling activity. Thus, unique primer pairs corresponding to each nucleosome can be employed to determine substrate-specific chromatin remodeling rate constants in heterogenous substrate reactions.

We assembled a nucleosome pool comprising equimolar concentrations of H2BS6ADPr$_1$, H2BS6ADPr$_3$, H2BS6ADPr$_4$, and two unmodified nucleosome controls. An additional unmodified nucleosome without the PstI restriction site and a free DNA template with the PstI site were also included as negative and positive digestion controls, respectively. The heterogeneous nucleosome substrate pool was employed in ALC1 remodeling reactions as described above, and DNA from various time points was isolated and analyzed via qPCR. We found that relative remodeling rate constants were consistent with those observed in our single substrate, densitometry-based assays (*Figure 6B* and *Supplementary file 7*). ALC1 again exhibits modest preference for the H2BS6ADPr$_4$ nucleosome relative to the H2BS6ADPr$_3$ nucleosome. Remodeling was very slow for the unmodified and H2BS6ADPr$_1$ nucleosomes and corresponding rate constants could not be determined in this assay platform. Substrate preferences were also maintained within a similar H3S10-modified substrate pool (*Figure 6C* and *Supplementary file 7*). Notably, H3 nucleosomes were analyzed as a separate population because they require a higher ALC1 concentration to achieve optimal dynamic range in the qPCR-based assay.

As a complementary approach, three asymmetrically ADP-ribosylated dinucleosome constructs were prepared wherein: (i) both nucleosome A and B are unmodified (unmod$^A$-unmod$^B$), (ii) nucleosome A is unmodified and nucleosome B comprises H2BS6ADPr$_4$ histones (unmod$^A$-ADPr$^B$), or (iii) nucleosome A comprises H2BS6ADPr$_4$ histones and nucleosome B is unmodified (ADPr$^A$-unmod$^B$) (*Figure 6D*, *Figure 6—figure supplement 1B*). In all dinucleosome constructs, nucleosome A bears a 45 base pair DNA overhang and is separated from nucleosome B by a 15 base pair DNA linker to allow for remodeling to occur. By removing the PstI site from nucleosome B, we were able to specifically monitor ALC1 remodeling activity on nucleosome A using our REA assay. Following incubation with ALC1, robust nucleosome A remodeling activity was observed (~79% remodeled in 60 min) with the ADPr$^A$-unmod$^B$ construct as expected (*Figure 6E*, *Figure 6—figure supplement 1C*). In contrast, relatively modest nucleosome A remodeling activity was observed (~21% remodeled in 60 min) with the unmod$^A$-ADPr$^B$ construct. While these single time-point analyses cannot be directly compared to the mononucleosome remodeling rate constant analyses, the dinucleosome remodeling activity results are consistent with the automodified PARP1 experiments; PARP1 auto-ADPr and adjacent nucleosome ADPr both tether poly-ADP-ribose in close proximity to unmodified nucleosomes but neither is able to fully stimulate ALC1. These experiments demonstrate that ALC1 preferentially remodels binding-competent nucleosome substrates and target disengagement triggers rapid transition back to an inactive conformation. This mechanism likely minimizes the potential for freely diffusing, activated ALC1 to be present in the nuclear milieu.

## Nucleosome serine ADPr triggers ALC1-dependent chromatin remodeling in nuclear extracts

It is possible that a poly-anionic chain fused to H2BS6 or H3S10 destabilizes the histone octamer:DNA complex and thereby non-specifically sensitizes nucleosomes to ATP-dependent chromatin remodelers. To examine this concept, we isolated the ATP-dependent chromatin remodeler CHD4 for activity analysis. CHD4 lacks a macrodomain while its ATPase domain shares a high degree of sequence similarity (63%) with ALC1 (*Figure 7—figure supplement 1A*), suggesting that the two enzymes may catalyze DNA translocation through similar mechanistic principles. The REA assay revealed that CHD4 remodels unmodified nucleosomes with a rate constant of ~0.01 min$^{-1}$ and this activity is not appreciably affected by the nucleosome ADPr status (*Figure 7A and B*, *Figure 7—figure supplement 1B*, and *Supplementary file 6*). These data suggest that nucleosome serine ADPr does not simply decrease the energy barrier to DNA translocation but rather serves to specifically stimulate ALC1-dependent chromatin remodeling.

To investigate the ability of nucleosome serine ADPr to stimulate ALC1 activity in a more physiological context, mammalian cell nuclear extracts were employed as a source of remodeling activity with the ADP-ribosylated nucleosome substrates. Nuclear extracts were prepared from wild-type or ALC1 knock-out (KO) HEK293T cells and the presence of various endogenous chromatin remodelers was confirmed (*Figure 7C*). Each extract was then incubated with unmodified, H2BS6ADPr$_1$, H2BS6ADPr$_4$, or H2BS6/H3S10ADPr$_4$ nucleosomes for 60 min and total remodeling activity was determined via the REA assay (*Figure 7D*). The wild-type extract exhibited a ~ 3 fold increase in total remodeling activity towards the H2BS6ADPr$_4$ and H2BS6/H3S10ADPr$_4$ nucleosomes when compared to their unmodified counterpart (*Figure 7E*). Contrastingly, there was no appreciable increase in activity towards the H2BS6ADPr$_1$ nucleosome. Strikingly, the ALC1-KO nuclear extract exhibited no remodeling substrate preference regardless of nucleosomes ADPr status (*Figure 7E*, and *Figure 7—figure supplement 1C*). We note that no accumulation of additional ADPr events was detected in these extracts throughout the duration of the assay (*Figure 7—figure supplement 1D*). Importantly, partial hydrolysis of nucleosome poly-ADPr was detected following the assay, likely due to the presence of ARH3, PARG, and/or other glycohydrolases. Therefore, the remodeler substrate preference in wild-type extracts may actually be even greater than what we observed in this assay. We also observed an increase in overall remodeling activity towards all substrates in the ALC1-KO extract (for raw data, see *Figure 7—figure supplement 1E* and Supplementary Dataset), which may be a consequence of subtle lysate preparation variables or represent a cellular mechanism to compensate for loss of ALC1.

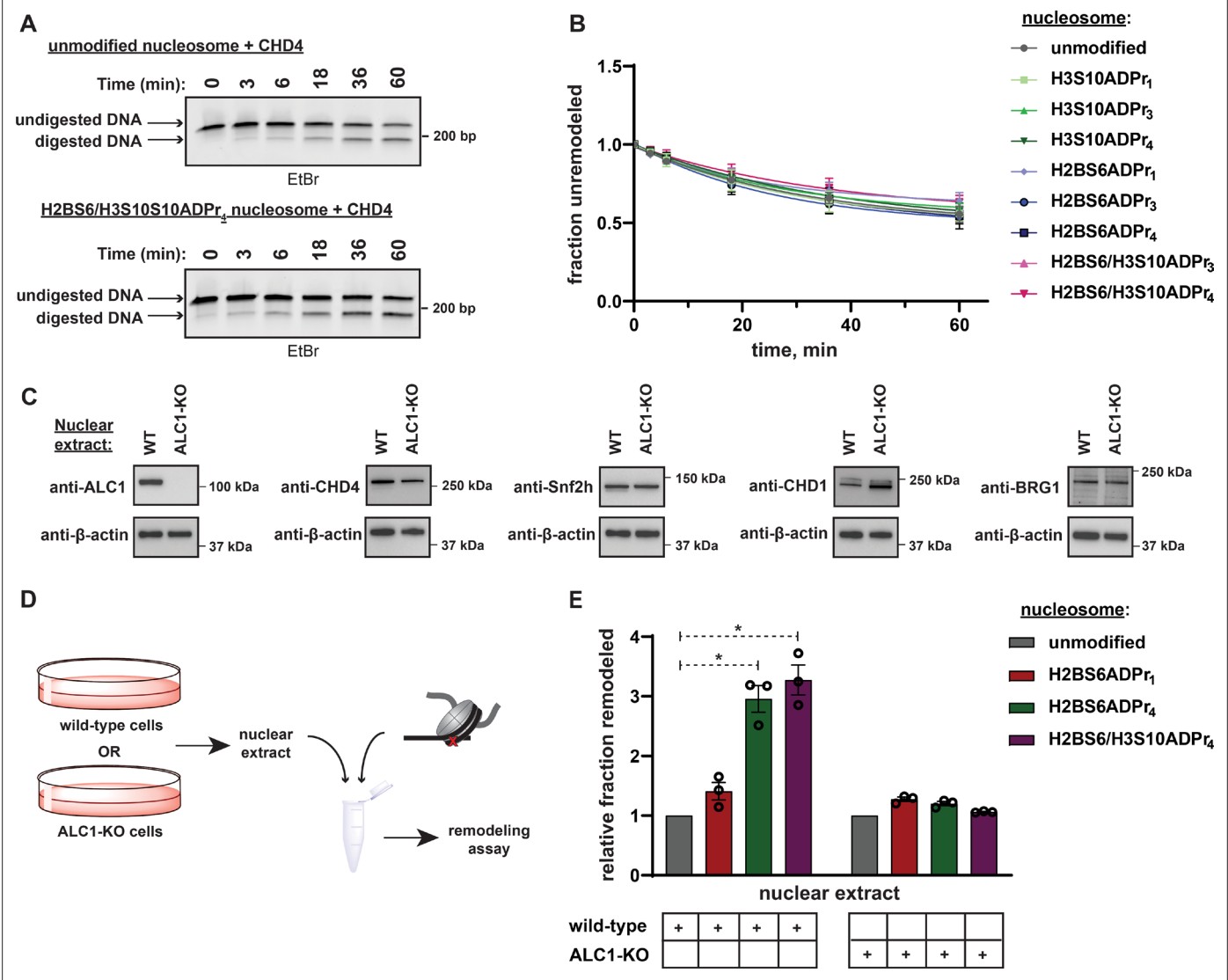

**Figure 7.** Nucleosome serine ADPr stimulates ALC1-dependent chromatin remodeling activity in nuclear extracts. (**A**) Representative TBE gel analysis from a REA assay corresponding to recombinant CHD4 chromatin remodeling activity on unmodified or H2BS6/H3S10ADPr$_4$ nucleosomes. (**B**) CHD4 nucleosome remodeling assay time-course wherein each reaction comprises CHD4 and the indicated nucleosome substrate. Data are represented as mean ± s.d. (n = 3). Curves represent fit of data into a non-linear regression model for one-phase exponential decay. (**C**) Western blot analysis demonstrating the presence of various chromatin remodelers in the wild-type or ALC1 knock-out (KO) HEK293T nuclear extracts. (**D**) Schematic depicting the strategy to analyze chromatin remodeling activity in wild-type or ALC1-KO HEK293T nuclear extracts. (**E**) Nuclear extract nucleosome remodeling activity assay wherein each reaction comprises the indicated nucleosome substrate and wild-type or ALC1-KO HEK293T cell nuclear extracts. Total remodeling for each ADP-ribosylated nucleosome substrate relative to the unmodified nucleosome substrate in the respective nuclear extract is shown. Data are represented as mean ± s.e.m. (n = 3). * indicates p-value < 0.02, obtained using an unpaired Student's t-test with Welch's correction.

The online version of this article includes the following source data and figure supplement(s) for figure 7:

**Source data 1.** Uncropped TBE gels and western blots from Figure 7.

**Figure supplement 1.** H3S10ADPr and H2BS6ADPr specifically sensitize nucleosomes to chromatin remodeling by ALC1.

**Figure supplement 1—source data 1.** Uncropped TBE gels and western blots from Figure 7-figure supplement 1.

## Discussion

Chemical and topological complexities have stymied previous efforts to synthesize poly-ADP-ribosylated proteins. Our investigation of HPF1-dependent and -independent PARP1 activities in peptide serine ADPr reactions guided the development of a multistep chemoenzymatic approach that

is broadly applicable for the preparation of poly-ADP-ribosylated peptides and fully compatible with protein ligation technologies. Through the use of chemically homogenous, ADP-ribosylated histones we were able to define a biochemical role for nucleosome serine ADPr and explore long-standing questions related to DNA damage-induced chromatin remodeling.

Multiple recent reports show that the PARP1/2:HPF1 complex catalyzes the formation of relatively short poly-ADP-ribose chains (*Bilokapic et al., 2020*; *Bonfiglio et al., 2020*; *Gibbs-Seymour et al., 2016*). Our study is unique in that we prepare unmodified and mono-ADP-ribosylated peptide substrates and use HPLC-MS to analyze PARP1 reaction products in the absence and presence of HPF1. This approach demonstrated that HPF1 simultaneously stimulates mono-ADPr activity and blocks ADP-ribose chain elongation on *trans*-peptide substrates. The biological two-step ADPr process explains why mono- and poly-ADPr reactions must be separated for scalable preparation of poly-ADP-ribosylated peptides – the high concentrations of HPF1 required for maximal conversion of unmodified peptides to the mono-ADP-ribosylated species simultaneously inhibits formation of the poly-ADP-ribosylated species (*Supplementary file 8*).

Our data support PARP2:HPF1 structural implications that mono- and poly-ADPr are mutually exclusive activities (*Suskiewicz et al., 2020*) and demonstrate that structural dynamics are insufficient to accommodate both catalytic mechanisms. More contemporaneous studies show that HPF1 rapidly associates and dissociates from PARP1 to regulate serine-specific mono- and poly-ADPr (*Langelier et al., 2021*; *Prokhorova et al., 2021*). This mechanism, together with ARH3- and PARG-dependent ADP-ribosylhydrolase activity (*Bonfiglio et al., 2020*), are believed to be critical regulators of ADPr at DNA damage sites. Notably, HPF1 and PARP1/2 undergo DNA damage-induced ADPr, which may serve to disrupt the complex and regulate ADP-ribose polymerization. This would explain why we and others observe elongation activity in recombinant assays that include relatively high molar ratios of HPF1 to PARP1; ADPr on one or both complex components decreases the effective PARP1:HPF1 concentration as the reaction progresses. It is also likely that high HPF1 concentration is necessary to ensure rapid re-association with the PARP1:DNA complex based on the 'rapid association/dissociation' model described above. Alternatively, we note that the cellular molar ratio of PARP1 to HPF1 (20:1) (*Hein et al., 2015*) is favorable for a mechanism wherein free PARP1 displaces the PARP1/2:HPF1 complex once mono-ADP-ribose seeding has occurred.

In chromatin remodeling experiments, ALC1 exhibits modest preference for the H2BS6 site and tetra-ADPr despite the observation that all H2BS6ADPr$_{3,4}$ and H3S10ADPr$_{3,4}$ peptides engage the ALC1 macrodomain with equal affinity. It is therefore likely that each histone modification site requires an ideal ADP-ribose chain length that allows the ATPase domain to progress through the DNA translocation cycle while the macrodomain:histone tail interaction is maintained. There are several factors that may explain why nucleosome serine ADPr more efficient than auto-ADP-ribosylated PARP1 for ALC1 activation in our assays: (i) robust ALC1 activation by auto-ADP-ribosylated PARP1 may require a specific modification site and ADP-ribose chain length that is only partially represented on our automodified PARP1 constructs, (ii) the PARP1:nucleosome interaction, while necessary for ALC1 recruitment and activation, may also sterically abrogate DNA translocation activity, and (iii) a direct interaction between ALC1 and ADP-ribosylated nucleosomes may be stronger than the ternary complex that is mediated by automodified PARP1, as evidenced from nucleosome pull-down efficiency in *Figure 5D*.

Critical distinctions unique to nucleosome ADPr over other ADP-ribosylated proteins are: (i) the nucleosome-incorporated histones cannot diffuse away from the DNA damage site, and (ii) the stimulatory ADP-ribose chain is not tethered to a DNA-bound protein that may sterically hinder remodeling by ALC1. Therefore, nucleosome ADPr offers a fail-safe mechanism to ensure that robust ALC1-dependent remodeling can persist in the event that automodified PARP1 dissociates from the damage site prior to ALC1 activation. It is also interesting that ALC1 exhibits prolonged retention at DNA damage sites in HPF1-null cells where serine ADPr does not occur (*Gibbs-Seymour et al., 2016*). This is consistent with our observation that aspartate/glutamate-automodified PARP1 is the least potent activator in biochemical assays. It is plausible that serine ADPr, be it tethered to the nucleosome or PARP1, is critical for ALC1 remodeling activity at DNA damage sites in cells. While our technology has allowed us to separate and characterize ALC1 activation by ADPr on nucleosomes or PARP1 in a reconstituted environment, new approaches

will be required to specifically control these parameters and analyze their contributions to ALC1-dependent remodeling at damage sites in cells.

Analyses of remodeling activity in biochemical assays and mammalian cell nuclear extracts show that nucleosome serine ADPr is sufficient to specifically activate ALC1 in the absence of auto-ADP-ribosylated PARP1. We surmise that other PARP1/2-dependent chromatin remodelers are recruited to damage sites via alternative ADPr modification sites or chain lengths, as has been reported for SMARCA5 (*Smeenk et al., 2013*). Additionally, these remodelers may not directly interact with ADP-ribose but are rather recruited by alternative PARP1/2-dependent activities, a phenomenon that has been demonstrated for CHD4 (*Smith et al., 2018*). Thus, our study supports the 'PAR code' hypothesis (*Aberle et al., 2020*) as it pertains to chromatin structure at DNA lesions wherein different ADPr sites and chain lengths may orchestrate spaciotemporal control over unique remodeler activities. Notably, dozens of proteins reportedly exhibit PARP1/2-dependent recruitment to DNA damage sites and have been annotated as ADP-ribose 'readers' (*Ray Chaudhuri and Nussenzweig, 2017*; *Teloni and Altmeyer, 2016*). With full-length ADP-ribosylated proteins, ADPr-mediated activities can now be reconstituted for rigorous biochemical, biophysical, and structural analysis.

Beyond protein recruitment, it will now be possible to explore the direct biophysical effects that H2B and H3 ADPr have on poly-nucleosome array structure and compaction. Our modular chemoenzymatic approach can also be expanded to other PARP1/2:HPF1 substrate proteins, wherein one would expect to find ADPr exerts its effects via unique regulatory mechanisms that are tailored to the target protein. As demonstrated here, critical aspects of PARP biological function can be unveiled by reconstituting ADP-ribosylated proteins and related signaling pathway components. A greater understanding of PARP-regulated biological processes, including ALC1 activation, may lead to identification of new biomarkers and therapeutic strategies for PARP inhibitor-sensitive diseases.

## Technological limitations

The method described here is currently limited to installation of ADP-ribose units ~ 4–5 linear units in length. Exceedingly large-scale reactions would be required to prepare peptides modified with longer ADP-ribose chains. Therefore, this method is ideal to study signal transduction events that are mediated by relatively short ADP-ribose chains. Our strategy also requires that a peptide of interest be a substrate for the PARP1:HPF1 complex. Alternative ADP-ribosyltransferases will be required to install ADPr on proteins that are not endogenous targets of this complex using the chemoenzymatic approach presented here. While all histones reported here were prepared via an N-terminal ligation (i.e. ADPr on the amino-terminal protein fragment), C-terminal and sequential ligations are also compatible with our method. As proof of feasibility, an H3S28ADPr$_4$ peptide construct (amino acids 21–34) was prepared with an N-terminal cysteine and an oxidized C-terminal bis(2-sulfanylethyl)amido (SEA) group. This peptide was then ligated to the H3S10ADPr$_4$ thioester peptide construct (amino acids 1–20) and the SEA group reduced to generate an H3S10/S28ADPr$_4$ product (amino acids 1–34) that can be employed in downstream ligations if desired (*Supplementary file 9*). Lastly, our method is still susceptible to restraints that exist throughout the field of protein chemistry. This means that alternative protein ligation technologies will be required to install modification onto full-length proteins that are not amenable to protein folding.

## Quantification and statistical analysis

Details related to replicates, error, and curve fitting are described in respective figure legends. In *Figure 6E*, the difference of means of fraction unremodeled between the unmod[A]-unmod[B] and ADPr[A]-unmod[B] dinucleosomes was statistically significant with a p-value of 0.0008 (95% CI ranging from –0.9521 to –0.6545), obtained using an unpaired Student's t-test with Welch's correction. In the remodeling experiment with wild-type nuclear extracts (*Figure 7E*), p-value for the difference of mean between the unmodified and H2BS6/H3S10ADPr$_4$ nucleosome data was 0.0120 (95% CI ranging from

1.195 to 3.355) and that for the difference of mean between the unmodified and H2BS6ADPr$_4$ nucleosome data was 0.0127 (95% CI ranging from 1.000 to 2.918).

## Contact for reagent and resource sharing

Further information and requests for reagents may be directed to and will be fulfilled by the Lead Contact, Dr. Glen Liszczak (glen.liszczak@utsouthwestern.edu).

# Materials and methods

**Key resources table**

| Reagent type (species) or resource | Designation | Source or reference | Identifiers | Additional information |
|---|---|---|---|---|
| Recombinant DNA reagent | PARP1 (pACEBac1) (plasmid) | This study | | See Materials and methods |
| Recombinant DNA reagent | PARP2 (pACEBac1) (plasmid) | Addgene | 111,574 | |
| Recombinant DNA reagent | HPF1 (pET30) (plasmid) | This study | | See Materials and methods |
| Recombinant DNA reagent | HPF1D283A (pET30) (plasmid) | This study | | See Materials and methods |
| Recombinant DNA reagent | ARH3 (pET30) (plasmid) | Addgene | 111,578 | |
| Recombinant DNA reagent | PARG (pET30) (plasmid) | This study | | See Materials and methods |
| Recombinant DNA reagent | ALC1 macrodomain (pET30) (plasmid) | This study | | See Materials and methods |
| Recombinant DNA reagent | ALC1 ATPase domain (pET30) (plasmid) | This study | | See Materials and methods |
| Recombinant DNA reagent | ALC1 (pACEBac1) (plasmid) | This study | | See Materials and methods |
| Recombinant DNA reagent | CHD4 (pACEBac1) (plasmid) | This study | | See Materials and methods |
| Recombinant DNA reagent | Histone H2A (pET30) (plasmid) | This study | | See Materials and methods |
| Recombinant DNA reagent | Histone H2B (pET30) (plasmid) | This study | | See Materials and methods |
| Recombinant DNA reagent | Histone H2B truncated (pET30) (plasmid) | This study | | See Materials and methods |
| Recombinant DNA reagent | Histone H3 (pET30) (plasmid) | This study | | See Materials and methods |
| Recombinant DNA reagent | Histone H3 truncated (pET30) (plasmid) | This study | | See Materials and methods |
| Recombinant DNA reagent | Histone H4 (pET30) (plasmid) | This study | | See Materials and methods |
| Sequence-based reagent | DNA oligonucleotides | This study | | See *Supplementary file 10* |
| Antibody | Antibodies used for western blot | | | See *Supplementary file 11* |
| Strain, strain background (*Escherichia coli*) | Mach1 (*Escherichia coli*) | ThermoFisher | C862003 | |

*Continued on next page*

*Continued*

| Reagent type (species) or resource | Designation | Source or reference | Identifiers | Additional information |
|---|---|---|---|---|
| Strain, strain background (*Escherichia coli*) | DH10Bac | ThermoFisher | 10361012 | |
| Strain, strain background (*Escherichia coli*) | Rosetta 2 | Sigma Aldrich | 714,023 | |
| Cell line (*Spodoptera frugiperda*) | Sf9 | ThermoFisher | 11496015 | |
| Cell line (*Homo sapiens*) | HEK293T | ATCC | CRL-3216 | |
| Peptide, recombinant protein | FLAG peptide | GenScript | RP10586-1 | Sequence: DYKDDDDK |
| Commercial assay or kit | Gibson Assembly Master Mix | New England Biolabs (NEB) | E2611S | |
| Commercial assay or kit | iTaq Universal SYBR Green Supermix | BioRad | 1725121 | |
| Commercial assay or kit | Pierce BCA Protein Assay Kit | ThermoFisher | 23,227 | |
| Chemical compound, drug | β-Nicotinamide adenine dinucleotide hydrate | Sigma Aldrich | N0632 | |
| Chemical compound, drug | Adenosine triphosphate | Sigma Aldrich | A26209-1G | |
| Chemical compound, drug | HisPur Ni-NTA Resin | ThermoFisher | 88,223 | |
| Chemical compound, drug | Anti-FLAG M2 Magnetic beads | Millipore Sigma | M8823-5ML | |
| Chemical compound, drug | Phusion High-Fidelity DNA Polymerase | NEB | M0530L | |
| Chemical compound, drug | PstI enzyme | NEB | R0140M | |
| Chemical compound, drug | DraIII enzyme | NEB | R3510S | |
| Chemical compound, drug | T4 Polynucleotide Kinase | NEB | M0201S | |
| Chemical compound, drug | T4 DNA Ligase | NEB | M0202L | |
| Chemical compound, drug | Cellfectin II Reagent | ThermoFisher | 10362100 | |
| Chemical compound, drug | Sf-900 II SFM Media | ThermoFisher | 10902096 | |
| Chemical compound, drug | Fetal Bovine Serum | ThermoFisher | 10438026 | |
| Chemical compound, drug | DMEM, high glucose, pyruvate | ThermoFisher | 11995065 | |
| Chemical compound, drug | Penicillin-Streptomycin | ThermoFisher | 15140122 | |

*Continued*

| Reagent type (species) or resource | Designation | Source or reference | Identifiers | Additional information |
|---|---|---|---|---|
| Chemical compound, drug | Lipofectamine 2000 | ThermoFisher | 11-668-019 | |
| Chemical compound, drug | Puromycin dihydrochloride | ThermoFisher | A1113803 | |
| Chemical compound, drug | Hydroxylamine hydrochloride | Sigma Aldrich | 379,921 | |
| Chemical compound, drug | N,N-dimethylformamide | Oakwood Chemical | 046776 | |
| Chemical compound, drug | Dichloromethane | Oakwood Chemical | 035912 | |
| Chemical compound, drug | Trifluoroacetic acid | Oakwood Chemical | 102,164 | |
| Chemical compound, drug | Acetonitrile | Oakwood Chemical | 099891 | |
| Chemical compound, drug | Formic Acid, LC/MS Grade | Thermo Fisher | A117-50 | |
| Chemical compound, drug | N,N'-Diisopropylcarbodiimide | Oakwood Chemical | M02889 | |
| Chemical compound, drug | Ethyl cyanohydroxyiminoacetate | Oakwood Chemical | 043278 | |
| Chemical compound, drug | N,N-Diisopropylethylamine | Sigma Aldrich | 496,219 | |
| Chemical compound, drug | Thionyl chloride | Sigma Aldrich | 230,464 | |
| Chemical compound, drug | Triisopropylsilane | Sigma Aldrich | 233,781 | |
| Chemical compound, drug | Piperidine | Sigma Aldrich | 104,094 | |
| Chemical compound, drug | PyAOP | Oakwood Chemical | 024898 | |
| Chemical compound, drug | Econo-Pac Chromatography Columns | Bio-Rad | 7321011 | |
| Chemical compound, drug | Sodium nitrite | Sigma Aldrich | 237,213 | |
| Chemical compound, drug | TCEP | GoldBio | TCEP50 | |
| Chemical compound, drug | Sodium 2-mercaptoethanesulfonate | Sigma Aldrich | 63,705 | |
| Chemical compound, drug | 2,2,2-Trifluoroethanethiol | Sigma Aldrich | 374008–1 G | |
| Chemical compound, drug | Trityl-OH ChemMatrix | Biotage | 7-420-1310 | |
| Chemical compound, drug | Rink-Amide-ChemMatrix | Biotage | 7-600-1310 | |
| Chemical compound, drug | Fmoc-Ala-OH | Oakwood Chemical | M03347 | |
| Chemical compound, drug | Fmoc-Arg(pbf)-OH | Oakwood Chemical | M03398 | |

*Continued on next page*

*Continued*

| Reagent type (species) or resource | Designation | Source or reference | Identifiers | Additional information |
|---|---|---|---|---|
| Chemical compound, drug | Fmoc-Thr(tBu)-OH | Oakwood Chemical | M03389 | |
| Chemical compound, drug | Fmoc-Lys(Boc)-OH | Oakwood Chemical | M03419 | |
| Chemical compound, drug | Fmoc-Gln(trt)-OH | Combi-Blocks | QB-0626 | |
| Chemical compound, drug | Fmoc-Ser(tBu)-OH | Combi-Blocks | SS-0149 | |
| Chemical compound, drug | Fmoc-Gly-OH | Oakwood Chemical | M03361 | |
| Chemical compound, drug | Fmoc-Leu-OH | Oakwood Chemical | M03365 | |
| Chemical compound, drug | Fmoc-Pro-OH | Oakwood Chemical | M03372 | |
| Chemical compound, drug | Fmoc-Cys(Trt)-OH | Oakwood Chemical | M03395 | |
| Chemical compound, drug | Boc-L-thiazolidine-4-carboxylic acid | Combi-Blocks | SS-9673 | |
| Chemical compound, drug | 5 (6)-Carboxyfluorescein | Sigma Aldrich | 21877–1 G-F | |
| Chemical compound, drug | SEA-PS Resin | Iris Biotech | 8551520001 | |
| Software, algorithm | Prism | GraphPad | | |
| Software, algorithm | Fiji Image J | Open source | | |
| Software, algorithm | Microsoft Office | Microsoft | | |
| Software, algorithm | Adobe Creative Cloud | Adobe | | |
| Software, algorithm | SnapGene | GSL Biotech LLC | | |
| Software, algorithm | ChemDraw | PerkinElmer | | |
| Software, algorithm | BLAST | NCBI | | |
| Other | Liberty Blue Automated Peptide Synthesizer | CEM | 925,600 | |
| Other | Agilent 1,260 Infinity II with quaternary pump and variable wavelength detector | Agilent | G7111B/G7114A | |
| Other | Agilent 1,260 Infinity II with preparatory pump and variable wavelength detector | Agilent | G7161A/G7114A | |
| Other | Agilent LC/MSD | Agilent | G6125BA | |
| Other | X500B QTOF | Sciex | X500B QTOF | |
| Other | 300 SB-C18, 4.6 × 100 mm, 3.5 um | Agilent | 861973–902 | |

*Continued on next page*

*Continued*

| Reagent type (species) or resource | Designation | Source or reference | Identifiers | Additional information |
|---|---|---|---|---|
| Other | XBridge Peptide C18 column, 5 um, 4.6 mm x 150 mm | Waters | 186003624 | |
| Other | XBridge Peptide C18 column, 5 um, 10 mm x 250 mm | Waters | 186008193 | |
| Other | XBridge Peptide C18 column, 10 um, 19 mm x 250 mm | Waters | 186003673 | |
| Other | AKTA pure 25 L | GE Healthcare | 29018224 | |
| Other | AKTA start | GE Healthcare | 29237234 | |
| Other | HiLoad 16/60 Superdex 200 pg | GE Healthcare | 28989335 | |
| Other | Superdex 200 Increase 10/300 GL | GE Healthcare | 28990944 | |
| Other | HiTrap Heparin HP | GE Healthcare | 17040701 | |
| Other | 5% Criterion TBE Polyacrylamide Gel | Bio-Rad | 3450048 | |
| Other | 4%–12% Criterion XT Bis-Tris Protein Gel, | Bio-Rad | 3450124 | |
| Other | Cytation 5 Imaging reader | Biotek | 17103120 | |
| Other | Green FP filter set: EX 485/20 \| EM 528/20 \| DM 510, polarizers | Biotek | 8040561 | |
| Other | Chemidoc MP Imaging System | Bio-Rad | 734BR2154 | |
| Other | CFX384 Real Time System C1000 Touch Thermal cycler | Bio-Rad | 786BR02877 | |

## Molecular cloning, protein expression, and protein purification

### General protocols

All PCR amplification steps described here were performed using the Phusion High-Fidelity DNA Polymerase (NEB) according to the manufacturer's protocols. All DNA oligonucleotides were synthesized by Sigma-Aldrich (Milwaukee, WI) or Integrated DNA Technologies (Coralville, IA). All plasmids used in this study were sequence verified by GENEWIZ (South Plainfield, NJ) or EurofinsGenomics (Louisville, KY). All cloning was carried out using Mach1 *E. coli* cells (ThermoFisher) and protein expression in *E. coli* was carried out in Rosetta2 cells (Sigma-Aldrich).

### PARP1/PARP2 expression and purification

The full-length PARP1 gene was purchased from GE Healthcare and subcloned into a pACEBac1 plasmid bearing an N-terminal 6xHis-tag via a Gibson Assembly (NEB). The PARP2 expression plasmid (C-terminal FLAG-6xHis-tag) is available on Addgene (plasmid #: 111574). PARP1 and PARP2 proteins were produced in Sf9 cells (ThermoFisher) using a baculovirus expression system. Corresponding plasmids were transformed into DH10Bac cells (ThermoFisher) and bacmids were isolated via manufacturer's protocols (ThermoFisher). All subsequent Sf9 cell and baculovirus manipulations were performed in a sterile biosafety cabinet. Cellfectin II (ThermoFisher) was employed to transfect 10 µg of bacmid into $1 \times 10^6$ attached Sf9 cells following manufacturer's protocols (ThermoFisher). P1 virus was harvested 3 days post-transfection. 1 mL of P1 virus was then used to infect 20 mL of Sf9 cells grown in suspension at $1.5 \times 10^6$ cells per mL, which were maintained in a dark orbital shaker at 27 °C. Cells were centrifuged and supernatant (P2 virus) was collected once cell viability dropped to 50%, as measured by trypan blue staining. P3 virus was generated by infecting 50 mL of Sf9 cells at $1.5 \times 10^6$ cells per mL with 0.5 mL of P2 virus. P3 virus was harvested once cells reached 50% viability. Protein production was achieved by treating 2 L of Sf9 cells at $2.0 \times 10^6$ cells per mL with 20 mL of P3 virus for 48 h.

For PARP1, cells were harvested by centrifugation and disrupted via sonication in a lysis buffer containing 50 mM Tris, pH 7.5, 1 M NaCl, 1 mM MgCl$_2$, 5 mM beta-mercaptoethanol (β-ME), and protease inhibitor cocktail (Roche). Soluble lysate was isolated via centrifugation at 100,000 RCF for 60 minutes at 4 °C. The target protein was captured on Ni-NTA resin that was pre-equilibrated in lysis buffer. Following 1 h batch binding, resin was washed with 50 column volumes (CV) of lysis buffer supplemented with 25 mM imidazole and eluted in a buffer containing 50 mM Tris, pH 7.0, 100 mM NaCl, 1.5 mM MgCl$_2$, and 5 mM β-ME. Target protein was then loaded onto a HiTrap Heparin (GE Healthcare) column pre-equilibrated in a low salt buffer (50 mM Tris, pH 7.0, 150 mM NaCl, 1 mM EDTA, 1 mM TCEP) and elution was achieved via an isocratic salt gradient to a high salt buffer (50 mM Tris, pH 7.0, 1 M NaCl, 1 mM EDTA, 1 mM TCEP). Fractions containing the target protein were concentrated to 2 mL using an Amicon Ultra Centrifugal filter (Millipore; 30 kDa molecular weight cut-off [MWCO]) and injected into a gel filtration column (HiLoad 16/60 Superdex 200; GE Healthcare) that had been pre-equilibrated with a buffer containing 50 mM Tris, pH 7.5, 150 mM NaCl, 10% glycerol, and 1 mM TCEP. Pure fractions (as judged by SDS-PAGE) were pooled and concentrated to 100 µM, flash frozen in single-use aliquots, and stored at –80 °C.

For PARP2, cells were harvested by centrifugation and disrupted via sonication in a lysis buffer containing 20 mM Tris, pH 7.9, 500 mM NaCl, 4 mM MgCl$_2$, 0.4 mM EDTA, 20% glycerol, 2 mM DTT, 0.4 mM PMSF, and protease inhibitor cocktail (Roche). Soluble lysate was isolated via centrifugation at 100,000 RCF for 60 min at 4 °C. The supernatant was carefully removed without disturbing the top layer and an equal volume of dilution buffer containing 20 mM Tris, pH 7.9, 10% glycerol, 0.02% NP-40 and protease inhibitor cocktail (Roche) was added to it. The target protein was captured on anti-FLAG M2 magnetic resin that was pre-equilibrated with dilution buffer. Following a 60 min batch binding, resin was washed with 50 CV of wash buffer containing 20 mM Tris, pH 7.9, 150 mM NaCl, 2 mM MgCl$_2$, 0.2 mM EDTA, 15% glycerol, 0.01% NP-40, 0.2 mM PMSF, 1 mM DTT and protease inhibitor cocktail (Roche), and eluted in the wash buffer supplemented with FLAG peptide at a concentration of 0.25 mg/mL. Pure protein was concentrated using an Amicon Ultra Centrifugal filter (Millipore; 30 kDa MWCO) to around 55 µM, as determined by BSA standards in SDS-PAGE, flash frozen in single-use aliquots, and stored at –80 °C.

## HPF1 (and HPF1D283A mutant)

A pET30 plasmid harboring the 6xHis-SUMO-FLAG-HPF1 protein (addgene plasmid #: 111577), encoding amino acids 27–346, was transformed into Rosetta2 (DE3) cells and inoculated into 6 L of Luria Broth (Miller). Cells were grown in a shaker at 37 °C up to an OD$_{600}$ of 0.6 and protein expression was induced with 0.5 mM IPTG at 18 °C for 16 h. Cells were harvested by centrifugation and disrupted via sonication in a lysis buffer containing 50 mM Tris, pH 7.5, 500 mM NaCl, 5 mM β-ME and 1 mM PMSF. Soluble lysate was isolated via centrifugation at 40,000 RCF for 40 min at 4 °C. Target protein was captured on Ni-NTA resin that was pre-equilibrated in lysis buffer. Following 1 h batch binding at 4 °C, resin was washed with 50 CV of lysis buffer supplemented with 25 mM imidazole and protein was eluted in lysis buffer supplemented with 300 mM imidazole. The elution was dialyzed into a buffer containing 50 mM Tris, pH 7.5, 200 mM NaCl, and 5 mM TCEP for 16 h at 4 °C in the presence of the Ulp1 protease to cleave the SUMO tag. The dialysate was then incubated with Ni-NTA resin pre-washed with the dialysis buffer for 1 h at 4 °C to capture the cleaved SUMO tag and the Ulp1, and the flow-through containing the target protein was collected. The flow-through was concentrated to 2 mL using an Amicon Ultra Centrifugal filter (Millipore; 30 kDa MWCO) and injected into a gel filtration column (HiLoad 16/60 Superdex 200) that had been pre-equilibrated with a buffer containing 50 mM Tris, pH 7.5, 200 mM NaCl, 10% glycerol, and 2 mM TCEP. Pure fractions (as judged by SDS-PAGE) were concentrated to around 600 µM, flash frozen in single-use aliquots, and stored at –80 °C. The HPF1D283A bacterial expression plasmid was generated via inverse PCR from the parent pET30 plasmid containing the HPF1 construct and transformed into Rosetta2 (DE3) cells. It was purified in the same way as described for HPF1.

## ARH3

A pET30 plasmid harboring the 6xHis-SUMO-ARH3 protein (addgene plasmid #: 111578) was transformed into Rosetta2 (DE3) cells and inoculated into 6 L of Luria Broth (Miller). Protein expression was induced with 0.5 mM IPTG at a cell OD$_{600}$ of 0.6. Expression was carried out at 18 °C for 16 h. Cells

were harvested by centrifugation and protein was purified using Ni-NTA resin followed by reverse nickel and size-exclusion chromatography (SEC) in a manner similar to that described for HPF1. Pure fractions from the SEC (as judged by SDS-PAGE) were concentrated to around 600 µM, flash frozen in single-use aliquots, and stored at –80 °C.

## PARG

A PARG gene fragment encoding amino acids 448–976 was synthesized by Integrated DNA Technologies and cloned into a modified pET30 vector via Gibson Assembly to produce an *E. coli* expression plasmid for the 6xHis-SUMO-PARG construct. The plasmid was transformed into Rosetta2 (DE3) cells and inoculated into 2 L of Luria Broth (Miller). Protein expression was induced with 0.5 mM IPTG at a cell $OD_{600}$ of 0.6, and carried out at 18 °C for 16 h. Cells were harvested by centrifugation and protein was purified using Ni-NTA resin followed by reverse nickel and size-exclusion chromatography (SEC) in a manner similar to that described for HPF1. Pure fractions from the SEC (as judged by SDS-PAGE) were concentrated to around 300 µM, flash frozen in single-use aliquots, and stored at –80 °C.

## ALC1 macrodomain

The full-length ALC1 gene was synthesized by Twist Biosciences. A fragment encoding amino acids 636–878, corresponding to the macrodomain (*Singh et al., 2017*), was cloned into a modified pET30 vector via Gibson Assembly to produce an *E. coli* expression plasmid for the 6xHis-SUMO-ALC1macrodomain construct. The plasmid was transformed into Rosetta2 (DE3) cells and inoculated into 6 L of Luria Broth (Miller). Protein expression was induced with 0.5 mM IPTG at a cell $OD_{600}$ of 0.6. Expression was carried out at 18 °C for 16 h. Cells were harvested by centrifugation and disrupted via sonication in a lysis buffer containing 50 mM Tris, pH 7.5, 500 mM NaCl, 5 mM β-ME and 1 mM PMSF. Soluble lysate was isolated via centrifugation at 40,000 RCF for 30 min at 4 °C. Target protein was captured on Ni-NTA resin that was pre-equilibrated in lysis buffer. Following 1 h batch binding, resin was washed with 50 CV of lysis buffer supplemented with 25 mM imidazole, and then 2 CV of lysis buffer supplemented with 80 mM imidazole, and target protein was eluted in lysis buffer supplemented with 300 mM imidazole. The elution was dialyzed into a buffer containing 50 mM Tris, pH 7.5, 200 mM NaCl, and 5 mM TCEP for 16 h at 4 °C in the presence of Ulp1 to cleave the SUMO tag. The dialysate was then incubated with Ni-NTA resin pre-washed with the dialysis buffer for 1 h at 4 °C to capture the cleaved SUMO tag and the Ulp1, and the flow-through containing the target protein was collected. The flow-through was concentrated to 2 mL using an Amicon Ultra Centrifugal filter (Millipore; 30 kDa MWCO) and injected into a gel filtration column (HiLoad 16/60 Superdex 200) that had been pre-equilibrated with a buffer containing 40 mM Tris, pH 7.5, 200 mM NaCl, 10% glycerol, and 2 mM TCEP. Pure fractions (as judged by SDS-PAGE) were concentrated to around 400 µM, flash frozen in single-use aliquots, and stored at –80 °C.

## ALC1 ATPase domain

A fragment of the ALC1 gene encoding amino acids 1–673, corresponding to the ATPase domain (*Singh et al., 2017*), was cloned into a modified pET30 vector via Gibson Assembly to produce an *E. coli* expression plasmid for the 6xHis-SUMO-ALC1-ATPasedomain construct. The plasmid was transformed into Rosetta2 (DE3) cells and inoculated into 6 L of Luria Broth (Miller). Protein expression was induced with 0.5 mM IPTG at a cell $OD_{600}$ of 0.6. Expression was carried out at 18 °C for 16 h. Cells were harvested by centrifugation and disrupted via sonication in a lysis buffer containing 50 mM Tris, pH 7.5, 500 mM NaCl, 5 mM β-ME and 1 mM PMSF. Soluble lysate was isolated via centrifugation at 40,000 RCF for 30 minutes at 4 °C. Target protein was captured on Ni-NTA resin that was pre-equilibrated in lysis buffer. Following 1 h batch binding, resin was washed with 50 CV of lysis buffer supplemented with 25 mM imidazole, and then 2 CV of lysis buffer supplemented with 50 mM imidazole, and target protein was eluted in lysis buffer supplemented with 300 mM imidazole. The elution was dialyzed into a buffer containing 50 mM Tris, pH 7.5, 500 mM NaCl, and 5 mM β-ME for 16 h at 4 °C in the presence of Ulp1 to cleave the SUMO tag. The dialysate was then incubated with Ni-NTA resin pre-washed with the dialysis buffer for 1 h at 4 °C to capture the cleaved SUMO tag and the Ulp1, and the flow-through containing the target protein was collected. The flow-through was concentrated to around 40 µM using an Amicon Ultra Centrifugal filter (Millipore; 30 kDa MWCO) and

centrifuged at 20,000 RCF for 10 min. The supernatant was supplemented with 10% glycerol, flash frozen in single-use aliquots, and stored at –80 °C.

## ALC1

The full-length ALC1 gene was cloned into a modified pACEBac1 vector via Gibson Assembly to produce the 6xHis-1xFLAG-ALC1 DNA construct. Bacmid and baculovirus preparation was performed as described for PARP1/2. Protein expression was achieved by treating 2 L of Sf9 cells at $2.0 \times 10^6$ cells per mL with 20 mL of P3 virus for 48 h. Cells were harvested by centrifugation and target protein was purified using anti-FLAG M2 magnetic resin in a procedure similar to that described for PARP2. Pure protein was concentrated using an Amicon Ultra Centrifugal filter (Millipore; 30 kDa MWCO) to around 20 µM, as determined by BSA standards in SDS-PAGE, flash frozen in single-use aliquots, and stored at –80 °C.

## CHD4

The full-length CHD4 gene was purchased from Horizon Discovery and cloned into a modified pACEBac1 vector via Gibson Assembly to produce the CHD4-1xFLAG DNA construct. Bacmid and baculovirus preparation was performed as described for PARP1/2. Protein production was achieved by treating 2 L of Sf9 cells at $2.0 \times 10^6$ cells per mL with 20 mL of P3 virus for 48 h. Cells were harvested by centrifugation and target protein was purified using anti-FLAG M2 magnetic resin in a procedure similar to that described for PARP2. Pure protein was concentrated using an Amicon Ultra Centrifugal filter (Millipore; 30 kDa MWCO) to around 20 µM, as determined by BSA standards in SDS-PAGE, flash frozen in single-use aliquots, and stored at –80 °C.

## Auto-ADP-ribosylated PARP1 (serine-linked)

The PARP1 purified by the above method was incubated in auto-ADP-ribosylation reactions with $NAD^+$ and activating DNA. An HPF1 was titration experiment was employed to identify two concentrations at which relatively short or long serine-linked ADP-ribose chains could be installed on PARP1. Reactions (5 mL) included 2 µM of purified recombinant PARP1, 5 µM of activating DNA, and 250 µM of $NAD^+$ and were incubated in a buffer containing 50 mM Tris (pH 7.5), 20 mM NaCl, 2 mM $MgCl_2$, 1 mM TCEP with either 5 µM or 25 µM HPF1 for 30 min at 30 °C. The reaction with 5 µM HPF1 yielded PARP1 automodified with long serine-linked ADP-ribose chains and that with 25 µM HPF1 yielded PARP1 automodified with short serine-linked ADP-ribose chains. After completion of the automodification reaction, the sample was injected onto a 5 mL Cytiva HiTrap Heparin column (GE) that was pre-equilibrated with low salt buffer (150 mM NaCl, 50 mM Tris pH 7.5, 2 mM βMe, 1 mM $MgCl_2$). The column was washed with 5 CV of low salt buffer and the protein was eluted using a gradient from the low salt buffer to a high salt buffer (1 M NaCl, 50 mM Tris pH 7.5, 2 mM βMe, 1 mM $MgCl_2$) over 20 CV at a flow rate of 3 mL/min. Fractions were analysed on SDS-PAGE and those containing pure protein were pooled, concentrated to around 20 µM, supplemented with 10% glycerol, flash-frozen into single-use aliquots and stored in –80 °C.

## Core histones (H2A, H2B, H3, H4)

Identical purification protocols were employed for each full-length histone. Expression plasmids were transformed into Rosetta2 (DE3) cells and inoculated into 1 L of Luria Broth (Miller). Protein expression was induced with 0.5 mM IPTG at a cell $OD_{600}$ of 0.6. Expression was carried out at 37 °C for 3 h. Cells were harvested by centrifugation and disrupted via sonication in a lysis buffer containing 40 mM Tris, pH 7.5, 0.3 M NaCl, 1 mM EDTA, 5 mM β-ME, and 1 mM PMSF. Following centrifugation at 20,000 RCF for 30 min at 4 °C, the inclusion body pellet was then washed with lysis buffer supplemented with 1% Triton X-100 and centrifuged at 20,000 RCF for 15 min. This wash was repeated two more times with the final wash being performed in the absence of Triton X-100. Next, recombinant histone protein was extracted from the insoluble pellet in a buffer containing 50 mM Tris, 7.5, 300 mM NaCl, 6 M guanidine hydrochloride, and 5 mM β-ME for 1 hour at 25 °C and centrifuged at 20,000 RCF for 30 min. The soluble extract was then centrifuged at 100,000 RCF, injected onto a preparative C18 RP-HPLC column equilibrated in Solvent A (0.1% TFA in water) and eluted via an isocratic gradient 20–80% Solvent B (90% acetonitrile, 0.1% TFA in water) over a period of 30 min. Pure fractions (as determined by LC–MS) were lyophilized and stored at −80 °C until use in histone octamer assembly.

## H2B (amino acids 17-125) and H3 (amino acids 21-135)

Identical protocols were employed for each truncated 6xHis-ketosteroid isomerase-SUMO-tagged histone. The ketosteroid isomerase tag (synthesized by IDT and incorporated into histone expression plasmids via Gibson Assembly) rapidly shuttles truncated histones to *E. coli* inclusion bodies to protect them from degradation and increase yield. Truncated histones were expressed and extracted as described for full-length histone constructs. Following extraction, the histones were immobilized on Ni-affinity resin in extraction buffer, washed with 50 mM Tris, pH 7.5, 300 mM NaCl, 6 M guanidine hydrochloride, 20 mM imidazole, and 5 mM β-ME, and eluted in wash buffer supplemented with 300 mM imidazole. The eluted protein was dialyzed for 16h at 4 °C into dialysis buffer (50 mM Tris, pH 7.5, 300 mM NaCl, 6 M urea, and 5 mM β-ME). Following dialysis, the sample was diluted three-fold with dilution buffer (50 mM Tris, pH 7.5, 300 mM NaCl, and 5 mM β-ME) in the presence of Ulp1 to cleave the ketosteroid isomerase-SUMO tag. This target proteins were then purified via preparative RP-HPLC and stored as described for full-length histone constructs.

## 601 DNA preparation

The 200bp template used to assemble all nucleosomes is shown below with the 601 sequence in bold, the PstI site in purple, and the overhangs underlined:

5′–<u>GGCCGCTCTAGAACTAGTGGATCCGATATCGCTGTTCACCGCGTG</u>**ACAGGATGTATATATCTGACACGTGCCTGGAGACTAGGGAGTAATCCCCTTGGCGGTTAAAACGCGGGGGGACAGCGCGTACGTGCGTTTAAGCGGTGCTAGAGCTGTCTACGACCAATTGAGCGG**CTGCAG**CACCGGGATTCTCCAG**<u>CATCAGAG</u>-3′.

The 601 sequence was purchased from IDT and incorporated into a pET30a plasmid via Gibson Assembly. DNA was amplified from the parent plasmid using Phusion polymerase and the primers shown in the *Supplementary file 10*. The PCR product was purified using QIAquick Spin Columns (Qiagen) following manufacturer's protocols. Following elution, an ethanol precipitation step was performed and DNA was resuspended to 1 µg/µL in water for use in nucleosome assembly.

To insert unique 5′ primer-binding sites for the nucleosome competition remodeling assays, primers bearing unique 5′ 15bp overhangs were employed in the protocol described above. Primer sequences are shown in the *Supplementary file 10*. The final template design is outlined below with the 601 sequence in bold, the PstI site in purple, the unique 5′ primer-binding site in blue, and the universal 3′ primer-binding site in red:

5′–nnnnnnnnnnnnnnnAGTGGATCCGATATCGCTGTTCACCGCGTG**ACAGGATGTATATATCTGACACGTGCCTGGAGACTAGGGAGTAATCCCCTTGGCGGTTAAAACGCGGGGGGACAGCGCGTACGTGCGTTTAAGCGGTGCTAGAGCTGTCTACGACCAATTGAGCGG**CTGCAG**CACCGGGAT**TCTCCAGCATCAGAG-3′.

For dinucleosomes, the 601 DNA was designed such that nucleosome A had asymmetric overhangs (45 bp on one side and 15 bp on the other) and nucleosome B had symmetric overhangs (15 bp on either side). The DraIII restriction site was mutated out of the parental 601 DNA plasmid, and the resultant plasmid was used as template to amplify the 601 DNA construct for nucleosome A:

5′–GGCCGCTCTAGAACTAGTGGATCCGATATCGCTGTT**CATCGCGTG**ACAGGATGTATATATCTGACACGTGCCTGGAGACTAGGGAGTAATCCCCTTGGCGGTTAAAACGCGGGGGGACAGCGCGTACGTGCGTTTAAGCGGTGCTAGAGCTGTCTACGACCAATTGAGCGG<u>CTGCAG</u>**CACCGGGATTCTCCAG**CAT<u>CACAGAGTG</u>AGGG-3′.

For nucleosome B, the template plasmid (without the DraIII site) was mutated once again to replace the PstI restriction site with a BamHI site. The resultant plasmid was then used as template for amplifying the 601 DNA construct for nucleosome B:

5′–CGCT<u>CACAGAGTG</u>GTG**ACAGGATGTATATATCTGACACGTGCCTGGAGACTAGGGAGTAATCCCCTTGGCGGTTAAAACGCGGGGGGACAGCGCGTACGTGCGTTTAAGCGGTGCTAGAGCTGTCTACGACCAATTGAGCGG**<u>GGATCC</u>**CACCGGGATTCTCCAG**CATCAGAGACCTAGG-3′.

For the dinucleosome constructs, the purple underlined sequence represents the PstI or BamHI sites and the plain underlined sequence represents the DraIII site. The italic sequence is the 601 sequence.

## Peptide synthesis

### General protocols

All fluorenylmethyloxycarbonyl (Fmoc)-protected amino acids were purchased from Oakwood Chemical or Combi-Blocks. Peptide synthesis resins (Trityl-OH ChemMatrix and Rink Amide ChemMatrix) were purchased from Biotage. All analytical reversed-phase HPLC (RP-HPLC) was performed on an Agilent 1,260 series instrument equipped with a quaternary pump and an XBridge Peptide C18 column (5 µm, 4 × 150 mm; Waters) at a flow rate of 1 mL/min. Similarly, semi-preparative scale purifications were performed employing a XBridge Peptide C18 semi-preparative column (5 µm, 10 mm × 250 mm, Waters) at a flow rate of 4 mL/min. Preparative RP-HPLC was performed on an Agilent 1,260 series instrument equipped with a preparatory pump and a XBridge Peptide C18 preparatory column (10 µM; 19 × 250 mm, Waters) at a flow rate of 20 mL/min. All instruments were equipped with a variable wavelength UV-detector. All RP–HPLC steps were performed using 0.1% (trifluoroacetic acid, TFA, Oakwood Chemical) in $H_2O$ (Solvent A) and 90% acetonitrile (Sigma-Aldrich), 0.1% TFA in $H_2O$ (Solvent B) as mobile phases. For LC/MS analysis, 0.1% formic acid (Sigma-Aldrich) was substituted for TFA in mobile phases. Gradients and run times are described in the characterization section for each molecule. Mass analysis was carried out for each product on an LC/MSD (Agilent Technologies) equipped with a 300 SB-C18 column (3.5 µM; 4.6 × 100 mm, Agilent Technologies) or a X500B QTOF (Sciex).

### Preparation of amidated peptides

Sequence of H3 (1–20)-CONH$_2$: **ARTKQTARKSTGGKAPRKQL-**CONH$_2$.
Sequence of H3S10A (1-20)-CONH$_2$: **ARTKQTARKATGGKAPRKQL-**CONH$_2$.
Sequence of H2B (1-16)-CONH$_2$: **PEPAKSAPAPKKGSKK-**CONH$_2$.
Sequence of H2BS6A (1-16)-CONH$_2$: **PEPAKAAPAPKKGSKK-**CONH$_2$
Sequence of PARP1 (501–515)- CONH$_2$: **AALSKKSKGQVKEEG-**CONH$_2$.
Sequence of PARP1S507A (501-515)- CONH$_2$: **AALSKKAKGQVKEEG-**CONH$_2$.
Sequence of TMA16 (2-19)- CONH$_2$: **PKAPKGKSAGREKKVIHP-**CONH$_2$.
Sequence of TMA16S9A (2-19)- CONH$_2$: **PKAPKGKAAGREKKVIHP-**CONH$_2$.

The above amidated peptides were synthesized via solid-phase peptide synthesis on a CEM Discover Microwave Peptide Synthesizer (Matthews, NC) using the Fmoc-protection strategy on Rink Amide-ChemMatrix resin (0.5 mmol/g). For coupling reactions, amino acids (5 eq) were activated with N,N'-diisopropylcarbodiimide (DIC, 5 eq, Oakwood Chemical)/Oxyma (5 eq, Oakwood Chemical) and heated to 90 °C for 2 min while bubbling with nitrogen gas in N,N-dimethylformamide (DMF, Oakwood Chemical). Fmoc deprotection was carried out with 20% piperidine (Sigma-Aldrich) in DMF supplemented with 0.1 M 1-hydroxybenzotriazole hydrate (HOBt, Oakwood Chemical) at 90 °C for 1 min while bubbling with nitrogen gas. The H3 Cleavage from the resin was performed with 92.5% TFA, 2.5% triisopropylsilane (TIS, Sigma-Aldrich), 2.5% 1,2-ethanedithiol (EDT, Sigma-Aldrich), and 2.5% $H_2O$ for 2 h at 25 °C. The crude peptide was then precipitated by the addition of a 10-fold volume of cold ether and centrifuged at 4000 RCF for 10 min at 4 °C. The pellet was resuspended in Solvent A and purified via preparative RP-HPLC using a linear gradient from 0 to 30% Solvent B over 30 min. Fractions were analyzed on analytical RP-HPLC and ESI-MS and those containing pure product ( > 95%) were pooled, lyophilized, and stored at –80 °C.

### Fluorescein-labeled H3.1 (1-20)-CONH$_2$, H2B (1-16)-CONH$_2$

Peptides were synthesized as described for the amidated species. Prior to cleavage, 5 (6)-carboxyfluorescein (3 eq, Sigma-Aldrich) was activated with PyAOP (3 eq, Oakwood Chemical) and N,N-diisopropylethylamine (DIPEA, 6 eq, Sigma-Aldrich) and coupled to the deprotected α-amine on resin for 30 min at 25 °C in DMF while bubbling with nitrogen gas. Resin was washed with DMF and treated with 20% piperidine in DMF prior to cleavage with 92.5% TFA, 2.5% TIS, 2.5% EDT, and 2.5% $H_2O$ for 2 h at 25 °C. The crude peptide was then precipitated by the addition of a 10-fold volume of cold ether and centrifuged at 4,000 RCF for 10 min at 4 °C. The pellet was resuspended in Solvent A and purified via preparative RP-HPLC using a linear gradient from 0 to 50% Solvent B over 40 min. Fractions were analyzed on analytical RP-HPLC and ESI-MS and those containing pure product ( > 95%) were pooled, lyophilized, and stored at –80 °C.

Sequence of H3.1 (1–20) -NHNH$_2$: **ARTKQTARKSTGGKAPRKQL-NHNH**$_2$.
Sequence of H2B (1-16) -NHNH$_2$: **PEPAKSAPAPKKGSKK-NHNH**$_2$.

## Synthesis of H3.1 (1-20) -NHNH$_2$, H2B (1-16) - NHNH$_2$

H3 (1–20) and H2B (1-16) containing C-terminal hydrazide were synthesized similarly to the amidated peptides described above with the following modifications. ChemMatrix Trityl-OH PEG resin (0.49 mmol/g) was washed with dichloromethane (DCM, Oakwood Chemical) and reacted with 5% (v/v) thionyl chloride (Sigma-Aldrich) in DCM for 90 minutes at 25 °C. Resin was washed with DCM and this step was repeated to ensure efficient resin chlorination. Next, the resin was washed with DCM, DMF, and 5% (v/v) DIPEA in DMF. The resin was reacted with 9-fluorenylmethyl carbazate (Combi-Blocks) in the presence of DIPEA (20 eq) in DMF for 2 h at RT. The resin was washed with DMF and the 9-fluorenylmethyl carbazate coupling step was repeated to ensure complete loading. The resin was washed with DMF and 5% (v/v) anhydrous methanol (Sigma-Aldrich) in DMF. For coupling reactions, amino acids (5 eq) were activated with DIC (5 eq) and Oxyma (5 eq) and heated to 50 °C for 10 min while bubbling with nitrogen gas in DMF. Fmoc deprotection was carried out with 20% piperidine in DMF supplemented with 0.1 M HOBt at 60 °C for 4 min while bubbling with nitrogen gas. Cleavage and purification were performed as described for amidated peptides.

For peptide thioesterification, purified peptides containing C-terminal hydrazide were dissolved in a de-gassed buffer of 6 M guanidine hydrochloride and 0.1 M sodium phosphate, pH 3.0. The reaction was initiated by adding sodium nitrite (15 eq, Sigma-Aldrich) at –15 °C 10 min. The pH was monitored and maintained at 3.0 throughout the reaction. Immediately following this reaction, MESNa (75 eq, Sigma-Aldrich) and TCEP (final concentration of 20 mM, GoldBio) were added and the pH was adjusted to 7.0. The mixture was incubated at 25 °C for additional 30 min and monitored by RP-HPLC and ESI-MS analyses. Once quantitative conversion was complete, the peptide was purified via preparative RP-HPLC with a linear gradient of 0–30% Solvent B over 30 min. Pure fractions were characterized as described for amidated peptides, pooled, lyophilized, and stored at –80 °C.

## Synthesis of H3 (21-34)-SEA, H3 (21-34, S28A)-SEA

Sequence of H3 (21–34)- SEA: Thz-**TKAARKSAPATGG-**SEA.
Sequence of H3S28A (21-34)- SEA: Thz-**TKAARKAAPATGG-**SEA where,
Thz = thiazolidine, and SEA = bis(2-sulfanylethyl)amido group.

H3 (amino acids 21–34) and the corresponding S28A mutant peptides containing N-terminal thiazolidine and C-terminal SEA were synthesized similarly to the amidated peptides described above with the following modifications. SEA resin (0.16 mmol/g; Iris Biotech) was weighed out, washed with DMF and bubbled in nitrogen for 15 min to swell the resin. Fmoc-glycine (5 eq) and HATU (1-Bis(dimethylamino)methylene-1H-1,2,3-triazolo [4,5-b]pyridinium 3-oxide hexafluorophosphate; 5 eq), and DIPEA (15 eq) were mixed in DMF and the resin was bubbled in this mixture for 1 h. This step was repeated with fresh reagents to ensure complete loading. The resin was then washed with DMF and bubbled in acetic anhydride:DIPEA (20 eq:40 eq) in DMF for 20 min for acetyl capping. For coupling reactions, amino acids (5 eq) were activated with DIC (5 eq) and Oxyma (5 eq) and heated to 50 °C for 10 min while bubbling with nitrogen gas in DMF. Fmoc deprotection was carried out with 20% piperidine in DMF supplemented with 0.1 M HOBt at 60 °C for 4 min while bubbling with nitrogen gas. Cleavage and purification were performed as described for amidated peptides. The use of thiazolidine offers a way to keep the thiol of the N-terminal cysteine protected while performing native chemical ligation on the C-terminus of the peptide.

## Recombinant PARP1:HPF1 complex ADPr activity assays and analysis

### General protocols

To analyze PARP1:HPF1 ADPr activity on synthetic peptide substrates, 1 μM PARP1 (or 2 μM for H2B peptides), 10 μM HPF1, 2 mM NAD$^+$ (Sigma-Aldrich), and 1 μM stimulating DNA (or 2 μM for H2B peptides; see *Supplementary file 10* for stimulating DNA sequence information) were combined into the ADPr reaction buffer (50 mM Tris, pH 7.5, 20 mM NaCl, 2 mM MgCl$_2$, 5 mM TCEP) at a final volume of 25 μL (or 50 μL for H2B peptides). All substrate peptides were initially analyzed at a concentration of 180 μM (40 μM for H2B peptides). The reaction was then incubated at 30 °C for 25 min and

quenched via addition of Solvent A to a final volume of 120 µL. Reactions were then centrifuged at 20,000 RCF for 5 min and 100 µL of the supernatant was injected onto an analytical C18 column for product analysis via RP-HPLC. An elution gradient of 0–35% Solvent B over 20 min was employed to separate the poly-ADP-ribosylated peptide products. Individual peaks corresponding to products with mono-, di-, tri-, tetra-, or penta-ADP-ribose were collected and analyzed by ESI-MS.

## Fluorescent peptide ADPr

Reaction volumes were scaled to 1 mL to obtain sufficient amounts of each purified product for fluorescence polarization assays. For purification via semi-preparative RP-HPLC, an elution gradient of 5–20% Solvent B over 40 min was employed to optimize separation of the poly-ADP-ribosylated peptide products. Peaks corresponding to products with mono-, di-, tri-, tetra-, or penta-ADP-ribose were collected separately, analyzed by ESI-MS, and the pure fractions were pooled, lyophilized, and stored at –80 °C. The reaction and purified peptides were kept wrapped in aluminum foil whenever possible. We note that 5 (6)-carboxyfluorescein causes peak splitting in HPLC characterization corresponding to individual fluorescein isomers. This phenomenon was unique to fluorescein-labeled peptides and peak resolution varied based on ADP-ribose chain length.

## HPF1 titration analysis on unmodified peptide substrates

For the HPF1 titration experiments described in *Figure 1E*, reactions were performed as described above in the presence of 0, 5, 10, 20, 50, or 100 µM HPF1. Histone peptide starting material was quantified via integration of the corresponding HPLC peak at $A_{214}$. Peak area was converted to peptide concentration via a standardization curve that was generated using known quantities of substrate peptide. ADP-ribosylated peptide products were quantified via integration of corresponding HPLC peaks at $A_{280}$. Peak areas were then converted to peptide concentrations via a standardization curve that was generated using known quantities of ADP-ribosylated peptides. Standardization curves were generated for the mono- and di- ADP-ribosylated products. We note that the peptide HPLC $A_{280}$ signal is dependent upon the ADP-ribose moiety and no $A_{280}$ signal is present for any unmodified peptides used in this study. Therefore, there is a linear relationship between product extinction coefficient at 280 nm and the number of ADP-ribose units that are attached to the peptide. This linear increase in extinction coefficient was extrapolated to quantify all products with chain lengths greater than or equal to di-ADP-ribose. All reactions and standardization curve samples were run on the same C18 column and HPLC instrument using identical mobile phase gradients. All reactions were performed in triplicate and error bars represent standard deviations.

The following formula was used to calculate percent conversion to each product in a given reaction:

$$\% \ conversion = \left\{ \frac{[ADPr_n]}{[unmodified] + \sum [ADPr_n]} \right\} \times 100$$

In the above formula:

$[ADPr_n]$ represents the concentration of an individual product modified with mono-, di-, tri-, tetra-, or penta-ADP-ribose [unmodified] represents the concentration of the unmodified peptide starting material $\sum [ADPr_n]$ represents the sum total concentration of all detectable ADP-ribosylated products.

## Optimized peptide mono-ADPr preparation

Optimal yield of mono-ADP-ribosylated peptides was achieved via a reaction of 1 µM PARP1 (or 2 µM PARP1 for H2B peptides), 20 µM HPF1, 5 µM of PARG, 10 mM $NAD^+$, 0.5 mM unmodified substrate peptide, and 3 µM stimulating DNA (or 6 µM for H2B peptides) in ADPr reaction buffer. Reactions were incubated at 30 °C for 30 min and quenched via addition of 6 M guanidine hydrochloride and 0.1 M sodium phosphate. Purification was carried out on a preparative RP-HPLC C18 column and characterization by ESI-MS and glycohydrolase treatment was performed. We have scaled to as high as 15 mL reaction volume and 2 mM substrate peptide. Percent conversion of peptide starting material drop precipitously at higher substrate peptide concentrations. Importantly, 5 µM of PARG is included throughout this reaction to cleave all poly-ADP-ribosylated products back to mono-ADP-ribose. We also noticed that PARG enhances percent conversion at higher peptide concentrations and is necessary for quantitative conversion under the conditions described here. We suspect this is because PARG reverses PARP1 auto-poly-ADPr that accumulates throughout the reaction. Notably, auto-ADPr

abrogates the PARP1:DNA interaction and inactivates the enzyme (*Kim et al., 2004*). For mono-ADPr of 0.5 mM of PARP1 (501–515) or TMA16 (2-19) peptide, 2 µM PARP1, 20 µM HPF1, 8 µM activating DNA, 2 µM PARG, and 10 mM NAD$^+$ was used. For mono-ADPr of 0.5 mM of H3 (21–34) SEA peptide, 2.5 µM PARP1, 25 µM HPF1, 10 µM activating DNA, 2 µM PARG, and 10 mM NAD$^+$ was used.

## Recombinant PARP1/2 ADPr polymerization activity assays and analysis

### General protocols

To analyze PARP1 and PARP2 ADPr activity on peptide substrates, 2 mM NAD$^+$ and 1 µM stimulating DNA were combined into the ADPr reaction buffer in the presence of 0.2, 1, or 5 µM PARP1 or PARP2, in a 25 µL reaction (2 µM PARP1/2, 2 µM DNA and 50 µL reaction volume for H2B peptides). All unmodified and mono-ADP-ribosylated substrate peptides were analyzed at a concentration of 180 µM (40 µM for H2B peptides). The reaction was incubated at 30 °C for 25 min, quenched via addition of 95 µL (70 µL in case of H2B peptide reactions) of Solvent A. It was then centrifuged at 20,000 RCF for 5 min and 100 µL of the supernatant was injected into an analytical C18 column for product analysis via RP-HPLC. An elution gradient of 0–35% Solvent B over 20 min was employed to optimize separation of the poly-ADP-ribosylated peptide products. Percent substrate turnover was calculated by integrating the peaks for the starting material and each product on the RP-HPLC A$_{280}$ trace, normalizing them depending on their number of ADP-ribose moieties, and calculating ratio of total product to total peptide amounts for each reaction. All reactions were performed in triplicate and error bars represent standard deviations. Individual peaks corresponding to products with unique ADP-ribose chain lengths were collected and analyzed by ESI-MS. For comparison of the HPLC traces, they were plotted on the same graph by using a common x-axis.

### HPF1 titration analysis on mono-ADP-ribosylated peptide substrates

To analyze HPF1-dependent inhibition of PARP1/2 elongation activity, elongation reactions were performed as described above in the presence of 0, 5, 10, 20, 50, or 100 µM of HPF1 or HPF1D283A, 1 µM of PARP1 or PARP2, and 180 µM of mono-ADP-ribosylated H3 peptide. Percent conversion of the mono-ADP-ribosylated peptide substrates to poly-ADP-ribosylated peptide products was calculated by integrating the peaks for the starting material and each product on the RP-HPLC A$_{280}$ trace. Peak areas were again converted to molar concentrations (as described in *HPF1 titration analysis on unmodified peptide substrates*).

The following formula was used to calculate fraction elongated in a given reaction:

$$fraction\ elongated = \frac{\sum[ADPr_{poly}]}{[ADPr_1] + \sum[ADPr_{poly}]}$$

In the above formula:

[ADPr$_1$] represents the concentration of the mono-ADP-ribosylated peptide starting material $\Sigma$[ADPr$_{poly}$] represents the sum total concentration of all detectable poly-ADP-ribosylated peptide products (products modified with di-, tri-, tetra-, or penta-ADP-ribose*) *in *Figure 2B and C*, the data is represented as the sum total of all poly-ADP-ribosylated peptide products. For distribution of product species, see *Figure 2—figure supplement 1A*, B and E. The relative fraction elongated at any concentration of HPF1 was calculated as:

$$relative\ fraction\ elongated = \frac{fraction\ elongated}{fraction\ elongated\ when\ [HPF1]=0}$$

### Optimized peptide poly-ADPr preparation

When optimal yield of poly-ADP-ribosylated peptides is desired, a reaction of 1 µM PARP1 (2 µM PARP1 for H2B peptides), 10 mM NAD$^+$, 500 µM mono-ADP-ribosylated substrate peptide, and 3 µM stimulating DNA (6 µM for H2B peptides) in ADPr reaction buffer is employed. Reactions are incubated at 30 °C for 30 min and quenched via addition of 6 M guanidine hydrochloride and 0.1 M sodium phosphate. Purification is carried out on a preparative RP-HPLC C18 column and characterization by ESI-MS and glycohydrolase treatment is performed. We have scaled to as high as 14 mL reaction volume and 1 mM mono-ADP-ribosylated substrate peptide. For poly-ADPr of 0.5 mM of mono-ADP-ribosylated PARP1 (501–515) peptide, 5 µM PARP1, 25 µM activating DNA, and 10 mM NAD$^+$ was used. For poly-ADPr of TMA16 (2-19) peptide, 2 µM PARP1, 8 µM activating DNA, and

10 mM NAD$^+$ was used. For poly-ADPr of 0.5 mM of H3 (21–34) SEA peptide, 5 µM PARP1, 40 µM activating DNA, and 10 mM NAD$^+$ was used.

## Glycohydrolase activity assays

For histone H3 peptide analysis, ADPr reactions containing 1 µM PARP1, 10 µM HPF1, 2 mM NAD$^+$, and 1 µM stimulating DNA, and 180 µM H3S10ADPr$_1$ peptide were combined into the ADPr reaction buffer at a final volume of 75 µL. Following a 25 min incubation at 30 °C, the reaction was quenched with 10 µM Olaparib (Selleckchem) and 25 µL was removed for pre-glycohydrolase treatment analysis. ARH3 or PARG was then added to a final concentration of 3 µM or 1 µM, respectively, and incubated at 37 °C for 2 h. Pre- and post-glycohydrolase-treated samples were then analyzed via analytical RP-HPLC on a C18 column using an elution gradient of 0–35% Solvent B over 20 min. Product identities were verified by ESI-MS.

We note that for H3 and H2B, glycohydrolase analysis was performed after the PARP1 elongation reaction from the mono-ADP-ribosylated peptide. This is because only low levels of poly-ADP-ribosylated products could be generated in the PARP1:HPF1 reaction. Elongation was much more efficient from mono-ADP-ribosylated peptides in reactions that lacked HPF1. For H2B product glyco-hydrolase analysis, peptide ADPr reactions containing 2 µM PARP1, 2 mM NAD$^+$, 40 µM mono ADP-ribosylated substrate peptide, and 2 µM stimulating DNA were combined into the ADPr reaction buffer at a final volume of 150 µL. Following a 25 min incubation at 30 °C, the reaction was quenched with 10 µM Olaparib and 50 µL was removed for pre-glycohydrolase treatment analysis. ARH3 or PARG was then added to a final concentration of 3 µM or 1 µM, respectively, to 50 µL of the elongated reaction and incubated at 37 °C for 2 hr. Pre- and post-glycohydrolase-treated samples were then analyzed via analytical RP-HPLC on a C18 column using an elution gradient of 0–35% Solvent B over 20 min. Product identities were verified by ESI-MS.

## LC-MS/MS analysis of PAR chains on a peptide substrate

To analyze branching of PAR chains installed on peptide substrates using our technology, we utilized the LC-MS/MS-based approach outlined by Chen, et al (*Chen et al., 2018*). PAR chains from a purified tetra-ADP-ribosylated H2B (1-16) peptide were subjected to treatment with Alkaline Phosphatase (Sigma-Aldrich) and Phosphodiesterase I (Sigma-Aldrich). ADP-ribosylated peptide (80 µM) was incubated with ~8 units of Phosphodiesterase I and ~300 units of alkaline phosphatase at 30 °C overnight in a 0.5 mL reaction in a buffer containing 50 mM Tris (pH 7.5), 20 mM NaCl, 2 mM MgCl$_2$, and 5 mM TCEP. The reaction products were then desalted and deproteinized via RP-HPLC (C18 column) and lyophilized. The lyophilized powder containing a mixture of the digestion products was then resuspended in water to a final concentration of 20 µg/mL and 12 µL was injected onto Phenomenex Synergi Polar-RP column (150 × 2 mm, 4 µm packing) and analyzed by LC-MS/MS using a Sciex QTRAP 6500+ mass spectrometer coupled to a Shimadzu Nexera X2 UPLC. The chromatographic conditions were as follows: Solvent A: dH$_2$0 +0.2% acetic acid, Solvent B: acetonitrile +0.2% acetic acid; flow rate: 0.46 mL/min; 0–2 min 1% B, 2–2.5 min gradient to 78% B, 2.5–3 min 78% B, 3–3.1 min gradient to 80% B, 3.1–3.5 min 80% B, 3.5–3.6 min gradient to 95% B, 3.6–6 min 95% B, 6.5 min gradient to 1% B, 6.5–7.5 min 1% B. Analytes were detected with the mass spectrometer in MRM (multiple reaction monitoring) mode by following the precursor to fragment ion transitions as follows: Adenosine 268 → 136 (2.27 min retention time), ribosyl-adenosine 400 → 268 and 136 (3.08 min retention time), diribosyl-adenosine 532.18 → 400, 268 and 136 (3.5 min retention time). Peaks were integrated and peak areas were determined by AB Sciex Analyst 1.71 with HotFix one software.

## PARP1 pull-down assays

### Immunoprecipitations with PARP1 and HPF1

A 100 µL solution of 5 µM FLAG-HPF1 (or HPF1D283A), 1 µM PARP1, 10 µM Olaparib, and 10 µM stimulating DNA in Pull-Down Buffer (50 mM Tris, pH 7.5, 50 mM NaCl, 2 mM MgCl$_2$, 0.1% Triton X-100, 1 mM DTT) was incubated for 25 min at 25 °C. This solution was then centrifuged at 20,000 RCF for 10 min and the supernatant was added to 10 µL of Anti-FLAG M2 magnetic resin (MilliporeSigma; pre-equilibrated in Pull-Down Buffer), after keeping aside 30 µL from the reaction as an input control for SDS-PAGE gel analysis. Resin was incubated on an end-over-end rotator at 4 °C for 30 min, washed for 3 times for 1 min each with 0.5 mL of Pull-Down Buffer, and eluted via incubation

in 2 X SDS loading dye at 95 °C for 5 min. Samples were analyzed on 10% SDS PAGE Bis-Tris gel and imaged via Coomassie Brilliant blue staining on a BioRad ChemiDoc.

## Immunoprecipitations with PARP1, nucleosomes, and ALC1

A 50 µL solution of 100 nM FLAG-ALC1, 50 nM unmodified or H3S10ADPr$_3$ nucleosomes, and 100 nM PARP1 (unmodified, PARP1 SerADPr$_{long}$, or PARP1 SerADPr$_{short}$) in IP buffer (100 mM KCl, 25 mM HEPES pH 7.9, 2 mM MgCl$_2$, 5% glycerol, 0.1% NP-40, 1 mM DTT) was incubated at 30 °C for 15 min. Binding reactions were then added to anti-FLAG M2 magnetic resin (MilliporeSigma; pre-equilibrated in IP Buffer) after keeping aside 5 µL as an input control for western blot analysis. Resin was incubated on an end-over-end rotator at 4 °C for 1 h, washed for three times for 1 min each with 0.5 mL of IP Buffer (with very gentle vortexing), and eluted via incubation in 2 X SDS loading dye at 95 °C for 5 min. Samples were run on 10% SDS PAGE Bis-Tris gels, analyzed via western blot and imaged on a BioRad ChemiDoc.

## Fluorescence polarization-based peptide interaction assays

Each fluorescently-labeled H2B (unmodified, mono-, di-, tri-, and tetra-ADP-ribose at H2BS6) and H3 peptide (unmodified, mono-, di-, tri-, tetra-, or penta-ADP-ribose at H3S10) was diluted to 2 nM in a buffer containing 25 mM Tris, pH 7.5, 100 mM NaCl, 2 mM MgCl$_2$, 0.001% Triton X100, and 1 mM DTT. Note, H3 elongates more efficiently than H2B and so the penta-ADP-ribosylated species could be isolated for this peptide. Peptide concentration was calculated via fluorescein extinction coefficient (A$_{480}$ = 70,000). To analyze peptide:pan-ADP-ribose detection reagent (Af1521 macrodomain fused to rabbit Fc tag, MilliporeSigma) interaction, the pan-ADP-ribose detection reagent was titrated into each peptide to final concentrations ranging from 0 to 2000 nM (points represent 3 x dilutions starting from 2000 nM; a higher concentration of 4000 nM was also included for mono- and di-ADP-ribosylated peptides). To analyze peptide:ALC1-macrodomain interaction, the ALC1-macrodomain was titrated into each peptide to final concentrations ranging from 0 to 3000 nM (points represent 3 x dilutions starting from 3000 nM). Reactions were added to a black, flat-bottom 96-well plate (Corning Costar) and analyzed on a BioTek Cytation five imager equipped with a Green FP filter set (excitation: 485 nm, emission: 528 nm). Polarization values were converted to anisotropy using the following formula: r=(2 P/(3 - P)) (*Lakowicz, 2006*). Following background subtraction and normalization, data was then processed in GraphPad Prism using a non-linear regression analysis to obtain K$_{d, app}$ values for each peptide:protein interaction. Error bars represent standard deviation value from three biological replicates.

## Assembly of full-length, ADP-ribosylated histones

Native chemical ligation reactions were performed by combining modified histone peptides bearing C-terminal MESNa moieties (H2BS6ADPr$_1$, H2BS6ADPr$_3$, H2BS6ADPr$_4$, H3S10ADPr$_1$, H3S10ADPr$_3$, or H3S10ADPr$_4$) with their corresponding recombinant C-terminal histone fragments (H2BA17C 17–125 or H3A21C 21–135). A typical reaction included 1 mM histone thioester peptide, 0.5 mM recombinant histone fragment, 20 mM TCEP, and 150 mM 2,2,2-trifluoroethanethiol (Sigma-Aldrich) in a degassed buffer of 6 M guanidine hydrochloride and 0.1 M sodium phosphate at pH 7.0. Reactions were incubated at 37 °C for 16 h and progress was monitored via RP-HPLC and ESI-MS analysis. Full-length histone products were purified on a semi-preparative C18 RP-HPLC column using a gradient from 10 to 80% Solvent B over 40 min. Fractions were analyzed via analytical C18 RP-HPLC and ESI-MS and those greater than 95% pure were pooled, lyophilized, and stored at –80 °C until use. All histones were characterized on 12% bis-tris SDS-PAGE gels (prepared in house) in MES running buffer.

## Preparation of histone octamers

Octamers and nucleosomes were prepared as previously described (*Luger et al., 1999*) with several modifications. Lyophilized recombinant and semi-synthetic histones were dissolved in a buffer containing 6 M guanidine hydrochloride, 20 mM Tris, pH 7.6, and 5 mM DTT at 4 °C. H2A, H2B, H3, and H4 were combined at a ratio of 1.2:1.2:1.0:1.0, respectively, and diluted to a final concentration of 1 mg/mL of total histone. The histone mixture was then injected into a Slide-A-Lyzer MINI dialysis cassette (3.5 kDa MWCO, ThermoFisher) and dialyzed at 4 °C into Octamer Refolding Buffer (10 mM Tris, pH 7.6, 2 M NaCl, 1 mM EDTA, and 1 mM DTT) for 20 h. The cassette was placed into

fresh Octamer Refolding Buffer at the 4 h and 16 h time-points during the dialysis. Next, the histone octamer solution was purified via gel filtration (Superdex 200 Increase 10/300 GL; GE Healthcare) that had been pre-equilibrated with Octamer Refolding Buffer. Injection volume did not exceed 0.5 mL to ensure efficient separation of histone octamers from sub-octamer species. Fractions containing the octamer complex (as judged by FPLC elution chromatogram and SDS–PAGE gel electrophoresis) were concentrated to 50 µM as quantified by $A_{280}$ for unmodified nucleosomes (extinction coefficient = 44,700) or $A_{260}$ for ADP-ribosylated nucleosomes (extinction coefficient = 13,500 x total ADP-ribose units), diluted two-fold with glycerol, and stored at a final concentration of 25 µM at −20 °C prior to nucleosome assembly. The following unique octamers were assembled for nucleosome preparation: unmodified, $H2BS6ADPr_1$, $H2BS6ADPr_3$, $H2BS6ADPr_4$, $H3S10ADPr_1$, $H3S10ADPr_3$, $H3S10ADPr_4$, $H2BS6/H3S10ADPr_3$, $H2BS6/H3S10ADPr_4$.

## Nucleosome assembly and characterization

Nucleosomes were assembled by combining 150 pmol histone octamer with 180 pmol 601 DNA in 75 µL of a buffer containing 2 M KCl, 10 mM Tris, pH 7.5, 0.1 mM EDTA, 1 mM DTT at 4 °C. The mixture was then injected into a Slide-A-Lyzer MINI dialysis button (3.5 kDa MWCO, ThermoFisher) and dialyzed against a buffer of 10 mM Tris, pH 7.0, 1.4 M KCl, 0.1 mM EDTA, 1 mM DTT at 4 °C for 1 h. Next, 350 mL of Nucleosome End Buffer (10 mM Tris, pH 7.5, 10 mM KCl, 0.1 mM EDTA, 1 mM DTT) was added at a rate of 1 mL/min. After 12 h, the cassette was dialyzed against Nucleosome End Buffer for 4 h with a fresh buffer exchange at the 2 h time-point. Following dialysis, precipitation was removed via centrifugation at 20,000 RCF for 10 min at 4 °C and $A_{260}$ of the supernatant was measured to calculate nucleosome concentration. Note that for individual remodeling experiments, all nucleosomes were assembled on an identical 601-containing 200 bp DNA template. For competition remodeling experiments, each nucleosome was assembled on the same 601-containing 200 bp DNA template except the 15 bp at the 5'-end were replaced with a unique priming sequence.

Nucleosome quality was analyzed by running the nucleosome on native PAGE on a 5% TBE gel in 0.5 X TBE buffer (BioRad) that was run for 60 min at 150 volts. Around 10 pmol of each nucleosome was incubated in 20 mL REA buffer for 1 h at 30 °C, and then supplemented with 12% sucrose prior to loading. Gels were stained with ethidium bromide and imaged on a BioRad ChemiDoc and the nucleosome band migrates around 500 bp. We noted that nucleosome migration is affected by ADP-ribose chain length. If any free 601 DNA was observed on the TBE gel, then a PstI (NEB) restriction digestion was performed to check if the free DNA was present in the nucleosome dialysate or was an artifact of the gel run. To further verify stability of ADPr throughout the histone octamer and nucleosome assembly, 2.5 pmol of nucleosome were run on a 12% SDS-PAGE gel and immuno-blot analysis was performed to detect ADP-ribose, H2B, and H3. ADP-ribosylated H2B and H3 proteins exhibit distinct migration profiles relative to the unmodified species, confirming that they are homogenously modified.

## Western blot protocol

SDS-PAGE gels were transferred to PVDF membranes at 100 volts for 1 h at 4 °C using a wet transfer protocol in Towbin Buffer (25 mM Tris, 192 mM glycine, 20% methanol, pH 8.3). Blots were then blocked for 1 h at 25 °C with 5% non-fat dry milk (BioRad) in TBST (50 mM Tris, pH 7.5, 150 mM NaCl, 0.1% Tween20) prior to incubation with primary antibodies for 12 h at 4 °C. Following primary antibody binding, blots were washed three times for 5 min each with TBST and then incubated with the appropriate fluorescent or HRP-conjugated secondary antibody for 1 h at 25 °C. Blots were then washed three times for a total of 15 min with TBST and imaged on a BioRad ChemiDoc. All antibodies used in this study and corresponding dilutions can be found in *Supplementary file 11*.

## Restriction enzyme accessibility-based nucleosome remodeling assay

REA assays and analysis were performed as previously described (*Dann et al., 2017*) with several modifications. Nucleosome remodeling reactions (25 µL) were carried out in REA Buffer (12 mM HEPES, pH 7.9, 4 mM Tris, pH 7.5, 60 mM KCl, 10 mM MgCl₂, 10% glycerol, and 0.02% NP-40) including 1 µL of PstI (NEB, at 100,000 U/mL), 2 mM ATP (Sigma-Aldrich), and final concentrations of 4 nM ALC1 or 10 nM CHD4 and 20 nM of the desired nucleosome substrate. The reaction was incubated for 5 min prior to addition of chromatin remodeler to ensure that any trace amount of free DNA from the

nucleosome assembly was digested prior to initiating the reaction. This is required to ensure that free DNA digestion can be assigned as background activity and is not interpreted as enzyme-dependent nucleosome remodeling in data processing. To each reaction, 37.5 μL of Quench Buffer (20 mM Tris, pH 7.5, 70 mM EDTA, pH 8, 2% SDS, 10% glycerol) was added at time points of 0, 3, 6, 18, 36, and 60 min. Samples were then deproteinized with 30 U/mL proteinase K (NEB) for 1 h at 37 °C. DNA purification was performed using the Qiagen PCR purification kit following manufacturer's protocols. Purple Gel Loading Dye (6 X, NEB) was added to a final concentration of 1 X to the quenched reaction and samples were loaded onto a 5% TBE gel and run for 60 min at 150 volts in 0.5 X TBE Buffer (BioRad). Gels were stained with ethidium bromide and imaged on a BioRad ChemiDoc. Gel densitometry measurements were performed using ImageJ. For each lane, the total densitometry signal was calculated by adding the densitometry values corresponding to the PstI-digested species (lower band) and undigested species (upper band). The fraction unremodeled value for each lane was then calculated using the following formula:

$$fraction\ unremodeled = \frac{signal_{undigested}}{signal_{undigested} + signal_{digested}}$$

For each chromatin remodeling reaction, activity at the zero time point was considered background activity (described above) and that value of fraction unremodeled was denoted as the reference for normalizing values from other time points in the corresponding reaction. Data was performed in biological triplicate and fit into one-phase exponential decay equation in GraphPad Prism to obtain the remodeling plots and corresponding k values (**Supplementary file 6**), where k denotes the rate constant for the exponential decay. For calculation of k values, plateau was constrained to zero and k > 0.

To probe ALC1 activation by freely diffusing ADP-ribosylated peptides, the H3S10ADPr$_4$ or H2BS6ADPr$_4$ (amino acids 1–20 or 1–16, respectively) peptides were added to the REA Buffer at 10, 40, or 200 nM. Remodeling reactions were then carried out as described for 1 h on unmodified nucleosomes (**Figure 5—figure supplement 1C**). Control reactions were also set up with the same concentrations of unmodified versions of the corresponding peptides. A full time-course (0, 3, 6, 18, 36, 60 min) was performed at the 40 nM peptide concentration. This concentration was selected for complete analysis (**Figure 5C**) because the final ADP-ribose concentration is equivalent to that of the modified nucleosome assays (20 nM nucleosome x 2 ADP-ribosylated histone tails per nucleosome). All reactions and controls were performed in triplicate. Similar experiments were carried out in which the H2BS6ADPr$_4$ peptide was titrated into ALC1 reactions containing the H2BS6ADPr$_4$ nucleosome. The enzyme concentration was lowered such that around 50% remodeling was observed at the end of 1 h. This was done to provide sufficient assay range for observing a stimulatory or inhibitory effect on the remodeling.

To investigate the role of the macrodomain in conferring substrate preference on ALC1, chromatin remodeling reactions with the ALC1 ATPase domain construct (amino acids 1–673) were conducted on unmodified and H2BS6ADPr$_4$ nucleosomes. A full time-course (0, 3, 6, 18, 36, 60 min) was performed using 20 nM nucleosome and 500 nM ALC1(1–673) in triplicate, and the reactions were analyzed as described.

To probe ALC1 activation by Asp-/Glu- auto-ADP-ribosylated PARP1, we added 5, 20, 50, 100, or 200 nM PARP1 and 2 mM NAD$^+$ to the remodeling assays. Remodeling reactions were then carried out as described for 1 h on unmodified nucleosomes (**Figure 5—figure supplement 2B**). A full time-course (0, 3, 6, 18, 36, 60 min) was performed at 20 and 100 nM PARP1 concentration in triplicate. These concentrations were selected for complete analysis because the stimulatory effect of ADP-ribosylated PARP1 on nucleosome remodeling by ALC1 plateaued at around 50 nM. Western blots were performed using the PARP1 antibody and the pan-ADPr-detection reagent at time-points 0, 3, 6, 18, 36, and 60 min to quantify conversion of PARP1 in the reaction to the auto-ADP-ribosylated species. Similar titrations were carried out for serine-linked auto-ADP-ribosylated PARP1 species and the stimulatory effect seemed to plateau around 100 nM, thus the full time-course (0, 3, 6, 18, 36, 60 min) remodeling experiments were performed at 100 nM serine automodified PARP1 concentration. Remodeling reactions for H2BS6ADPr$_4$ nucleosomes in the presence of 100 nM PARP1 SerADPr$_{long}$ were also carried out in a similar way. Again, as this was designed to test for an effect on an already activated remodeling reaction, the enzyme concentration was lowered such that around 50% remodeling was observed at the end of 1 h.

## Hydroxylamine treatment

A fresh stock solution of 3.3 M hydroxylamine was prepared in 10 mM Tris in water and adjusted to pH ~6 using filtered 5 M KOH. Automodified PARP1 constructs (1 µM) were added to a buffer containing 50 mM Tris (pH 7.5), 20 mM NaCl and 2 mM $MgCl_2$. Hydroxylamine was added to this solution to a final concentration of 0.8 M and the solution was incubated at room temperature for 1 h. The reaction was quenched using 0.3% HCl. SDS-PAGE loading dye was added to the samples at a final concentration of 1 X and boiled before being run on an SDS-PAGE gel. The bands on the gels were visualized via silver-stain.

## Nucleosome remodeling competition assay

Two nucleosome substrate pools were prepared for the competition assays, with each substrate pool containing seven unique species. The first pool (H2B Pool) included $H2BS6ADPr_1$, $H2BS6ADPr_3$, and $H2BS6ADPr_4$ nucleosomes, each of which contained a unique 5′ priming site as outlined in the '601 DNA preparation' section above. The second pool (H3 Pool) included $H3S10ADPr_1$, $H3S10ADPr_3$, and $H3S10ADPr_4$ nucleosomes, each of which contained a unique 5′ priming site. Each pool also included two unmodified nucleosomes assembled on unique templates to serve as internal reproducibility controls. Free DNA templates with the PstI site and an unmodified nucleosome without the PstI site were also included as internal controls for PstI activity and data normalization, respectively. To ensure that PCR amplification artifacts do not influence cycle threshold determination, we selected primer:-template pairs with similar primer efficiencies.

Each nucleosome substrate pool was prepared by combining equal volumes of each nucleosome (stock solutions = 250 nM) or free DNA species (stock solutions = 250 nM). Therefore, the total species concentration in each assembled substrate pool is 250 nM (~36 nM per species). The final total nucleosome species concentration used in remodeling assays was 20 nM. ALC1 was used at a concentration of 4 nM for the H2B substrate pool and 8 nM for the H3 substrate pool. Remodeling assays were carried out, quenched at six different time points (0, 3, 6, 18, 36, 60 min) and DNA was isolated as described in the 'Restriction enzyme accessibility-based nucleosome remodeling assay' section. Real-time PCR was then performed with each unique primer pair according to manufacturer's protocols (iTaq Universal SYBR Green Supermix, BioRad) to quantify undigested (that is, unremodeled) template for each unique species at every time point. Fold-decrease in template quantity from t = 0 to t = x was calculated by determining the ΔΔCt for a species of interest relative to the unmodified nucleosome lacking the PstI site. Note: the template lacking PstI site cannot be digested and thus serves as an internal control for ΔΔCt calculation. Fold-decrease in template quantity was then converted to fraction unremodeled. Each competition assay was performed in triplicate and data points take into account an average of three independent amplifications for each primer pair (see Supplementary Dataset for primer pair:substrate combinations). The data was processed in GraphPad Prism and fit into a one-phase exponential decay equation with plateau constrained to zero and k > 0 to obtain the remodeling plots and corresponding k values.

## Preparation of DNA for dinucleosome

The overall methodology of preparation of dinucleosomes was adapted with minor changes from a previously described protocol (*McGinty et al., 2008*). After PCR amplification of the 601 DNA constructs designed for nucleosome A and nucleosome B, DNA was purified using a Qiagen PCR purification kit and resuspended in water. Approximately 100 µg of 601 DNA was digested with 750 U of DraIII-HF (NEB, at 20,000 U/mL) in a total volume of 1 mL of 1 X CutSmart buffer (NEB) for 8 h at 37 °C. The 601 DNA was then purified again using a Qiagen PCR purification kit and resuspended in water to a concentration of 1 µg/µL.

## Dinucleosome formation and remodeling assays

The purified and DraIII-digested 601 DNA constructs (see '601 DNA preparation' for sequence information) were used to assemble mononucleosomes according to the protocol described in the section 'Nucleosome assembly and characterization'. Around 6.25 pmol of each nucleosome was used in a ligation reaction that contained 4000 U of T4 DNA ligase (NEB, at 400,000 U/mL) in a total volume of 200 µL. The resultant mixture was then concentrated to around 20 µL using Vivaspin 500 centrifugal concentrators (Sartorius, 10,000 MWCO) and analyzed on a 5% TBE gel, as described

for mononucleosomes. The approximate concentration of the dinucleosome was derived from the mononucleosome input concentration and estimated ligation efficiency from the gel analysis. The dinucleosome remodeling experiments were carried out with 5 nM of dinucleosome in each reaction, under conditions as described for the mononucleosome remodeling assays. Two time points (0 min and 60 min) were collected from the reaction and the fraction unremodeled was calculated by using the following formula:

$$fraction\ unremodeled = \frac{signal_{undigested}\ at\ 60\ min}{signal_{undigested}\ at\ 0\ min}$$

The restriction digestion of the dinucleosome DNA gives rise to two smaller products, which are close in size to the DNA fragments generated from the residual mononucleosomes present in the dinucleosome stock. In order to avoid interference from such DNA, only the band corresponding to the undigested dinucleosome DNA species was used for densitometry analyses for this experiment.

## Mammalian cell culture

HEK293T cells (Source: ATCC; authentication by STR profiling provided by ATCC; confirmed negative for mycoplasma by ATCC) were cultured in high-glucose DMEM (MilliporeSigma) supplemented with 10% Fetal Bovine Serum (Gibco), 100 units/mL of penicillin (Sigma), and 100 µg/mL of streptomycin (Sigma). Cells were maintained at 37 °C and 5% $CO_2$ and passaged/frozen down according to manufacturer's protocols (ATCC). Plasmid transfection was accomplished with Lipofectamine 2000 according to manufacturer's protocols (Invitrogen).

### Generation of ALC1 knockout cell lines

CRISPR-Cas9 plasmids (pSpCas9(BB)–2A-Puro (PX459) v2.0; Addgene plasmid #: 62988) targeting the ALC1 gene (for gRNA targeting sequences, see *Supplementary file 10*) were transfected into HEK293T cells. Targeting sequences were obtained using the Genetic Perturbation Platform (Broad Institute). After 24 h, 2 µg/mL of puromycin (Sigma) was added to growth medium and cells were selected for 48 hr. Puromycin was then removed, dead cells were washed away, and the adhering live cells were left to recover for 24 h prior to dilution for single colony selection. Clones were screened via western blot for ALC1 and those with no detectable ALC1 were frozen down and stored in liquid nitrogen.

## Nuclear extract preparation

Nuclear extract was prepared as previously described (*Carey et al., 2009*) with some modifications. The cells were dounced with a B-type pestle (Kontle Glass Co) until they were lysed. Lysis was confirmed by staining with Trypan Blue dye and visualizing under a microscope. The cell number was estimated using a hemocytometer and the volumes of the different buffers were added depending on that. The nuclear lysate was homogenized using pestle B until it was properly resuspended in Buffer C. The crude nuclear lysate was dialyzed into Buffer D in a dialysis tubing (FisherScientific, 6–8 kDa MWCO) at 4 °C, and dialysis was stopped before first signs of precipitation (around 3–4 h).

## Nuclear extract nucleosome remodeling assay

Nucleosome remodeling reactions (25 µL) were carried out in REA Buffer with 1 µL of PstI (NEB, at 100,000 U/mL), 2 mM ATP, and 8 µL of nuclear extract derived from either wild-type or ALC1 knockout HEK293T cells and 20 nM of the desired nucleosome substrate. The reactions were carried out at 30 °C and quenched with Quench Buffer at 0 min and 60 min time points. We note that reactions contain 0.5 mM DTT (carried over from the lysate storage buffer). The 601 DNA was isolated as described in 'Restriction enzyme accessibility-based nucleosome remodeling assay' section and analyzed on a 5% TBE gel. Western blot analyses of the ADPr profile of each nucleosome employed in this assay were carried out after incubation with or without either nuclear extract under identical reaction conditions.

## Preparation of H3S10/H3S28ADPr$_4$-modified C-terminal SEA peptide

The tetra-ADP-ribosylated species of H3 (1–20) thioester and H3 (21–34, N-terminal thiazolidine, C-terminal SEA) peptides were purified to homogeneity using RP-HPLC and lyophilized. The tetra-ADP-ribosylated H3 (21–34) SEA peptide was resuspended in a degassed buffer containing 6 M

guanidine hydrochloride, 0.1 M sodium phosphate pH 8.0% and 5% DMSO to a concentration of 1 mM. This reaction was rotated at room temperature for 16 h and SEA ring oxidation was confirmed by a –2 Da mass change via LC-MS. A stock of 1 M O-Methylhydroxylamine HCl (Combi-Blocks) was prepared in a degassed buffer containing 6 M guanidine hydrochloride and 0.1 M sodium phosphate pH 4.0, and added to the reaction to a final concentration of 200 mM. The pH of the reaction was adjusted to 4.0 and the reaction incubated at 37 °C for 2 h. Deprotection of the thiazolidine ring was confirmed by a –12 Da mass shift via LC-MS. The tetra-ADP-ribosylated H3 (1–20) thioester peptide was resuspended in a degassed buffer containing 6 M guanidine hydrochloride and 0.1 M sodium phosphate pH 7.0 to a concentration of 1 mM and a volume equal to that of the above reaction. The two peptide solutions were combined and 2,2,2-trifluoroethanethiol (Sigma-Aldrich) was added to a final concentration of 150 mM. The pH of this reaction was then adjusted to 7.0 and it was incubated at 37 °C for 16 h. The ligation product was purified over semi-preparative RP-HPLC and lyophilized. This product was resuspended in a buffer containing 6 M guanidine hydrochloride and 0.1 M sodium phosphate pH 7.0, and TCEP was added to it to a final concentration of 20 mM. The ligated product with a reduced, C-terminal SEA ring was confirmed by a single peak on RP-HPLC and a single mass on LC-MS. For synthetic scheme and characterization, see *Supplementary file 9*.

## Acknowledgements

We thank Dr. W Lee Kraus, Dr. Deepak Nijhawan, Dr. Steven McKnight, Dr. Benjamin Tu and members of the Liszczak laboratory for insightful discussions. We thank Dr. Andrew Lemoff and the UT Southwestern Proteomics Core for technical assistance. We also thank members of the laboratories of Dr. Ping Mu and Dr. Michael Roth for help with reagents. This work was supported by grants from the Welch Foundation (I-2039-20200401 to G L), the Cancer Prevention Research Institute of Texas (RR180051 to G L), and the American Cancer Society (UTSW-IRG-17-174-13). G L is the Virginia Murchison Linthicum Scholar in Medical Research.

## Additional information

### Funding

| Funder | Grant reference number | Author |
|---|---|---|
| Cancer Prevention and Research Institute of Texas | RR180051 | Glen Liszczak |
| Welch Foundation | I-2039-20200401 | Glen Liszczak |
| American Cancer Society | UTSW-IRG-17-174-13 | Glen Liszczak |

The funders had no role in study design, data collection and interpretation, or the decision to submit the work for publication.

### Author contributions

Jugal Mohapatra, Conceptualization, Data curation, Formal analysis, Investigation, Methodology, Validation, Visualization, Writing - original draft, Writing - review and editing; Kyuto Tashiro, Conceptualization, Data curation, Formal analysis, Investigation, Methodology, Writing - review and editing; Ryan L Beckner, Conceptualization, Data curation, Formal analysis, Investigation, Methodology, Writing - original draft, Writing - review and editing; Jorge Sierra, Data curation, Formal analysis; Jessica A Kilgore, Noelle S Williams, Data curation; Glen Liszczak, Conceptualization, Data curation, Formal analysis, Funding acquisition, Investigation, Methodology, Supervision, Validation, Visualization, Writing - original draft, Writing - review and editing

### Author ORCIDs

Jugal Mohapatra http://orcid.org/0000-0001-6604-3749
Kyuto Tashiro http://orcid.org/0000-0002-0174-1789
Glen Liszczak http://orcid.org/0000-0001-8194-5281

### Decision letter and Author response

Decision letter https://doi.org/10.7554/eLife.71502.sa1

Author response https://doi.org/10.7554/eLife.71502.sa2

---

## Additional files

### Supplementary files

• Supplementary file 1. RP-HPLC and ESI-MS characterization of all unmodified and ADP-ribosylated peptides described in this study.

• Supplementary file 2. LC-MS/MS analysis of enzymatic digestion products from PAR chains installed on a peptide using our technology. The spectra were produced on a Sciex QTRAP 6500+ mass spectrometer.

• Supplementary file 3. ESI-MS characterization of all fluorescein-labeled peptides described in this study.

• Supplementary file 4. Affinity measurements from fluorescent polarization-based binding assays described in *Figure 3*.

• Supplementary file 5. RP-HPLC and ESI-MS characterization of all thioester peptides and corresponding ligation products described in this study.

• Supplementary file 6. Chromatin remodeling rate constants for single-substrate assays with ALC1 and CHD4.

• Supplementary file 7. Chromatin remodeling rate constants for multi-substrate assays with ALC1.

• Supplementary file 8. Comparison of one-step and two-step enzymatic strategies to generate poly-ADP-ribosylated peptides. A, Representative HPLC chromatogram corresponding to a reaction comprising 40 μM unmodified H2B (amino acids 1–16) peptide, 2 μM PARP1 and 10 μM HPF1. B, Representative HPLC chromatogram corresponding to a reaction comprising 40 μM H2BS6ADPr$_1$ peptide and 2 μM PARP1. We observe that the presence of 10 μM HPF1, while necessary for maximum substrate turnover here, restricts the amount of poly-ADP-ribosylated products formed in the reaction. Hence, separation of the two enzymatic steps, that is, 'priming' of unmodified peptide with mono-ADP-ribose using PARP1:HPF1 and 'elongation' of mono-ADP-ribosylated peptide with PARP1, not only allows us to obtain longer ADP-ribose chains on the peptide but also makes the process more efficient in terms of resource consumption and yield.

• Supplementary file 9. Proof-of-concept for the strategy to synthesize site-specifically ADP-ribosylated proteins via C-terminal or sequential protein ligations. A, A schematic depicting the strategy for C-terminal ligation of ADP-ribosylated peptides. SEA = bis(2-sulfanylethyl)amido functional group. B, ESI-MS and RP-HPLC characterization of a C-terminal ligation product which has tetra-ADP-ribose on S10 and S28 residues of the H3 peptide (amino acids 1–34), as well as a reduced C-terminal SEA moiety. RP-HPLC gradient is from 0%–80% Solvent B (2–22 min). The yellow star represents ADP-ribose modification. 'n' and 'n'' represent the number of ADP-ribose units on the corresponding modification sites.

• Supplementary file 10. Sequences of the various DNA oligonucleotides described in this study.

• Supplementary file 11. Details of the various antibodies used in this study.

• Transparent reporting form

### Data availability

All data generated or analysed during this study are included in the manuscript and supporting files. Supplementary Dataset is available online on dryad- https://doi.org/10.5061/dryad.z612jm6cc.

The following dataset was generated:

| Author(s) | Year | Dataset title | Dataset URL | Database and Identifier |
|---|---|---|---|---|
| Mohapatra J, Liszczak G | 2021 | Data from: Serine ADP-ribosylation marks nucleosomes for ALC1-dependent chromatin remodeling | https://doi.org/10.5061/dryad.z612jm6cc | Dryad Digital Repository, 10.5061/dryad.z612jm6cc |

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
