## [Editor Report]

Poly-ADP-ribosylation (poly-ADPr) is a major histone modification that plays critical roles in DNA damage. However careful mechanistic dissection of the role of poly-ADPr has been challenging as the modification is found on multiple proteins and there is heterogeneity in terms of poly-ADP-ribosylation chain length and amino acid location of attachment. The PARP1-dependent semi-synthetic strategy developed by the authors allows generation of nucleosomes with mono ADP ribose and defined lengths of poly-ADPr chains at specific histone serine residues. The utility of this method is clearly demonstrated by the authors' findings that ALC1, a chromatin remodeler that recognizes poly-ADPr is stimulated substantially by the presence of poly-ADPr on H2A and H3.

---

## [Decision Letter]

**Decision letter after peer review:**

Thank you for submitting your article "Serine ADP-ribosylation marks nucleosomes for ALC1-dependent chromatin remodeling" for consideration by *eLife*. Your article has been reviewed by 3 peer reviewers, one of whom is a member of our Board of Reviewing Editors, and the evaluation has been overseen by Jessica Tyler as the Senior Editor. The reviewers have opted to remain anonymous.

As you will see all reviewers found the work to be significant and enabling. Two of them also had some concerns and suggestions that need to be addressed before we can consider publication in *eLife*. When you submit your revision, please describe the changes you have made through your point-by-point responses to the individual comments by each reviewer.

Please let us know if you have any questions.

Essential revisions:

Please address the individual suggestions and comments made by reviewers 2 and 3.

*Reviewer #1 (Recommendations for the authors):*

Histone modifications play diverse roles in regulating DNA based transactions. Understanding the biochemical mechanisms underlying such roles has been greatly enabled by the ability to reconstitute chromatin templates with defined modifications. Poly-ADP-ribosylation (poly-ADPr) is a major modification that plays critical roles in DNA damage. However a few features have slowed careful mechanistic dissection of the role of poly-ADP-ribosylation. These features include that (i) the modification is found on multiple proteins such as histones and the PARP enzymes that deposit the modification and (ii) there is heterogeneity in terms of the poly-ADP-ribosylation chain length and the amino-acid location of the modification. Additionally, unlike with other histone modifications such as acetylation, methylation and ubiquitylation, generating homogenously modified poly-ADPr chromatin in vitro has been technically challenging. Here the authors generate nucleosomes with homogenously and site-specifically modified poly-ADPr marks through a series of readily separable steps. Using these modified nucleosomes the authors test the impact of the poly-ADPr modifications on the activity of the chromatin remodeler ALC1. Previous work has shown that ALC1 is activated by binding poly-ADPr chains that have 3+ units. However, it has not been clear which modified substrates are being recognized by ALC1. Using their site-specifically modified reagents the authors show clearly that poly-ADP-ribosylation of H2A (and H3) has a much larger effect on stimulating nucleosome remodeling than auto-ADP-ribosylated PARP.

Overall, the work in this resource article is carefully carried out with several thoughtful controls and provides a highly enabling new method for the chromatin field. What further raises the impact of this work is the use of this new method to make a significant new mechanistic finding about ALC1regulation. I think the work in this resource article will be of much interest to the chromatin community and those studying DNA damage.

*Reviewer #2 (Recommendations for the authors):*

This interesting manuscript from Liszczak and coworkers highlights the strength of a semisynthetic chemical approach to obtain both monomeric and well-defined polymeric ADP-ribosylated histones H2B and H3. The authors have applied these reagents toward dissecting the role of ADP-ribose in facilitating nucleosome remodeling by the macrodomain-containing remodeler ALC1. Unfortunately, the synthetic novelty is somewhat diminished by a very recent manuscript from Muir and coworkers in JACS reporting the semisynthesis of identical mono-ADP-ribosylated H2B and H3. However, the current manuscript focuses on biochemical roles of ADP-ribosylation that were not addressed by Muir, and also presents the elegant enzymatic extension of semisynthetic mono-ADP ribosylated histones to obtain well-defined polymers of ADP-ribose by using varying ratios of PARP1:HPF1. The rigorous series of experiments and interesting results, that begin to clarify prior hypotheses generated from cellular data, will be of broad interest to the readers of *eLife*. I recommend a few figure/text revisions and the inclusion of two additional experiments, with the reagents on hand, to help increase the clarity of techniques and conclusions presented.

One key limitation of the work – which the authors clearly state in the methods for assembly of full-length ADP-ribosylated histones – is the requirement for a Cys-containing mutant H2B and H3 in biochemical assays. While there is no doubt, given Muir's recent work and the authors' claims, that the Cys may be effectively desulfurized in future experiments, the current substrates tested did not include this step. Therefore, I would suggest the authors move their statement on desulfurization to the Results in the main text and cite the Muir JACS paper. Accordingly, Figure 4A should clearly show the thiol is present in the histone tail. Finally, given the risk of disulfide-tethering cysteine-containing proteins to this accessible thiol in the H2B/H3 tail, the authors should mention in the discussion if reducing agents were included for all enzymatic assays and whether assay products were analyzed to ensure Cys-containing H2B/H3 remained as monomeric species at the end of the assay period. Particularly when mononucleosomes were subjected to nuclear lysates in remodeling assays.

In Figure 4. It is surprising that H3 and H2B appear to migrate at the same size prior to ADP-ribosylation despite previous reports of their differing migration (see https://www.epicypher.com/products/nucleosomes/histone-octamer-recombinant-human for one example). The authors should indicate the percentage of polyacrylamide or any changes in gel running conditions from standard SDS-PAGE conditions that led to this result.

Supplementary Figure 5F. The PARP1-SerADPlong size clearly changes with hydroxylamine treatment (comparing lanes 3 and 4) indicating that a minority of ADP-ribose linkages are not to Ser in PARP1. I would suggest noting this in the main text as potential heterogeneity in the ADP-ribosylated protein being employed in assays. Particularly if the ADP-ribosylated PARP1 tested in biochemical assays was not first treated with hydroxylamine.

Two important biochemical assays are missing in the manuscript that would help readers understand the specific contributions of nucleosome ADP-ribosylation and PARP1 ADP-ribosylation toward remodeling by ALC1.

(A) The first assay is to investigate competition of ADP-ribosylated nucleosomes by ADP-ribosylated peptides. This would validate the authors' statement that, "in addition to disrupting an autoinhibited conformation, the modified histone tail:macrodomain interaction is crucial for presenting the ATPase domain to the nucleosome." If this statement were indeed true and no additional nucleosomal contacts were needed for ALC1 function, then ADP-ribosylated H2B/H3 peptides would be expected to effectively inhibit nucleosome remodeling of the corresponding ADP-ribosylated nucleosome.

(B) The second assay involves an ALC1-mediated remodeling experiment with ADP-ribosylated nucleosomes in the presence of ADP-ribosylated PARP1. The authors, surprisingly, only test ADP-ribosylated PARP1 with unmodified nucleosomes and the rationale for this is unclear if DNA-damage is indeed accompanied by histone ADP-ribosylation, which is also the focus of this manuscript. Results from an assay with ADP-ribosylated H2B containing nucleosomes would clearly indicate if nucleosome ADP-ribosylation remains relevant when PARP1 is also ADP-ribosylated, and if any additive effects are observed from both species being ADP-ribosylated.

Finally, the pooled mononucleosome remodeling experiment provides no additional information beyond the initial information gained from individual mononucleosome assays and the fact that a ADP-ribosylated histone peptide does not activate ALC1 on non-ribosylated nucleosomes. The peptide in trans assay result already indicated that ALC1 does not stay active after disengaging the ADP-ribosylated peptide. If the authors' goal was to demonstrate that ribosylated nucleosomes do not stimulate activity of ALC1 on non-ribsoylated nucleosomes, or that ALC1 "immediately turns off" after disengagement from a ADP-ribosylated nucleosome, then this needs to be tested in an asymmetric dinucleosome model wherein the two nucleosomes are tethered in close proximity and not diffusing independently of each other. A mixture of varying ADP-ribosylated mononucleosomes at nM concentrations is unlikely to be an approximation of the chromatin context of ALC1, where varying degrees of ribosylation may exist in close apposition. Hence, I suggest the authors reword the conclusion from the pooled experiment to suggest that ALC1 only acts on polyADP-ribosylated mononucleosomes within a mixture of varying ADP-ribosylation states and avoid speculation of its diffusivity.

*Reviewer #3 (Recommendations for the authors):*

– While authors show that multiple modified histones can be made by this method (including S28 modified H3), claiming broad applicability to preparation of ADPr proteins is a misnomer for the exact reasons listed in the last paragraph of the discussion. The claims of broad applicability should be revised accordingly.

– Protein modification community would benefit from better understanding of pros and cons of the polyADP ribosylation method presented here. Given a need for extensive HPLC purification of the mixtures formed in both cases, a side-by-side comparison to experimentally simpler one pot reactions with PARP1:HPF1 would help.

– It would be important to assess whether ALC1 itself is ADP ribosylated in the remodeling assays where PARP is supplemented together with NAD+. The supplemental figure 5D, while showing PARP1 is autoribosylated based on the band shift, suggests broad ADP ribosylation signal as well. ALC1 autoribosylation might contribute to relief of its autoinhibition, contributing to the slight enhancement in remodeling observed, rather than a direct interaction with autoribosylated PARP1.

– Remodeling in nuclear extracts: The representative gel results in Figure 6 supplement 1C of remodeling in the presence of nuclear extracts does not support the claim that, with poly ADP ribosylated substrates, remodeling is enhanced compared to that observed with ALC1 knockout extracts. The relative remodeled and unremodeled bands appear similar regardless of the presence of ALC1 in the extract. It appears the specificity of remodeling gained for the ADPr4 substrate in the presence of ALC1 might result from slightly less remodeling of the unmodified substrate in the presence of ALC1 compared to ALC1 knockout extract unmodified substrate remodeling. This point would be apparent if the data was plotted as raw fraction remodeled values or relative remodeled values calculated to the unmodified substrate with WT extract for all experiments in Figure 6E.

– More rigorous characterization is needed for the poly ADP ribosylated substrates used with nuclear extracts in Figure 6E to support the claim that no hydrolysis is present of the poly ADP ribose chains. The band shifts due to poly ADP ribosylation are less pronounced as shown in Figure 4C-4E and the band shift is decreased upon addition of extract to H2BS6/H3S10ADPr4 substrate, suggesting chain hydrolysis. Authors should test if the polyADP ribose chains are retained after incubation with nuclear extracts. Alternatively, authors could test nuclear extract activity with and without ALC1 in the presence of PARG to assess if the enhancement is abolished.

– If it still holds, any speculation on attenuation of ALC1 stimulation in nuclear extracts, as opposed to that observed with recombinant components?

---

## [Author Response]

Essential revisions:Please address the individual suggestions and comments made by reviewers 2 and 3.Reviewer #1 (Recommendations for the authors):Histone modifications play diverse roles in regulating DNA based transactions. Understanding the biochemical mechanisms underlying such roles has been greatly enabled by the ability to reconstitute chromatin templates with defined modifications. Poly-ADP-ribosylation (poly-ADPr) is a major modification that plays critical roles in DNA damage. However a few features have slowed careful mechanistic dissection of the role of poly-ADP-ribosylation. These features include that (i) the modification is found on multiple proteins such as histones and the PARP enzymes that deposit the modification and (ii) there is heterogeneity in terms of the poly-ADP-ribosylation chain length and the amino-acid location of the modification. Additionally, unlike with other histone modifications such as acetylation, methylation and ubiquitylation, generating homogenously modified poly-ADPr chromatin in vitro has been technically challenging. Here the authors generate nucleosomes with homogenously and site-specifically modified poly-ADPr marks through a series of readily separable steps. Using these modified nucleosomes the authors test the impact of the poly-ADPr modifications on the activity of the chromatin remodeler ALC1. Previous work has shown that ALC1 is activated by binding poly-ADPr chains that have 3+ units. However, it has not been clear which modified substrates are being recognized by ALC1. Using their site-specifically modified reagents the authors show clearly that poly-ADP-ribosylation of H2A (and H3) has a much larger effect on stimulating nucleosome remodeling than auto-ADP-ribosylated PARP.Overall, the work in this resource article is carefully carried out with several thoughtful controls and provides a highly enabling new method for the chromatin field. What further raises the impact of this work is the use of this new method to make a significant new mechanistic finding about ALC1regulation. I think the work in this resource article will be of much interest to the chromatin community and those studying DNA damage.

We thank reviewer one for the positive feedback and are excited to see that experts in the field appreciate the value of our work.

Reviewer #2 (Recommendations for the authors):This interesting manuscript from Liszczak and coworkers highlights the strength of a semisynthetic chemical approach to obtain both monomeric and well-defined polymeric ADP-ribosylated histones H2B and H3. The authors have applied these reagents toward dissecting the role of ADP-ribose in facilitating nucleosome remodeling by the macrodomain-containing remodeler ALC1.

Reviewer 2 brings up an excellent point and one that we have been very keen to explore. This comment, along with a related comment from Reviewer 2 (discussed further below), inspired us to generate asymmetrically ADP-ribosylated dinucleosome substrates for ALC1 remodeling experiments. Our results are presented in a new main text figure (Figure 6) and described in the Results section as follows:

Lines 489-508:

“three asymmetrically ADP-ribosylated dinucleosome constructs were prepared wherein: (i) both nucleosome A and B are unmodified (unmod^A^-unmod^B^), (ii) nucleosome A is unmodified and nucleosome B comprises H2BS6ADPr_4_ histones (unmod^A^-ADPr^B^), or (iii) nucleosome A comprises H2BS6ADPr_4_ histones and nucleosome B is unmodified (ADPr^A^-unmod^B^) (Figure 6D, Figure 6—figure supplement 1B). In all dinucleosome constructs, nucleosome A bears a 45 base pair DNA overhang and is separated from nucleosome B by a 15 base pair DNA linker to allow for remodeling to occur. By removing the PstI site from nucleosome B, we were able to specifically monitor ALC1 remodeling activity on nucleosome A using our REA assay. Following incubation with ALC1, robust nucleosome A remodeling activity was observed (~79% remodeled in 60 min) with the ADPr^A^-unmod^B^ construct as expected (Figure 6E, Figure 6—figure supplement 1C). In contrast, relatively modest nucleosome A remodeling activity was observed (~21% remodeled in 60 min) with the unmod^A^-ADPr^B^ construct. While these single time-point analyses cannot be directly compared to the mononucleosome remodeling rate constant analyses, the dinucleosome remodeling activity results are consistent with the automodified PARP1 experiments; PARP1 auto-ADPr and adjacent nucleosome ADPr both tether poly-ADP-ribose in close proximity to unmodified nucleosomes but neither is able to fully stimulate ALC1. These experiments demonstrate that ALC1 preferentially remodels binding-competent nucleosome substrates and target disengagement triggers rapid transition back to an inactive conformation. This mechanism likely minimizes the potential for freely diffusing, activated ALC1 to be present in the nuclear milieu.”

While we have not yet progressed to 12mer nucleosome array substrates, these results strongly indicate that ALC1 preferentially targets ADP-ribosylated nucleosomes even in the context of chromatin arrays. As our technologies mature, it will be interesting to explore if significantly longer histone ADP-ribose chains increase the ALC1 ‘radius of activity’ and stimulate greater activity towards adjacent unmodified substrates. However, the relatively poor ALC1 stimulation by automodified PARP1 constructs (which tether relatively long chains to the nucleosome substrate) suggest this may not be the case.

Unfortunately, the synthetic novelty is somewhat diminished by a very recent manuscript from Muir and coworkers in JACS reporting the semisynthesis of identical mono-ADP-ribosylated H2B and H3.

We note that we were unaware of this work until its publication, which occurred after submission of our manuscript to *eLife*. We hope the reader will appreciate that, despite many years of work in this area, there remain no methods to install homogenous poly-ADP-ribose chains onto synthetic peptides. Here we have developed strategies to prepare, purify, and characterize a wide array of poly-ADP-ribosylated peptides, and optimized downstream N- and C-terminal protein ligation protocols. Considering the overwhelming number of reported poly-ADP-ribose reader domains that do not engage mono-ADP-ribose(Teloni and Altmeyer, 2016), and the breadth of poly-ADPr-regulated cellular processes(Gibson and Kraus, 2012; Leung, 2014), we anticipate that our technology will be liberally deployed to better understand ADPr-mediated signaling pathways and mechanisms. Until new synthetic routes are developed to control ADP-ribose polymerization on synthetic peptides, the strategy reported herein is the sole gateway to studying poly-ADPr via protein chemistry. Thus, we believe our work represents a significant and impactful advance.

However, the current manuscript focuses on biochemical roles of ADP-ribosylation that were not addressed by Muir, and also presents the elegant enzymatic extension of semisynthetic mono-ADP ribosylated histones to obtain well-defined polymers of ADP-ribose by using varying ratios of PARP1:HPF1. The rigorous series of experiments and interesting results, that begin to clarify prior hypotheses generated from cellular data, will be of broad interest to the readers of eLife. I recommend a few figure/text revisions and the inclusion of two additional experiments, with the reagents on hand, to help increase the clarity of techniques and conclusions presented.One key limitation of the work – which the authors clearly state in the methods for assembly of full-length ADP-ribosylated histones – is the requirement for a Cys-containing mutant H2B and H3 in biochemical assays. While there is no doubt, given Muir's recent work and the authors' claims, that the Cys may be effectively desulfurized in future experiments, the current substrates tested did not include this step. Therefore, I would suggest the authors move their statement on desulfurization to the Results in the main text and cite the Muir JACS paper. Accordingly, Figure 4A should clearly show the thiol is present in the histone tail.

We agree that these suggestions are appropriate to ensure that desulfurization details are clear to all readers. We have modified Figure 4A as requested, moved our statement about desulfurization to the main text, and included the citation (Lines 344-349).

Finally, given the risk of disulfide-tethering cysteine-containing proteins to this accessible thiol in the H2B/H3 tail, the authors should mention in the discussion if reducing agents were included for all enzymatic assays and whether assay products were analyzed to ensure Cys-containing H2B/H3 remained as monomeric species at the end of the assay period. Particularly when mononucleosomes were subjected to nuclear lysates in remodeling assays.

We have now added a line to the main text stating that our ADP-ribosylated mononucleosome TBE gel analysis in Figure 4F was performed after incubation in remodeling assay conditions to ensure that no nucleosome disulfide tethering has occurred (Lines 345-347). We have used standard remodeling assay conditions that are typically used in the field to carry out ATP-dependent remodeling reactions (Dann et al., 2017; He et al., 2006). We also note that lysate assays contain 0.5 mM DTT (which is carried over from our nuclear lysate storage buffer).

In Figure 4. It is surprising that H3 and H2B appear to migrate at the same size prior to ADP-ribosylation despite previous reports of their differing migration (see https://www.epicypher.com/products/nucleosomes/histone-octamer-recombinant-human for one example). The authors should indicate the percentage of polyacrylamide or any changes in gel running conditions from standard SDS-PAGE conditions that led to this result.

This observation is correct. We have aligned our protein ladder with the protein ladder on the Epicypher image (both ladders have the same molecular weight markers) and it is clear that we use different gel percentage and/or running conditions. For histone characterization, we use 12% Bis-Tris SDS-PAGE gels (prepared in house) in 1X MES running buffer. We have now included this detail in both the Figure 4 legend (Lines 317-319) and the Methods section (Lines 1360-1362).

Supplementary Figure 5F. The PARP1-SerADPlong size clearly changes with hydroxylamine treatment (comparing lanes 3 and 4) indicating that a minority of ADP-ribose linkages are not to Ser in PARP1. I would suggest noting this in the main text as potential heterogeneity in the ADP-ribosylated protein being employed in assays. Particularly if the ADP-ribosylated PARP1 tested in biochemical assays was not first treated with hydroxylamine.

It is true that our PARP1 SerADPr_long_ contains a minor fraction of hydroxylamine-sensitive ADPr. This is likely due to the relatively low HPF1:PARP1 ratio that was required to produce long ADP-ribose chains. This was also one of the reasons we elected to prepare two samples (one with relatively short chains but negligible hydroxylamine sensitivity). Additionally, while hydroxylamine treatment is an option to ‘clear off’ Asp/Glu ADPr after the automodification reaction, the ADP-ribosylated Asp/Glu residues become covalently modified with a hydroxamic acid moiety(Zhang et al., 2013). Thus, hydroxylamine treatment introduces a different type of sample heterogeneity and so we elected to forgo this step.

For clarification, we have now included this note in the main text:

Lines 424-428:

“We note that a small amount of hydroxylamine-dependent ADP-ribose chain cleavage was observed in the PARP1 SerADPr_long_ sample. Thus, a small population of aspartate- and/or glutamate-linked ADPr is present in this sample, likely due to the relatively low HPF1:PARP1 ratio that was required to produce long ADP-ribose chains.”

Two important biochemical assays are missing in the manuscript that would help readers understand the specific contributions of nucleosome ADP-ribosylation and PARP1 ADP-ribosylation toward remodeling by ALC1.(A) The first assay is to investigate competition of ADP-ribosylated nucleosomes by ADP-ribosylated peptides. This would validate the authors' statement that, "in addition to disrupting an autoinhibited conformation, the modified histone tail:macrodomain interaction is crucial for presenting the ATPase domain to the nucleosome." If this statement were indeed true and no additional nucleosomal contacts were needed for ALC1 function, then ADP-ribosylated H2B/H3 peptides would be expected to effectively inhibit nucleosome remodeling of the corresponding ADP-ribosylated nucleosome.

We apologize for the confusion regarding the statement: "in addition to disrupting an autoinhibited conformation, the modified histone tail:macrodomain interaction is crucial for presenting the ATPase domain to the nucleosome." We did not mean to imply that the histone tail:macrodomain is the only crucial interaction for presenting the ATPase domain to the nucleosome. In fact, there are several studies, one of which was published while our work was under review, that identify additional ALC1:nucleosome interaction interfaces(Bacic et al., 2021; Lehmann et al., 2020) (now cited in our manuscript). We have performed the control experiment requested and found that the freely diffusing peptides are unable to inhibit ALC1. This is not surprising considering the high density of ADP-ribosylated molecules near the DNA damage site; if non-nucleosome ADPr were to inhibit ALC1 then this pathway would be incredibly inefficient.

We have replaced this line in the text with several statements that more clearly describe our findings:

Lines 372-375: “Furthermore, freely diffusing macrodomain ligands do not inhibit nucleosome ADPr-dependent stimulation of ALC1 (Figure 5—figure supplement 1D), suggesting that substrate engagement is also influenced by additional ALC1:nucleosome interfaces, of which several have been identified(Bacic et al., 2021; Lehmann et al., 2020).”

Lines 367-368: “Therefore, the modified histone tail:macrodomain interaction is important for ALC1 substrate selectivity.”

The latter statement is also supported by new experiments showing that an ALC1 construct lacking the macrodomain exhibits similar activity towards unmodified and ADP-ribosylated nucleosomes (discussed further below).

(B) The second assay involves an ALC1-mediated remodeling experiment with ADP-ribosylated nucleosomes in the presence of ADP-ribosylated PARP1. The authors, surprisingly, only test ADP-ribosylated PARP1 with unmodified nucleosomes and the rationale for this is unclear if DNA-damage is indeed accompanied by histone ADP-ribosylation, which is also the focus of this manuscript. Results from an assay with ADP-ribosylated H2B containing nucleosomes would clearly indicate if nucleosome ADP-ribosylation remains relevant when PARP1 is also ADP-ribosylated, and if any additive effects are observed from both species being ADP-ribosylated.

We agree that this is an interesting idea and have now performed the requested experiment (Figure5—figure supplement 1K). We found that addition of our most potent ALC1 activating automodified PARP1 construct (PARP1 SerADPr_long_) is unable to further stimulate ALC1 remodeling when an ADP-ribosylated nucleosome substrate is employed. Thus, as Reviewer 2 states, nucleosome ADPr clearly remains relevant in the presence of automodified PARP1. We have pointed this out in text additions as well:

Lines 435-438: “We also found that automodified PARP1 is unable to further stimulate ALC1 remodeling activity when an ADPr nucleosome substrate is employed (Figure 5—figure supplement 1K), which clearly demonstrates that nucleosome ADPr significantly influences ALC1 remodeling even in the presence of automodified PARP1.”

Finally, the pooled mononucleosome remodeling experiment provides no additional information beyond the initial information gained from individual mononucleosome assays and the fact that a ADP-ribosylated histone peptide does not activate ALC1 on non-ribosylated nucleosomes. The peptide in trans assay result already indicated that ALC1 does not stay active after disengaging the ADP-ribosylated peptide. If the authors' goal was to demonstrate that ribosylated nucleosomes do not stimulate activity of ALC1 on non-ribsoylated nucleosomes, or that ALC1 "immediately turns off" after disengagement from a ADP-ribosylated nucleosome, then this needs to be tested in an asymmetric dinucleosome model wherein the two nucleosomes are tethered in close proximity and not diffusing independently of each other. A mixture of varying ADP-ribosylated mononucleosomes at nM concentrations is unlikely to be an approximation of the chromatin context of ALC1, where varying degrees of ribosylation may exist in close apposition. Hence, I suggest the authors reword the conclusion from the pooled experiment to suggest that ALC1 only acts on polyADP-ribosylated mononucleosomes within a mixture of varying ADP-ribosylation states and avoid speculation of its diffusivity.

We agree with this comment and have now performed ALC1 remodeling experiments on asymmetrically ADP-ribosylated dinucleosomes (as described in detail above). Importantly, our new results support our initial claims. As described above, the dinucleosome remodeling activity results are consistent with the automodified PARP1 experiments; PARP1 auto-ADPr and adjacent nucleosome ADPr both tether poly-ADP-ribose in close proximity to unmodified nucleosomes but neither is able to fully stimulate ALC1. We also believe our pooled nucleosome approach is not without value here. While we do show that ADP-ribosylated peptides do not activate ALC1 towards unmodified nucleosomes, there is no evidence that ALC1 assumes a remodeling-competent conformation at any point during these reactions. Use of pooled nucleosomes allowed us to definitively demonstrate that even in a solution wherein robust ALC1 remodeling activity is observed, substrate selectivity is maintained. Together, these experiments are all consistent with our hypothesis that ALC1 preferentially remodels binding-competent nucleosome substrates and target disengagement triggers rapid transition back to an inactive conformation.

Reviewer #3 (Recommendations for the authors):– While authors show that multiple modified histones can be made by this method (including S28 modified H3), claiming broad applicability to preparation of ADPr proteins is a misnomer for the exact reasons listed in the last paragraph of the discussion. The claims of broad applicability should be revised accordingly.

We do not disagree with this assessment and are happy to revise accordingly. As pointed out by Reviewer 3, we initially described the fact that our technology is “…still susceptible to restraints that exist throughout the field of protein chemistry” in the ‘Technical Limitations’ section of our manuscript. We have now modified our abstract, introduction, and discussion to state that our technology is broadly applicable for the preparation of poly-ADP-ribosylated peptides and fully compatible with protein ligation technologies. As discussed above, we have also included new C-terminal ligation chemistry to further support this claim. We are grateful for this suggestion from Reviewer 3, as corresponding edits will ensure that the capabilities of our method are more clearly articulated throughout the manuscript.

– Protein modification community would benefit from better understanding of pros and cons of the polyADP ribosylation method presented here. Given a need for extensive HPLC purification of the mixtures formed in both cases, a side-by-side comparison to experimentally simpler one pot reactions with PARP1:HPF1 would help.

We agree it would be helpful to the reader to have a direct comparison of poly-ADP-ribosylated product yields from the one-step and two-step processes. We have added a sentence to our discussion (Lines 574-578) to point out why the two-step process is required to prepare poly-ADP-ribosylated peptides, and have included an additional Supplementary file that directly compares poly-ADP-ribosylated product from a PARP1:HPF1 reaction (from an unmodified H2B substrate) to a PARP1 elongation reaction (from a mono-ADP-ribosylated H2B peptide). This direct comparison employs the lowest HPF1 concentration required for maximum conversion of unmodified peptide. Thus, the products represent the optimal poly-ADP-ribosylated product yields from a single step reaction. The reader will find that, while it may be possible to isolate milligram quantities of the di-ADP-ribosylated product, it is not feasible to prepare appreciable quantities of the tri- and tetra-ADP-ribosylated species.

Lines 574-578: “The biological two-step ADPr process explains why mono- and poly-ADPr reactions must be separated for scalable preparation of poly-ADP-ribosylated peptides – the high concentrations of HPF1 required for full conversion of unmodified peptides to the mono-ADP-ribosylated species simultaneously inhibits formation of the poly-ADP-ribosylated species (Supplementary file 5).”

– It would be important to assess whether ALC1 itself is ADP ribosylated in the remodeling assays where PARP is supplemented together with NAD+. The supplemental figure 5D, while showing PARP1 is autoribosylated based on the band shift, suggests broad ADP ribosylation signal as well. ALC1 autoribosylation might contribute to relief of its autoinhibition, contributing to the slight enhancement in remodeling observed, rather than a direct interaction with autoribosylated PARP1.

We have performed an ADP-ribose western blot-based analysis of these remodeling assays and observed partial ADPr of ALC1 (Figure 5—figure supplement 1H, discussed in lines 413-414). We initially cautioned the reader against analyzing the unmodified nucleosomes, ALC1, PARP1, and NAD^+^ remodeling reaction (henceforth the *in situ* reaction) because no serine ADPr is present. Therefore, the *in situ* reaction does not appropriately mimic DNA damage-induced ADPr, which predominantly targets serine residues. We performed the *in situ* reaction as described because this precise workflow has been implemented by several previous ALC1 activation studies(Gottschalk et al., 2009; Ooi et al., 2021). Our goal was to provide a direct comparison between the previously employed *in situ* approach and our new analyses. The observation that ALC1 is ADP-ribosylated during this reaction is additional motivation to use extreme caution when evaluating ALC1 activation using the *in situ* workflow.

Importantly, ALC1 ADPr is not a concern when analyzing ALC1 activation by the PARP1 SerADPr_short_ and PARP1 SerADPr_long_ constructs, which were separated from NAD^+^, DNA, and HPF1 prior to the remodeling reaction. Using this approach, we were able to specifically compare ALC1 activation by automodified PARP1 and poly-ADP-ribosylated nucleosomes. We would also like to point out that this control further emphasizes the value of a method to install poly-ADPr at specific sites on proteins for mechanistic interrogation. Moving forward, it may be interesting to explore ALC1 ADPr.

– Remodeling in nuclear extracts: The representative gel results in Figure 6 supplement 1C of remodeling in the presence of nuclear extracts does not support the claim that, with poly ADP ribosylated substrates, remodeling is enhanced compared to that observed with ALC1 knockout extracts. The relative remodeled and unremodeled bands appear similar regardless of the presence of ALC1 in the extract. It appears the specificity of remodeling gained for the ADPr4 substrate in the presence of ALC1 might result from slightly less remodeling of the unmodified substrate in the presence of ALC1 compared to ALC1 knockout extract unmodified substrate remodeling. This point would be apparent if the data was plotted as raw fraction remodeled values or relative remodeled values calculated to the unmodified substrate with WT extract for all experiments in Figure 6E.

We thank the reviewer for this comment and have amended the text to more clearly explain our interpretation of the nuclear extract experiment. Indeed, we observe a change in overall remodeling activity (or basal remodeling activity towards the unmodified nucleosome substrate) when comparing wild-type lysate with ALC1-null lysate. This could be a consequence of subtle lysate preparation variables or represent a cellular mechanism to compensate for the loss of ALC1. Therefore, it is not possible to draw any conclusions based on total remodeling activity. However, this assay does allow us to analyze changes in substrate selectivity between the two lysates. Using our unmodified nucleosomes as an internal normalization factor, we observed increased remodeling activity with the poly-ADP-ribosylated nucleosome substrates (relative to unmodified nucleosome substrates) in wild-type lysates. In contrast, substrate preference was abolished in the ALC1-null lysates, and equal remodeling activity was observed with all substrates regardless of ADPr status. In the updated manuscript, we have carefully edited the text to discuss our results in terms of substrate selectivity and not overall activity. Considering that normalization to unmodified nucleosome remodeling activity within each individual lysate is a key component of data analysis and substrate selectivity, we think this data is most clearly presented to the reader in Figure 6E as is. However, we have included the raw data as a figure supplement as suggested (Figure 7—figure supplement 1E). We have also added text to point out that increased overall activity was observed in the ALC1-null lysate:

Lines 555-558: “We also observed an increase in overall remodeling activity towards all substrates in the ALC1-KO lysate, which may be a consequence of subtle lysate preparation variables or represent a cellular mechanism to compensate for loss of ALC1.”

– More rigorous characterization is needed for the poly ADP ribosylated substrates used with nuclear extracts in Figure 6E to support the claim that no hydrolysis is present of the poly ADP ribose chains. The band shifts due to poly ADP ribosylation are less pronounced as shown in Figure 4C-4E and the band shift is decreased upon addition of extract to H2BS6/H3S10ADPr4 substrate, suggesting chain hydrolysis. Authors should test if the polyADP ribose chains are retained after incubation with nuclear extracts. Alternatively, authors could test nuclear extract activity with and without ALC1 in the presence of PARG to assess if the enhancement is abolished.

We agree it is important to edit the text to include the alternative explanation put forth by Reviewer 3 as follows:

Lines 552-555: “Importantly, partial hydrolysis of nucleosome poly-ADPr was detected following the assay, likely due to the presence of ARH3, PARG, and/or other glycohydrolases. Therefore, the remodeler substrate preference in wild-types lysates may actually be even greater than we observed in this assay.”

In lieu of adding PARG to our assays, we initially designed the lysate experiments to include the H2BS6ADPr_1_ nucleosomes (the PARG reaction product), as well as two nucleosomes that we found to be potent ALC1 stimulators (H2BS6ADPr_4_, and H2BS6/H3S10ADPr_4_) in biochemical assays. Our lysate results are highly consistent with our reconstitution data showing that ALC1, but not similar remodelers such as CHD4, preferentially target the poly-ADP-ribosylated nucleosomes. Therefore, we hope all reviewers agree that the lysate data are an appropriate complement to our reconstitution assays and that all possible interpretations have now been discussed.

– If it still holds, any speculation on attenuation of ALC1 stimulation in nuclear extracts, as opposed to that observed with recombinant components?

References

Bacic, L., Gaullier, G., Sabantsev, A., Lehmann, L.C., Brackmann, K., Dimakou, D., Halic, M., Hewitt, G., Boulton, S., and Deindl, S. (2021). Structure and dynamics of the chromatin remodeler ALC1 bound to a PARylated nucleosome. *eLife 10*.

Bonfiglio, J.J., Leidecker, O., Dauben, H., Longarini, E.J., Colby, T., San Segundo-Acosta, P., Perez, K.A., and Matic, I. (2020). An HPF1/PARP1-Based Chemical Biology Strategy for Exploring ADP-Ribosylation. Cell *183*, 1086-1102 e1023.

Dann, G.P., Liszczak, G.P., Bagert, J.D., Muller, M.M., Nguyen, U.T.T., Wojcik, F., Brown, Z.Z., Bos, J., Panchenko, T., Pihl, R.*, et al.* (2017). ISWI chromatin remodellers sense nucleosome modifications to determine substrate preference. Nature *548*, 607-611.

Gibbs-Seymour, I., Fontana, P., Rack, J.G.M., and Ahel, I. (2016). HPF1/C4orf27 Is a PARP-1-Interacting Protein that Regulates PARP-1 ADP-Ribosylation Activity. Mol Cell *62*, 432-442.

Gibson, B.A., and Kraus, W.L. (2012). New insights into the molecular and cellular functions of poly(ADP-ribose) and PARPs. Nat Rev Mol Cell Biol *13*, 411-424.

Gottschalk, A.J., Timinszky, G., Kong, S.E., Jin, J., Cai, Y., Swanson, S.K., Washburn, M.P., Florens, L., Ladurner, A.G., Conaway, J.W.*, et al.* (2009). Poly(ADP-ribosyl)ation directs recruitment and activation of an ATP-dependent chromatin remodeler. Proc Natl Acad Sci U S A *106*, 13770-13774.

He, X., Fan, H.Y., Narlikar, G.J., and Kingston, R.E. (2006). Human ACF1 alters the remodeling strategy of SNF2h. J Biol Chem *281*, 28636-28647.

Langelier, M.F., Billur, R., Sverzhinsky, A., Black, B.E., and Pascal, J. (2021). HPF1 dynamically controls the PARP1/2 balance between initiating and elongating ADP-ribose modifications. bioRxiv *doi:* https://doi.org/10.1101/2021.05.19.444852.

Lehmann, L.C., Bacic, L., Hewitt, G., Brackmann, K., Sabantsev, A., Gaullier, G., Pytharopoulou, S., Degliesposti, G., Okkenhaug, H., Tan, S.*, et al.* (2020). Mechanistic Insights into Regulation of the ALC1 Remodeler by the Nucleosome Acidic Patch. Cell Rep *33*, 108529.

Leung, A.K. (2014). Poly(ADP-ribose): an organizer of cellular architecture. J Cell Biol *205*, 613-619.

Ooi, S.K., Sato, S., Tomomori-Sato, C., Zhang, Y., Wen, Z., Banks, C.A.S., Washburn, M.P., Unruh, J.R., Florens, L., Conaway, R.C.*, et al.* (2021). Multiple roles for PARP1 in ALC1-dependent nucleosome remodeling. Proc Natl Acad Sci U S A *118*.

Prokhorova, E., Agnew, T., Wondisford, A.R., Tellier, M., Kaminski, N., Beijer, D., Holder, J., Groslambert, J., Suskiewicz, M.J., Zhu, K.*, et al.* (2021). Unrestrained poly-ADP-ribosylation provides insights into chromatin regulation and human disease. Mol Cell *81*, 2640-2655 e2648.

Teloni, F., and Altmeyer, M. (2016). Readers of poly(ADP-ribose): designed to be fit for purpose. Nucleic Acids Res *44*, 993-1006.

Zhang, Y., Wang, J., Ding, M., and Yu, Y. (2013). Site-specific characterization of the Asp- and Glu-ADP-ribosylated proteome. Nat Methods *10*, 981-984.